# Weddell Sea Polynya Analysis Using SMOS-SMAP Apparent Sea Ice Thickness Retrieval

Alexander Mchedlishvili[1], Gunnar Spreen[1], Christian Melsheimer[1], and Marcus Huntemann[1]

[1]University of Bremen, Institute of Environmental Physics, Bremen, Germany

**Correspondence:** Alexander Mchedlishvili (alexander.mchedlishvili@uni-bremen.de)

**Abstract.**

The Weddell Sea is known to feature large openings in its winter sea ice field, otherwise known as open-ocean polynyas. An area within the Weddell Sea region that has repeatedly featured open-ocean polynyas in the past is that which encompasses the Maud Rise seamount. Within this area, after 40 years of intermittent, smaller openings, a larger, more persistent polynya appeared in early September, 2017, and remained open for approximately 80 days until spring ice melt. In this study we present proof that polynya-favourable activity in the Maud Rise area is taking place more frequently and on a larger scale than previously assumed. By investigating thin ($< 50\,\mathrm{cm}$) apparent sea ice thickness (ASIT) retrieved from the satellite microwave sensors Soil Moisture Ocean Salinity (SMOS) and Soil Moisture Active Passive (SMAP), we find an anomaly of thin sea ice spanning an area comparable to the polynya of 2017 over Maud Rise which occurred in September 2018. In this paper, we look at sea ice above Maud Rise in August and September of 2017 and 2018 as well as all years from 2010 until 2020 in an 11-year time series. Using ERA5 surface wind reanalysis data, we corroborate previous findings (e.g., Campbell et al., 2019; Francis et al., 2019; Wilson et al., 2019) on the strong impact that storm activity can have on sea ice above Maud Rise and help consolidate the theory that the evolution of Weddell Sea polynya is controlled by local atmospheric as well as oceanographic variability. Based on the results presented, we propose that the Weddell Sea polynya, rather than being a binary phenomenon with one principal cause, is a dynamic process caused by various different preconditioning factors that must occur simultaneously for it to appear and persist. Moreover, we show that rather than an abrupt stop to anomalous activity above Maud Rise in 2017, the very next year shows signs of polynya-favourable activity that, although insufficient to open the polynya, were present in the region. This phenomenon, as we have shown in the 11-year SMOS record, was not unique to 2018 and was also identified in 2010, 2013 and 2014. It is demonstrated that L-band microwave radiometry from the SMOS and SMAP satellites can provide additional useful information, which helps to better understand dynamic sea ice processes like polynya events when compared to the use of satellite sea ice concentration products alone.

## 1 Introduction

From 1974 to 1976, for three consecutive winters, the satellite microwave radiometer record shows a roughly $250 \cdot 10^3\,\mathrm{km}^2$ opening in the sea ice cover near the Maud Rise seamount that is now known as the Weddell Polynya (Carsey, 1980). After these repeated polynya openings, for the next 40 years the few polynya events were comparatively smaller (Campbell et al.,

2019) and often in the form of a low sea ice concentration (SIC) halo around Maud Rise (Lindsay et al., 2004). 2016 and more so 2017 were the largest and longest-lived polynyas since 1976 (e.g., Swart et al., 2018; Cheon and Gordon, 2019; Campbell et al., 2019; Jena et al., 2019). For the purposes of this paper, we refer to both the 1970s Weddell Polynya (e.g., Carsey, 1980; Martinson et al., 1981; Motoi et al., 1987) and 2010s Maud Rise Polynya (e.g., Cheon and Gordon, 2019; Jena et al., 2019) occurrences as Weddell Sea polynya to signify any sizeable sea ice opening near Maud Rise.

The Weddell Sea polynya is an anomalous opening in sea ice that is generally classified as an open-ocean polynya (Morales Maqueda et al., 2004). An open-ocean or 'sensible heat' polynya is distinguished from the coastal 'latent heat' polynya by being maintained and opened by upwelling and/or mixing as opposed to wind-driven ice advection. The main mechanism that preconditions the Weddell Sea polynya is described by Martinson et al. (1981) as the winter surface layer becoming dense through heat loss and brine rejection from ice formation, resulting in density overturning (Martinson et al., 1981; Martinson and Ianuzzi, 1998) which deepens the mixed layer and initiates deep convection. Deep convection and heat ventilation into the mixed layer are thought to be primary causes for the Weddell polynya (Martinson et al., 1981; de Steur et al., 2007; Wilson et al., 2019; Cheon and Gordon, 2019).

One of the primary mean-state factors that precondition polynya at Maud Rise is the lack of strong stratification (Gordon and Huber, 1990; McPhee et al., 1996). Weak stratification leaves the ocean susceptible to density overturning and is facilitated by winter surface mixed layer salt content as well as the topography present at Maud Rise. Another polynya-favourable factor that is reported on is the anomalously warm waters found over the flanks of the rise, with a colder cap of water lying over the top of the rise (e.g., Gordon and Huber, 1990; Bersch et al., 1992; Martinson and Ianuzzi, 1998; Lindsay et al., 2004; Muench et al., 2001; Wilson et al., 2019). Muench et al. (2001) went further to describe how interactions between the mean flow (eastern limb of the Weddell Gyre) and topography (Maud Rise) preconditions the area above Maud Rise for anomalously high vertical heat fluxes, which favors thinner sea ice. Due to the presence of Maud Rise and the surrounding elevated Warm Deep Water, the 23–year mean sea ice concentration (SIC) for the months of July through November (1979–2001) shows a distinct halo of low ice concentration with a diameter of about 300 km (Lindsay et al., 2004). A factor that contributes to the existence of said halo and, by extension, the Weddell Sea polynya that occur locally is the cyclonic eddies that adhere to the flanks of the rise (de Steur et al., 2007; Holland, 2001). Northeast of Maud Rise specifically, is where Holland (2001) suggested water columns leaving the seamount are stretched and acquire cyclonic vorticity thereby applying divergent strain to the sea ice layer from below.

Cheon and Gordon (2019) described the 2016 and 2017 Weddell Sea polynya formation process in detail by explaining the preconditioning as well as the large-scale climate events that lead to it. They discussed large-scale atmospheric influences, specifically the influence of positive Southern Annular Mode (SAM) which intensifies the negative wind stress curl over the Weddell Sea and thereby the Weddell Gyre which, in turn, weakens upper ocean stratification. In addition to large-scale climate processes, localized atmospheric forcing has been shown to impact the ice layer from above (e.g., McPhee et al., 1996; Goosse and Fichefet, 2000; Francis et al., 2019; Campbell et al., 2019; Wilson et al., 2019; Heuzé et al., 2021). As early as 1996, during the Antarctic Zone Flux Experiment (McPhee et al., 1996) it was shown that storms featuring gusts of up to 25 m/s can produce ocean heat fluxes that exceed 100 W/m$^2$. Under weak stratification that encourages heat ventilation from below,

the thermocline is exposed to more intense wind-driven turbulent mixing aiding the formation of the polynya (Wilson et al., 2019; Campbell et al., 2019). Francis et al. (2019) went further to state that anomalous atmospheric influence triggers polynya formation which they hypothesized was the case for the 2017 Weddell Sea polynya. Ice divergence due to strong winds enables rapid ice production and brine rejection that prevents stabilization from ice melt as wind-driven turbulent mixing entrains warm
and saline water into the surface mixed layer (Campbell et al., 2019).

This study contributes to the recently-emerging understanding of direct atmospheric influence over the Maud Rise region and supports the notion that the Weddell Sea polynya is not purely an ocean-driven polynya (Heuzé et al., 2021). However, the primary investigation is done during polynya-free years. Through analysis of the thin ($< 50\,\mathrm{cm}$) sea ice thickness (SIT) product (which for the purposes of this study we relabel as apparent SIT or ASIT for reasons elaborated on in section 2.1) from
70 the spaceborne passive microwave sensors Soil Moisture Ocean Salinity (SMOS) and Soil Moisture Active Passive (SMAP), we aim to challenge the notion that anomalous activity atop Maud Rise is purely a binary phenomenon. Rather, the Weddell Sea polynya is the result of independent as well as dependent preconditioning effects that occasionally but not exclusively interfere with one another constructively to form the polynya, like in 2016, 2017 and mid-1970s. Using the ASIT retrieval over years where the polynya did not occur, we aim to identify low sea ice thickness areas that demonstrate anomalous behaviour
taking place in the absence of the Weddell Sea polynya suggesting that the existence of polynya-favorable conditions, although present, are insufficient to produce the polynya. Previous studies used satellite sea ice concentration to analyse the size and development of the polynya. Since 2010, the SMOS satellite has allowed us to analyse thin ice area anomalies, i.e., thinning of ice on the same scale as the polynya that is subject to similar underlying causes.

## 2 Data and Methods

For this study a combined SMOS-SMAP SIT retrieval and SMOS SIT retrieval (for time periods preceding the installment of SMAP in 2015) are used to identify low sea ice thickness areas above Maud Rise. In addition, the ARTIST sea ice (ASI) algorithm is used to access sea ice concentration and ERA5 meteorological reanalysis data is used to look at winds at the surface level.

### 2.1 SMOS-SMAP Apparent Sea Ice Thickness Retrieval

The space-borne passive microwave sensors Soil Moisture Ocean Salinity (SMOS) and Soil Moisture Active Passive (SMAP) are working at $1.4\,\mathrm{GHz}$ (L-band) which allows information on the thickness of thin sea ice (SIT) to be obtained (Paţilea et al., 2019; Huntemann et al., 2014; Tian-Kunze et al., 2014). From modeling and observations it has been established that emission at L-band show sensitivity to ice thickness up to about 50 cm (Kaleschke et al., 2010). The atmosphere has negligible influence on surface emission at L-band (Zine et al., 2008; Kaleschke et al., 2013). The footprint size of both sensors is around $40\,\mathrm{km}$
(Paţilea et al., 2019).

The SMOS-SMAP SIT retrieval builds upon its predecessor SMOS SIT retrieval (Huntemann et al., 2014). The SMOS SIT retrieval uses the average of horizontally and vertically polarized brightness temperatures as well as the polarisation difference

(i.e., the difference between horizontally polarized and vertically polarized brightness temperature) averaged over the incidence angle range between 40° and 50° from the synthetic aperture antenna observations. SMAP, on the other hand, uses a real

aperture antenna and observes the earth surface at a fixed incidence angle of 40° resulting in a narrower swath than SMOS (Paţilea et al., 2019). The combined SMOS-SMAP thin ice sea ice thickness retrieval improves the SMOS retrieval by adapting it to SMAP with the modification that uses fixed 40° incidence angle observations instead of average in the range 40 to 50°. This is achieved by fitting a function to the brightness temperature to incidence angle relation for all overflights of a geographic location of one day. This results in an average resolution of approximately 43 km. In addition, a linear regression between the

SMOS and SMAP brightness temperatures at a 40° incidence angle is performed to align the brightness temperatures of the two instruments with a root mean square difference at horizontal and vertical polarization of 2.7 and 2.8 K, respectively (Paţilea et al., 2019). The combined SIT retrieval offers more stable sea ice thicknesses, which are less influenced by radio-frequency interference. Because of the up to 12 hour difference in the Equator crossing time between SMAP and SMOS, ice thicknesses retrieved from the daily mean brightness temperatures are more likely to include more of the brightness temperature variations within a day, which also helps the stability of the retrieval. Therefore, we prefer the use of the combined SMOS-SMAP SIT

retrieval above the SMOS-only one for the study of SIT over Maud Rise in cases when it is available.

    Sea ice concentration (SIC) data (Section 2.2) is used for comparison with the SIT data. The SMOS-SMAP retrieval algorithm assumes near-100% SIC when retrieving SIT and since we look at a region prone to polynya and low SIC (Lindsay et al., 2004), it is important to consider this factor. The SMOS-SMAP SIT retrieval has no SIC dataset correction implemented be-

cause uncertainty of SIC algorithms at high concentration and their covariation at thin thicknesses will cause high amounts of error (Paţilea et al., 2019). Using SIC maps and data in combination with SIT counterparts, we can better infer the location and degree of error in our SIT retrieval. As a general rule, the SMOS-SMAP as well as SMOS SIT retrievals tend to underestimate the sea ice thickness at SIC below 100%; the degree to which this underestimation occurs is heavily influenced by the SIC value. However this interaction between SIC and SIT retrievals is two-sided as most sea ice concentration algorithms show

less than 100% SIC for thicknesses below 30 cm (Heygster et al., 2014). Paţilea et al. (2019) estimated the uncertainty at 90% SIC from SIT values up to 50 cm. The higher the SIT that is being retrieved the higher the uncertainty e.g. an area that is 90% SIC and 50 cm thick is expected to be retrieved as only 28 cm. This is simply the limitation imposed by the penetration depth into sea ice at the given frequency (Kaleschke et al., 2010) and is also why the retrieval is capped at 50 cm as any attempt to retrieve thicker SIT values would be accompanied by an even higher amount of error.

Both the SMOS-SMAP and SMOS are empirical retrievals that were initially developed for monitoring the sea ice thickness of growing sea ice in the Arctic during freeze-up through comparison with a Cumulative Freezing Degree Days (CFDD) model and thereafter calibration and validation using observations (Huntemann et al., 2014). As such, this compromises the validity of the sea ice thicknesses retrieved in areas that are prone to polynya. While the degree by which the ice thins is difficult to quantify in terms of uncertainty, our analysis has shown that the pattern of thin ice anomalies above Maud Rise is not random

nor is it identical to the distribution of low SIC areas, and instead adheres to the general understanding of processes present in the region. As a result for the purposes of this study, we present the retrieved SIT values as apparent sea ice thickness (ASIT) that is meant to depict the distribution of sea ice that is subject to sea ice thinning rather than the exact thickness of each

individual pixel. We take this approach in part due to the low sea ice concentrations that are to be expected near Maud Rise (Lindsay et al., 2004), as well as the melting conditions at the end of the winter season which neither of the SMOS retrievals were originally made for. While the SMOS-SMAP retrieval from a physical point of view works for both hemispheres there is a lack of validation data in the Antarctic. Uncertainty from flooded ice and slush caused by snow pushing down the sea ice such that water floods from the sides or from below through the cracks is expected to influence the SIT retrieval. These events are not typical for thin ice and can happen in both hemispheres, but might be more common in the Antarctic. The uncertainty caused by the flooding cannot be assessed without in-situ measurements. As such, no attempt at calculating ice volume was made in this study so as not to carry over errors that affect the retrieved ASIT values.

## 2.2 ASI Ice Concentration Algorithm

The ARTIST Sea Ice (ASI) algorithm calculates SIC from the difference between brightness temperatures at 89 GHz at vertical and horizontal polarizations which are retrieved by the Advanced Microwave Scanning Radiometer 2 (AMSR2) onboard the Global Change Observation Mission-Water (GCOM-W1) satellite. This polarization difference is then converted into SIC using pre-determined fixed values for 0% and 100% SIC polarization differences known as tie points. It is known from surface measurements that the polarization difference of the emissivity near 90 GHz is similar for all ice types and much smaller than for open water (Spreen et al., 2008). At 89 GHz the spatial resolution with 5 km is the highest of all AMSR2 channels but the atmospheric influence is high also. This effect is dealt with in a bulk correction for atmospheric opacity and by implemented weather filters over open water. Because the Bootstrap (BBA) (Comiso et al., 1997) algorithm uses the 19 and 37 GHz channels, which are less sensitive to atmospheric phenomena, it is also used to essentially filter the produced ASI SIC concentration by setting SIC to zero where the Bootstrap algorithm retrieves less than 5% SIC. The finalized and filtered ASI SIC data has $6.25\,\text{km}^2$ grid resolution.

## 2.3 ERA5 Climate Reanalysis

ERA5 atmospheric reanalysis data is used to study direct atmospheric forcing on the opening of the polynya as well as on anomalous regional sea ice thinning to investigate whether the Weddell Sea polynya is purely ocean-driven or maintained by a combination of both processes.

ECMWF Reanalysis 5th Generation (ERA5) provides a detailed record of the global atmosphere, land surface and ocean from 1950 onwards. It replaces the ERA-Interim reanalysis (spanning 1979 onwards) and is based on the Integrated Forecasting System (IFS) Cy41r2. ERA5 benefits from a decade of developments in model physics, core dynamics and data assimilation (Hersbach et al., 2020). In addition to a significantly enhanced horizontal resolution of 31 km, compared to 80 km for ERA-Interim, ERA5 has hourly output throughout.

Campbell et al. (2019) reported that there was sufficient agreement between mean sea level pressure (MSLP) data obtained from the SANAE-AWS weather station and the nearest ERA-Interim grid cell for ERA-I to be used in gathering signs of storm activity as it skillfully represented MSLP variability near Maud Rise. ERA5 is a reanalysis with a higher temporal and spatial resolution than ERA-I. It improves upon its predecessor in terms of information on variation in quality over space and time as

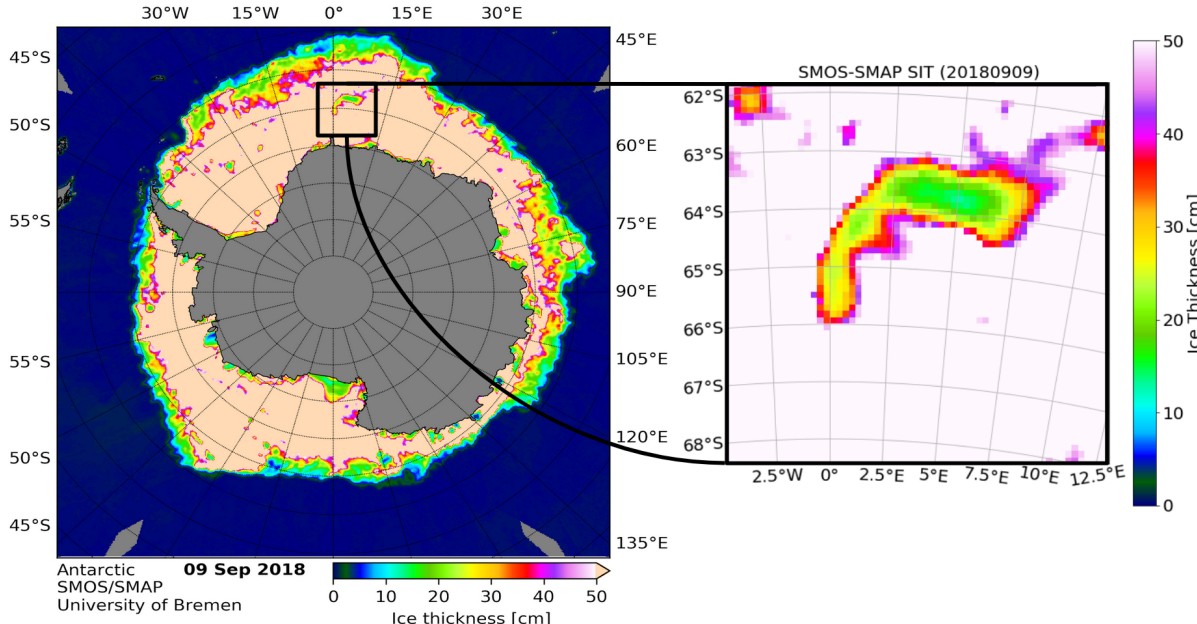

**Figure 1.** Left: SMOS-SMAP apparent sea ice thickness (ASIT) retrieval for the sea ice zone around Antarctica from 9 September 2018. The segment contained in the black square depicts sea ice above Maud Rise (66°S,3°E). Right: the zoomed-in local SMOS-SMAP ASIT of the outlined region which serves as the boundaries of all time series generated in this study.

well as an improved troposphere modelling (Hersbach et al., 2020). As a result, for the purposes of this study, it should offer a better, or at least identical, assessment of the wind speeds near Maud Rise that are going to be cross-referenced with the presented ASIT retrievals in this study.

## 3 Results

The left image in Fig. 1 shows a standard Southern Hemisphere SMOS-SMAP ASIT retrieval at grid resolution of 12.5 km (actual SIT product resolution is lower, section 2.1). The black frame (northwest corner: 61.78°S, 3.57°W, southeast corner : 67.88°S, 13.11°E), which is zoomed in on the right, shows austral winter sea ice above Maud Rise (66°S,3°E) and it is the area this study focuses on. All time series as well as individual maps included are evaluated for and depicting the area within the black frame, respectively.

Figure 2 shows the full 11-year record of SMOS sea ice thickness from 2010 to 2020 above Maud Rise. For a detailed analysis of the anomalous sea ice behaviour atop Maud Rise, the years 2017 and 2018 are chosen. In 2017 the largest Weddell Sea polynya of this century occurred, and the following year 2018 also exhibits anomalous thin ice behaviour as will be shown later. The full 11-year time series will be discussed at the end of the Result section. In 2017 the polynya showed the largest extent and is open the longest time period since the 1976; in 2018 there was no polynya but sea ice thinning is observed over

multiple weeks. Here the advantage of the SMOS-SMAP ice thickness retrieval shows its strength by detecting anomalous sea ice behaviour where traditional sea ice concentration datasets cannot. This section presents findings that suggest a previously unrecognized similarity between the two September anomalies.

Fig. 3 shows maps of the SMOS-SMAP ASIT development from 11 to 13 September in 2017 and Fig. 4 for the same three days in 2018. The SMOS-SMAP ASIT maps are accompanied by ASI SIC counterparts at a nominal resolution of $6.25\,\mathrm{km}$

covering the same region for comparison.

The 2017 Weddell Sea polynya is a well-documented event and its preconditioning as well as existence until melt of that year has been shown via in situ ocean data and also satellite imagery; most commonly via SIC retrieval (e.g., Campbell et al., 2019). Here the advantages of SMOS-SMAP ASIT retrieval are limited by the high open water fraction but nevertheless help to demonstrate the full extent of the anomaly that SIC maps of the region can only partially depict (see Fig. 3). Additionally,

the ASIT maps in Fig. 3 show a broader gradient of ASIT encompassing a larger area on all sides of the polynya than the sharp SIC gradients surrounding the polynya in the ASI maps. While the coarser $12.5\,\mathrm{km}$ SMOS-SMAP ASIT data product is less resolved than the $6.25\,\mathrm{km}$ ASI SIC data counterpart, we do not expect any underestimation of area by ASI retrieval as it is estimating open water as a percentage on a sub-footprint scale. In Fig. 3 we can see a substantial difference between both the scale and gradient between the SMOS-SMAP ASIT and ASI SIC data maps.

Similarly for 2018, we show a side-by-side comparison of SMOS-SMAP ASIT and ASI SIC maps (Fig. 4). On 11 and 12 September 2018, the low SIC halo (Lindsay et al., 2004) can be seen in the ASI SIC maps as a thin ring surrounding Maud Rise. Interestingly the SMOS-SMAP ASIT map counterparts for that time period instead show a wide-scale thinning; what we will refer to as the SIT anomaly from now on. While the the bottom portion of the halo is not visible in the ASIT record, the northeastern crescent is enlarged, indicating a much wider area of anomalous activity than suggested by the SIC maps.

13 September 2018 tells a different story wherein the SIC map can no longer distinguish the halo feature whereas its SMOS-SMAP counterpart still contains the thinning from previous days. For a better resolved image from visual MODIS data of both the Weddell Sea polynya of 2017 and the SIT anomaly of 2018 see Fig. A2. Such images are only available for cloud free conditions and thus cannot be used to monitor the polynya development in detail.

Fig. 5 depicts the Weddell Sea polynya until the end of September 2017 as well as the weeks leading up to the event.

Atmospheric data (Fig 5a) in the form of wind speed derived from 10-m $u$ and $v$ components of wind velocity vectors at $1000\,\mathrm{hPa}$ are presented as daily average (in blue) and maximum (in red) magnitude in the region of interest. Interpreting Fig. 5a as compared to the lower polynya area and thickness plots, we see that the highest maximum (in red) and mean (in blue) wind speed both coincide with the 13 September polynya opening date. This agrees with the general conclusions reached by both Campbell et al. (2019) and Francis et al. (2019). From the ASI SIC record (Fig. 5b), we can see both the similarities

it shares with the ASIT record (Fig. 5c) as well as clear differences that will be further discussed below. Important to note is that the blue line in both SIC and ASIT records represents the area that is classified as open water, so 0% sea ice concentration and 0 cm thick ice (so no ice at all), respectively. These lines are also present in the 2018 Fig. 6b and 6c but are consistently at $0\,\mathrm{km}^2$ and therefore hidden because of the overlap with low SIC and low ASIT lines.

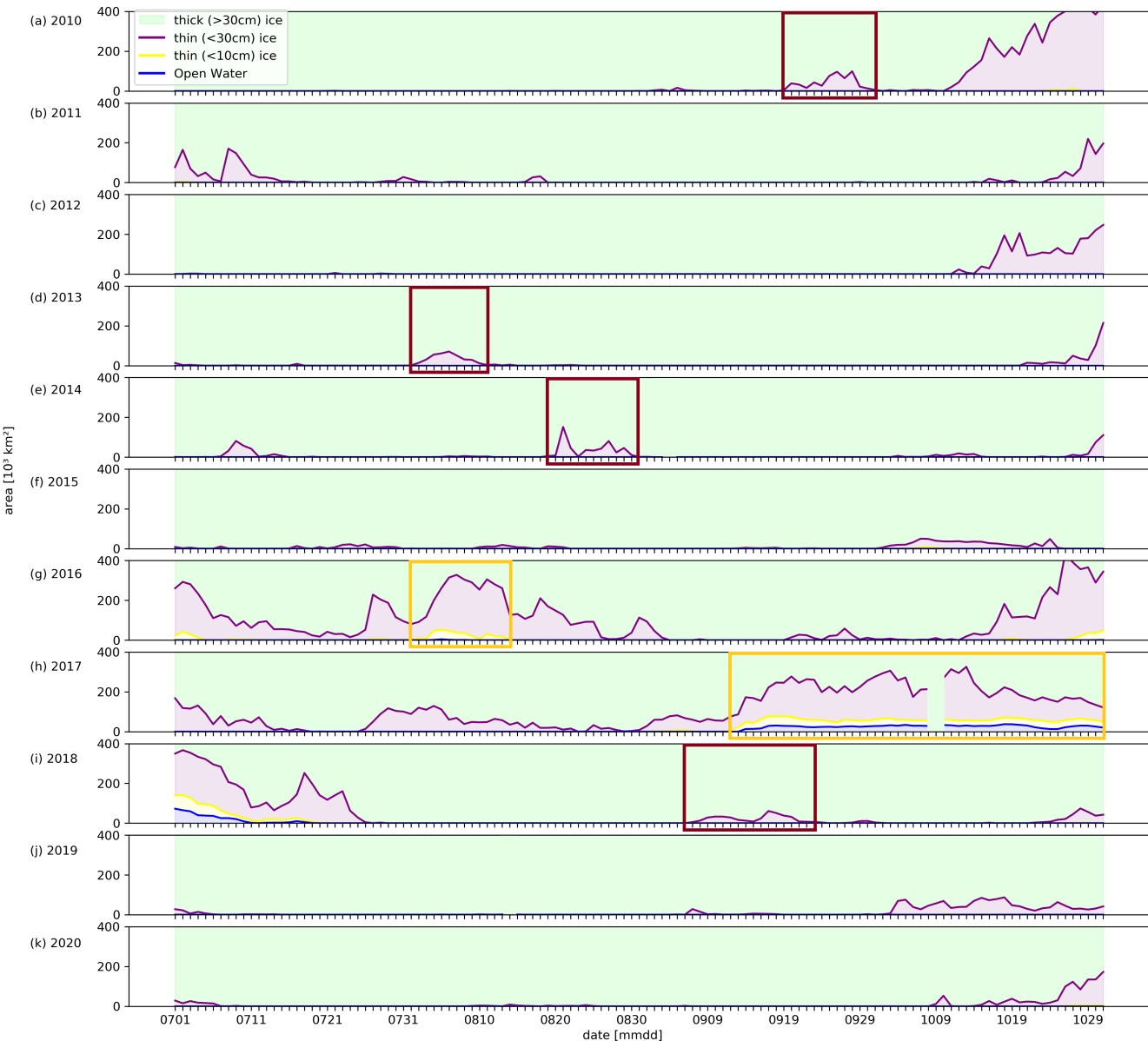

**Figure 2.** (a-k) the SMOS apparent sea ice thickness (ASIT) retrieval time series from 2010 to 2020 over the area of interest outlined in Fig. 1. Each line represents sea ice below a thickness threshold shown in the legend in the top left (blue: open water, yellow: < 10 cm ice, red: < 3 cm ice). Each filled-in area represents sea ice within the range set by the lines (blue: open water area, yellow: 0–10 cm ice, red: 10–30 cm ice, green: > 30 cm ice). Polynya events are highlighted in yellow whereas ice thinning anomalies are highlighted in red (see also maps in Fig. 7). Years 2017 and 2018 are discussed in more detail in this manuscript.

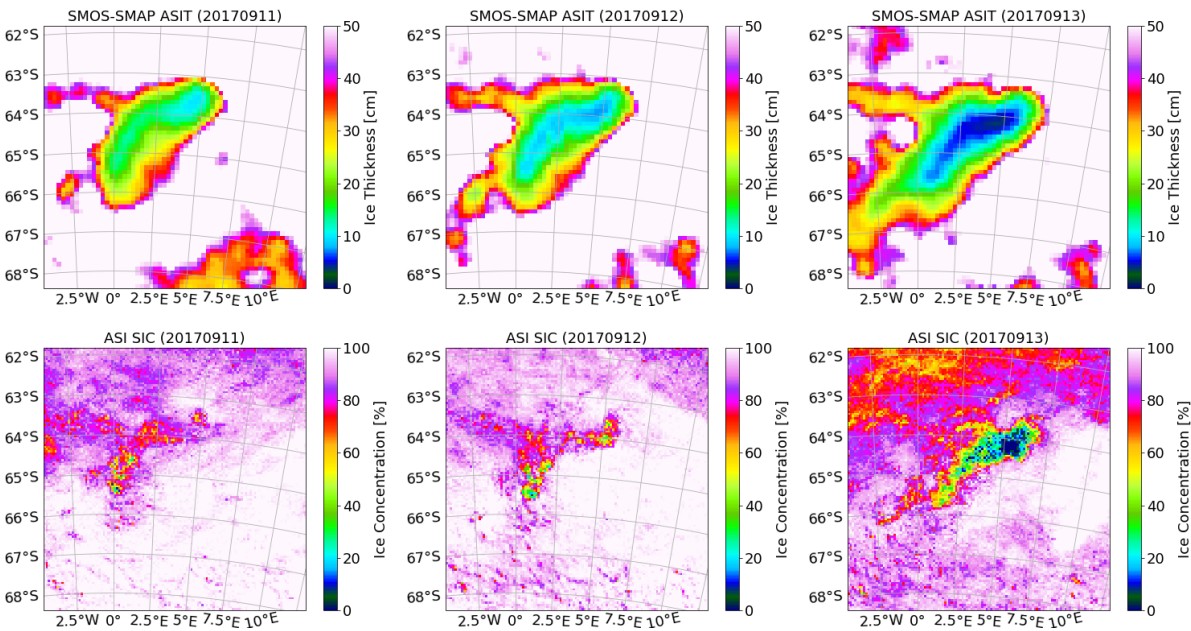

**Figure 3.** SMOS-SMAP ASIT (top row) and ASI SIC (bottom row) retrieval of the days leading up to the 2017 Weddell Sea Polynya: 11–13 September 2017.

August and September 2018, shown in the time series plots of Fig. 6, is the time period of interest for this research, where the area that featured a polynya the year prior shows a low SIT anomaly. Looking at the ASIT record for the two months (Fig. 6c), we can see the area thinner than 50 cm (brown line) exceeds $250 \cdot 10^3$ km$^2$ (17–20 Sep) and ice thinner than 20 cm (green line) is detected on multiple days (8–12 Sep, 16–19 Sep) in an anomaly spanning almost the entirety of September. SIC time series (Fig. 6b) seems to vaguely reflect the SIT anomaly of late September by sporadic episodes of below 80% SIC (purple line) but does not describe the anomaly like the ASIT record does. No area below 40% SIC is detected and thus no significant open water area prevailed that year. Notably we can see the highest mean (9–10 Sep) at the start of the SIT anomaly and the highest maximum at the same time as the peak of the anomaly (17–18 Sep).

In Fig. 2 we show the entire 11-year SMOS record in the form of a time series. For the time periods highlighted with colored frames, maps of the ice anomaly are shown in Fig. 7. Time frames highlighted in yellow are the 2016 and 2017 Weddell Sea polynya events whereas red frames surround the periods of ice thinning anomalies. Notably, ice thinning anomalies seem to have a higher frequency of occurrence.

## 4 Discussion

The individual maps Fig. 3 and Fig. 4 for 2017 and 2018, respectively, are useful for accessing fine details of low ASIT distributions as well as comparing the ASIT retrieval with ASI SIC. By capturing the low sea ice thickness anomaly in 2018

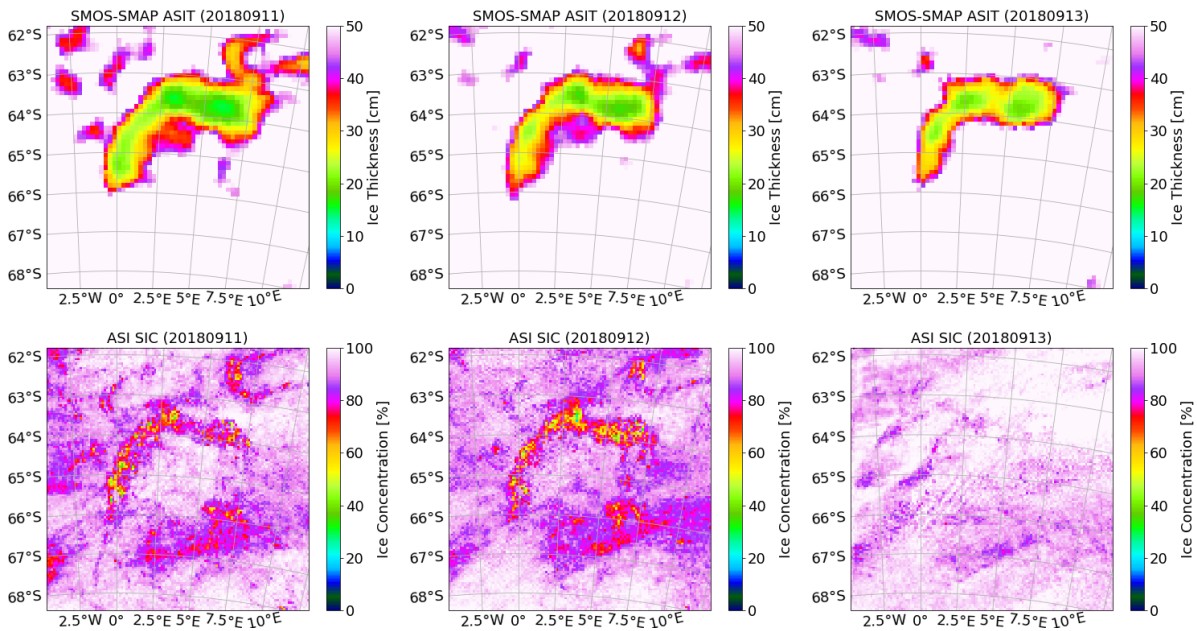

**Figure 4.** SMOS-SMAP ASIT (top row) and ASI SIC (bottom row) retrieval of the days during the 2018 sea ice anomaly: 11–13 September 2018.

and at the beginning of the 2017 polynya event in the ASIT record we can infer that there were residual polynya-favourable
effects that produced a forcing that was insufficient to open the polynya but sufficient to still impact the overlying sea ice. This
is similar to the 1970s polynya cases, where the 1973 smaller polynya preceded the larger Weddell Sea polynya visible from
1974 to 1976 (e.g., Martinson et al., 1981; Motoi et al., 1987; Comiso and Gordon, 1998; Cheon and Gordon, 2019). Cheon
and Gordon (2019) attribute the lack of any polynya in 2018 in part to the positive state of the Southern Annular Mode inducing
fresh surface water conditions effectively capping warmer deep water convection and the weakening of the Weddell Gyre in
the years that followed its peak activity in 2015 and 2016. This study aims to present a more complete perspective where rather
than an abrupt change from the largest Weddell Sea polynya in 2017 to lack thereof we observe a waning of this phenomenon
with peak activity in 2017.

In order to analyse the time periods during which the polynya of 2017 (Fig. 5) and the sea ice anomaly of 2018 (Fig. 6)
occurred, we view the respective time series. In Fig. 5c we see a progression of events in terms of ASIT of how the 2017
polynya formed and expanded. First and foremost we observe a major regional ice thinning in early August that peaks on
the fourth of August. Looking at Fig. 5b we can see how much smaller the area affected by SIC variations is and how it is
different in behaviour to the ASIT time series. At this point in time, low-SIC area is small and predominantly above 80%. This
is especially true during the brief period (6–12 Sep) leading up to the polynya, which is promising because it suggests a lack
of low SIC-induced ASIT values due to the SMOS-SMAP ASIT retrieval full ice cover assumption. In total, compared to the
240 $50 \cdot 10^3$ km$^2$ of below 100% SIC area, sea ice thinner than 50 cm spans over $300 \cdot 10^3$ km$^2$ of the region of interest. Following

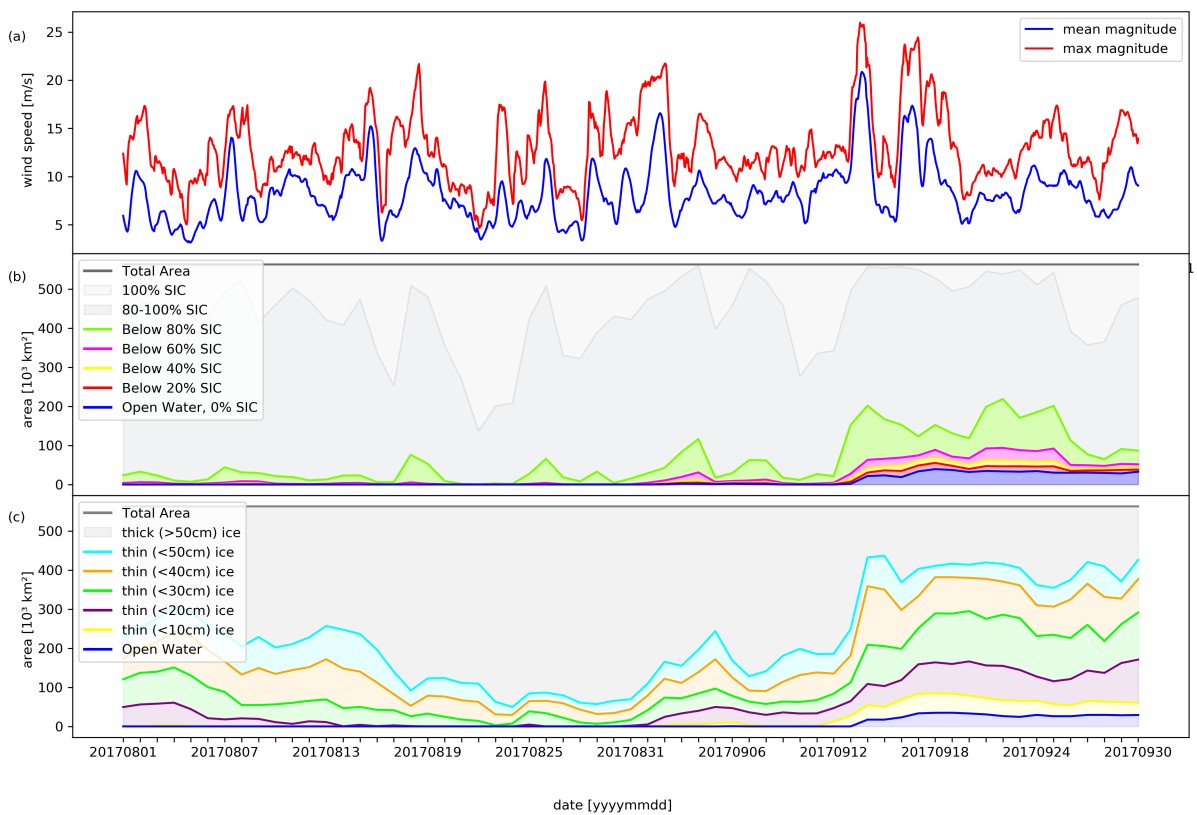

**Figure 5.** August–September 2017: (a) Daily ERA5 wind speed at 1000 hPa (red: daily maximum; blue: mean computed for the region outlined by the bounding box in Fig. 1). (b) ASI SIC where each line represents ice that falls below a SIC value shown in the legend. Each filled-in area represents sea ice within the range set by the lines such that all the colours match the colour of the lines that are directly above them (e.g. the green shading below the green line represents 60–80% SIC area). The uppermost grey line represents total area ($562.5 \cdot 10^3$ km) and the variation in shading below it is as follows: light grey is for all area that is 100% SIC and the darker grey is reserved for all that fall between 80 and 100% SIC. (c) SMOS-SMAP ASIT where each line represent sea ice below a thickness shown in the legend. Each filled-in area represents sea ice within the range set by the lines such that all the colours match the colour of the lines that are directly above them (e.g. the cyan shading below the cyan line represents 50–40 cm ASIT area). The grey line represents total area ($562.5 \cdot 10^3$ km) like in the SIC plot and the grey shading represents sea ice that is identified to be $> 50$ cm. All plots cover the area of interest outlined in Fig. 1.

the period mentioned (6–12 Sep), we see the sudden peak (12–13 Sep) in both lower sea ice concentrations and thin sea ice. Based on Fig. 3, we see that this smaller opening in sea ice (Campbell et al., 2019), paved way to the Weddell Sea polynya of the year 2017.

Campbell et al. (2019) report on the highly variable salt content in the vicinity of Maud Rise during this time period in 2017 indicative of cycles of melt and refreeze. Thus the negative feedback of melting sea ice wasn't able to fully re-stratify the ocean. This is reflected in the ASIT record (Fig. 5b) as large parts of the ice pack appear to be thin ice, especially when

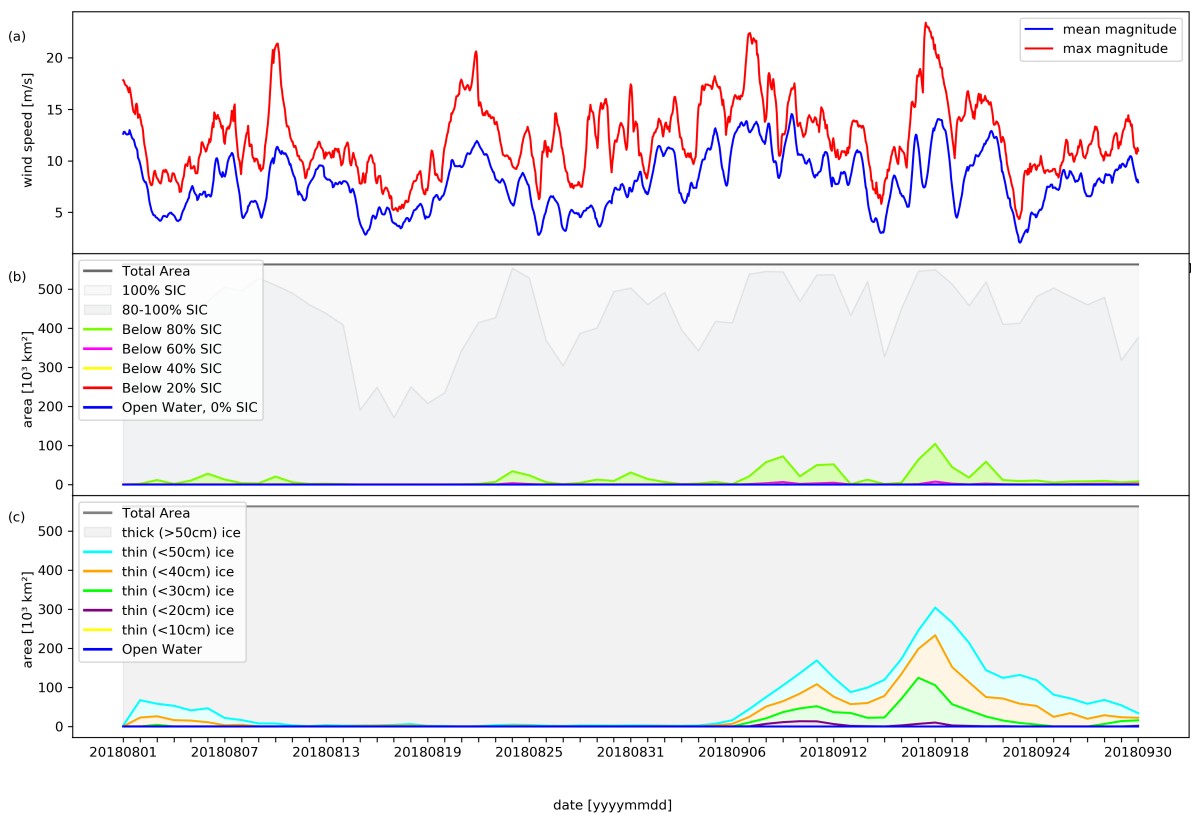

**Figure 6.** Same format as Fig. 5 for August–September 2018. See Fig. 5 caption for specifications.

compared to (Fig. 6b) where there is no thinning prior to the 2018 anomaly. Weak stratification coupled with strong winds (Campbell et al., 2019) (Fig. 5a, 1–2 Sep) enhanced turbulent mixing and entrained heat into the surface mixed layer. Francis et al. (2019) report on the unusual amount of cyclones during 2017 austral winter while Martinson and Ianuzzi (1998) detail how such events may serve to reduce the bulk stability of the water column. In Fig. 3 we can see the much larger scale effect this is having on ASIT rather than SIC in Fig. 5b and Fig. 5c, respectively. While the thinning can be attributed to the entrained heat, salt is also entrained from water below which reduces stability (Martinson and Ianuzzi, 1998). In addition to atmospheric effects, Campbell et al. (2019) attribute the 2016 and partly 2017 polynyas to salinity fluxes from deep water into the mixed layer. Thus, the sea ice melts from below due to added heat and the melt-water is unable to stabilize the water column due to the salinity flux, facilitating thinning of sea ice and its eventual melting that results in the Weddell Sea polynya of 2017.

Fig. 6 shows that 2018 is less anomalous than 2017 for the first one and half months until the sea ice anomaly begins to form on the 6 of September 2018. There is an initial thinning and occasional sporadic "below 80%" SIC events distributed throughout the period. Notably, the events on 24 August and 31 August, seen in Fig. 6b could be lead openings in thick pack ice as there is no thinning recorded in the ASIT retrieval for those days. The sea ice anomaly itself, as can be seen in Fig. 6c, is very well defined in the ASIT record and has a clear beginning and an end. Notably, of the two consecutive low ASIT area

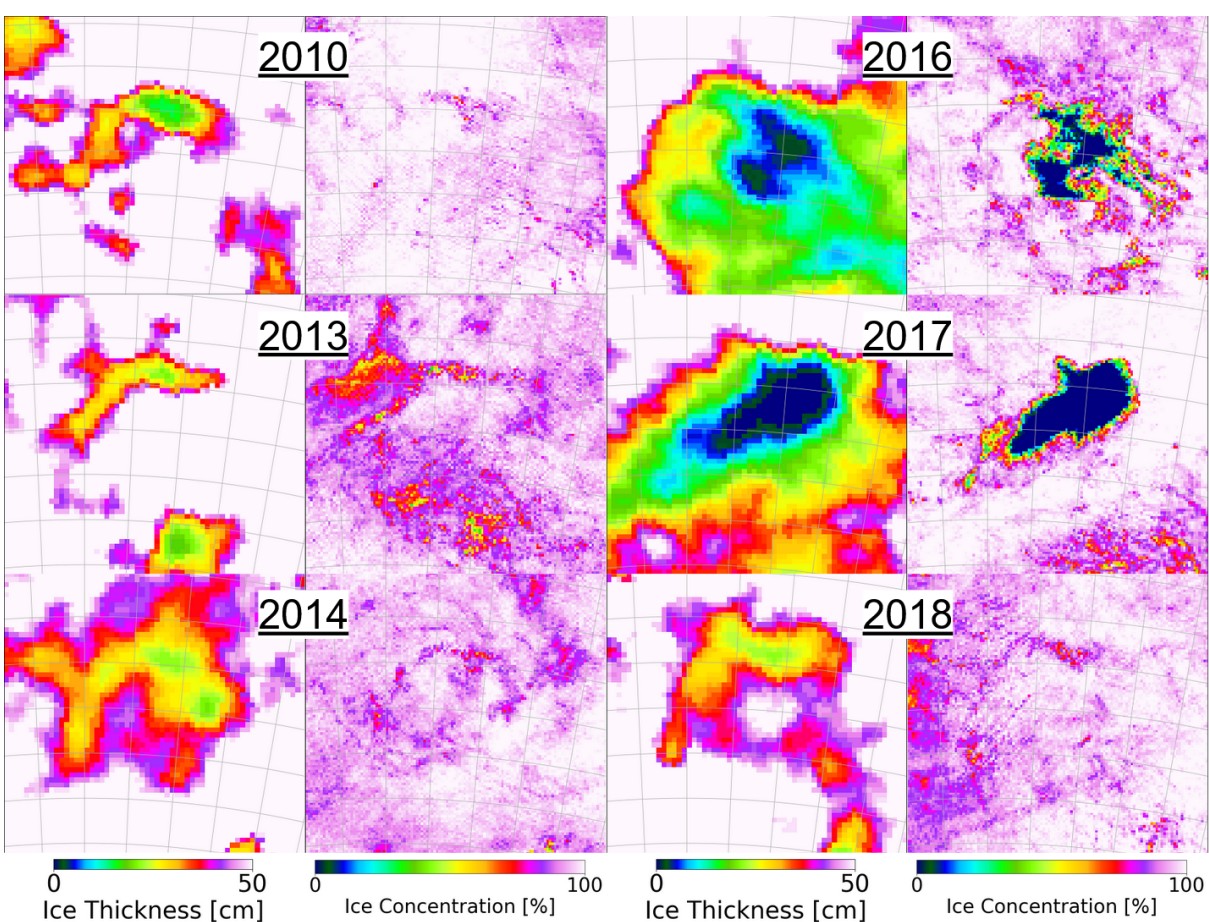

**Figure 7.** ASI SIC and SMOS ASIT retrieval maps covering all major thinning events that can be seen in the 11-year SMOS record (Fig. 2). All maps cover the area of interest outlined in Fig. 1 with and all the grid-lines remaining identical (removed here to maximize the size of each map).

peaks (7–13 September and 15–21 September), the first is characterized by ice thinner than 20 cm for a longer period that the second which instead has a much larger area thinner than 50 cm. This anomaly follows a period of relatively strong mean and maximum wind speed from 3 August to 13 Sep (Fig. 6) towards the East and Southeast directions (Fig. 9) that could imply wind-driven turbulent mixing influencing the sea ice anomaly in much the same way the added heat and salinity fluxes

preconditioned the polynya the year before (Campbell et al., 2019). Due to the lack of in-situ ocean data analysis from the 2018 period and the absence of such analysis in this study, any proposed ocean-driven polynya preconditioning is purely speculative. Nevertheless, it can be assumed that the negative feedback of melting sea ice freshened the mixed layer thereby stabilizing the water column and suppressing further exchange with Warm Deep Water from below (Wilson et al., 2019).

Through the comparison of our ASIT and SIC data with ERA5 atmospheric data we can speculate what wind conditions are

270 favourable for polynya formation. Fig. 8 show the wind conditions on 13 September 2017 where for several hours strong winds

(20 m/s) prevailed above the region of interest suggesting heavy storm activity, corroborating the findings of Campbell et al. (2019) and Francis et al. (2019). In contrast, regional winds in September 2018 are consistently below 15 m/s (mean) and areas of strong wind seem to be localised around rather than on top of Maud Rise (not shown) like on September 7 Fig. 9. It is hard to say whether stronger storms during the sea ice anomaly of 2018 would have caused a polynya to open as it is not known quantitatively how much different factors contribute to the formation of the polynya. But it is clear that atmospheric forcing is a strong contributing factor especially towards the start of the polynya. Thereafter also turbulent mixing of warm salty water plays an increasing role (Campbell et al., 2019).

Lastly, we use the SMOS ASIT retrieval instead of the combined SMOS-SMAP to also include years before 2015 (the year when SMAP was put into orbit) to make a consistent 11 year SMOS ASIT time series over the months of July, August, September and October (Fig. 2) that fully includes the freezing periods of the relevant region over the years. Notably, the sea ice thinning of 2018 is by no means an isolated event and the Maud Rise region seems to be regularly subject to sea ice thinning events. While the SIC record offers two prominent anomalous events: the Weddell Sea polynya of 2016 and 2017, respectively (highlighted in yellow in Fig. 2), it is through the ASIT retrieval that we identify all other anomalies that have occurred over the years. Specifically, years in which the polynya did not occur but still showed signs of ice thinning are 2010, 2013, 2014 and 2018 (not counting thinning episodes that follow freeze-up or precede melt); they are highlighted in red in Fig. 2. In Fig. 7, we look specifically at the events highlighted in Fig. 2 to get a better picture of ice thinning anomalies and polynya throughout the 11-year SMOS record. The similarities of these anomalous events further consolidate the idea of many polynya-favourable events taking place in the region with each having their own effect on the ice cover (e.g., Martinson et al., 1981; Holland, 2001; de Steur et al., 2007; Cheon and Gordon, 2019; Francis et al., 2019; Campbell et al., 2019; Wilson et al., 2019; Heuzé et al., 2021).

## 5   Conclusions

From the SIC data product it is known that major Weddell Sea polynya events occurred in August 2016 (3-9 Aug) and September 2017 (13 Sep until melt of that year), respectively. From the SMOS-SMAP ASIT record we now know that the episodes of anomalous wintertime sea ice loss span a wider time span than previously assumed. With the sea ice anomaly of 2018 (5-30 Sep) as well as thinning events in 2010, 2013 and 2014 that can be identified in Fig. 2, we can assume that anomalous behaviour of sea ice above Maud Rise is more pronounced than previously suggested by SIC data and is indicative of a more regular pattern of thin sea ice in the region.

By analysing the three different data products (ASIT, SIC and ERA5 meteorological reanalysis) and comparing them with one another, we tested the two hypotheses proposed in this manuscript: whether atmospheric forcing influences the sea ice region above Maud Rise and more importantly, whether ASIT retrieval is a viable candidate for the study of the Weddell Sea polynya. As previously reported on (e.g., McPhee et al., 1996; Francis et al., 2019; Campbell et al., 2019; Wilson et al., 2019; Heuzé et al., 2021) we corroborate that direct atmospheric forcing is very much involved in wide-scale drops in ASIT and SIC above Maud Rise, in addition to oceanographic forcing.

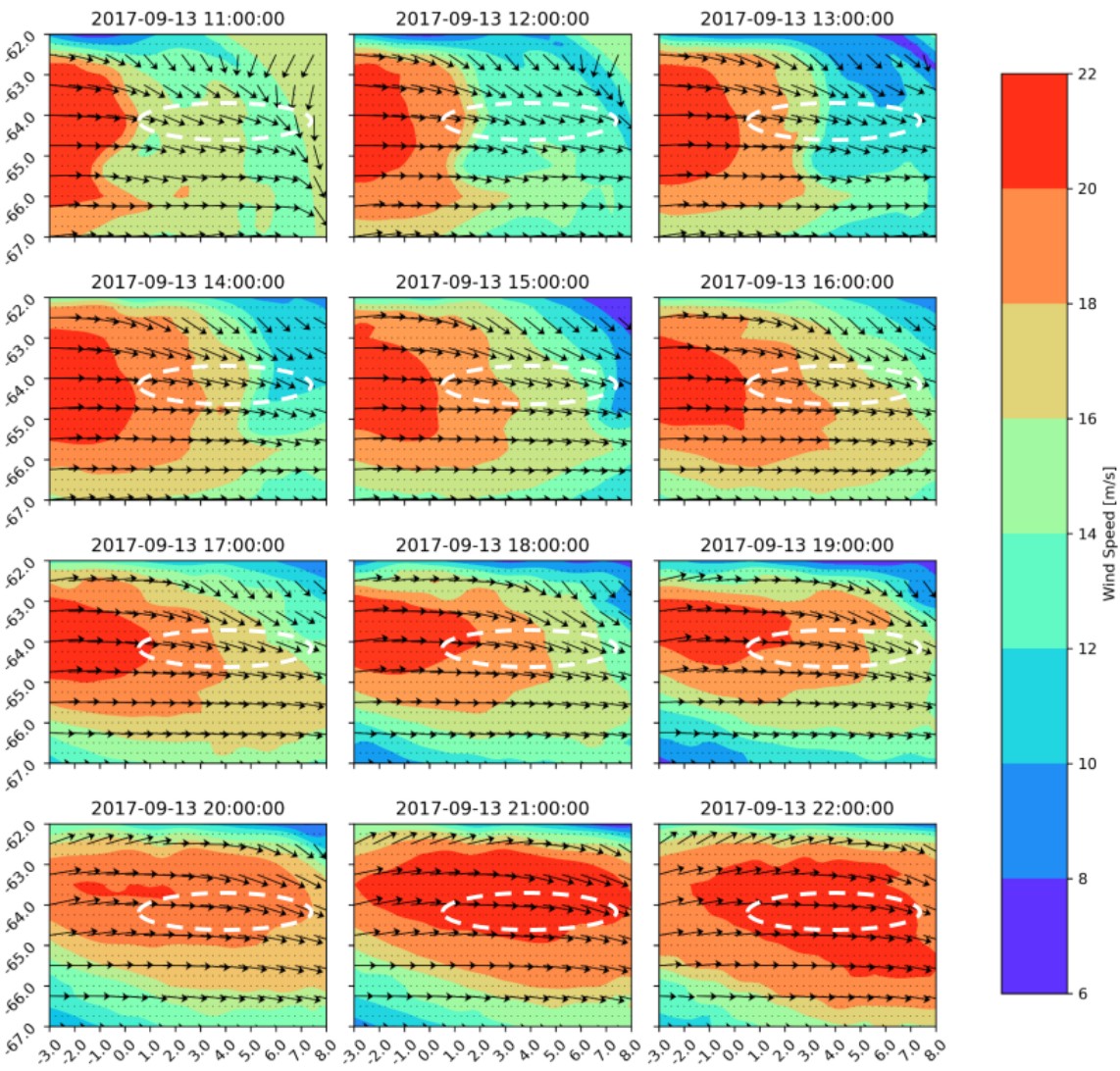

**Figure 8.** ERA5 quiver and contour plots of wind activity above Maud Rise for 9 hours on 13 September 2017, the day the 2017 Weddell Sea polynya rapidly expands. All times are reported as UTC. The polynya extent from 13 September 2017 is shown as a white dashed reference oval. All plots cover the area of interest outlined in Fig. 1.

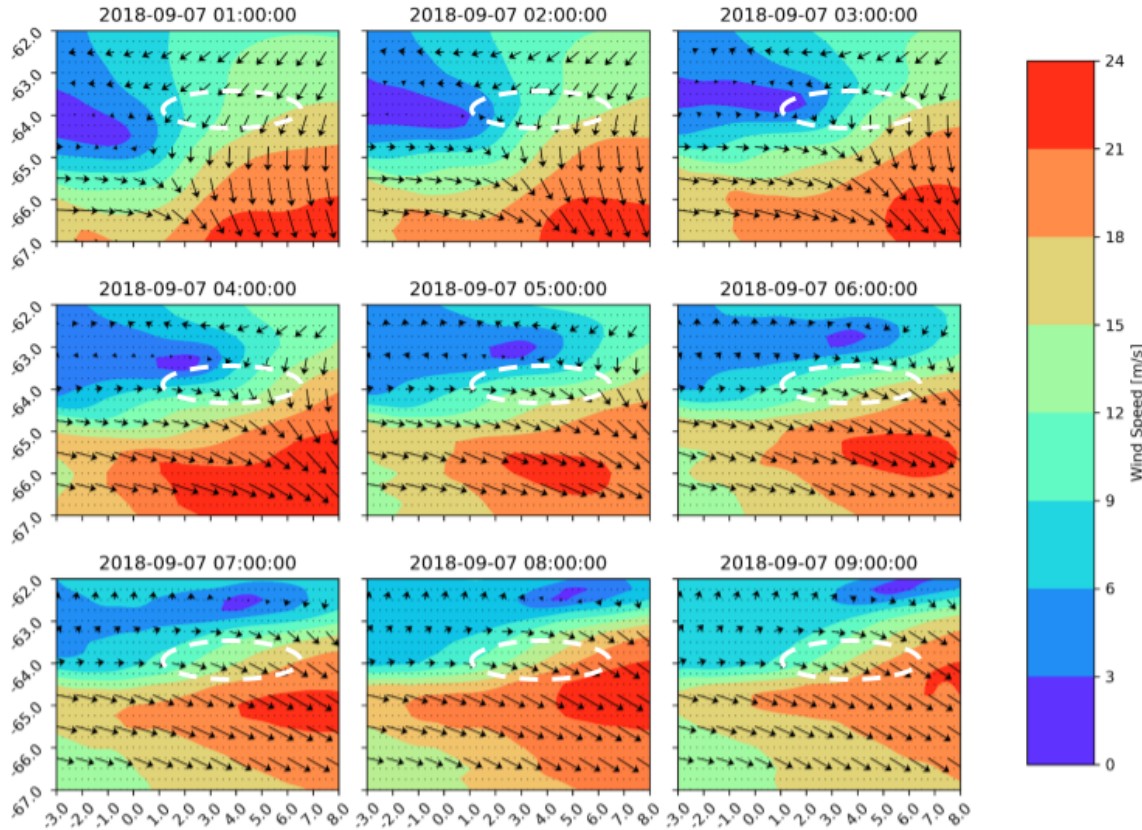

**Figure 9.** ERA5 quiver and contour plots of wind activity above Maud Rise for 9 hours on 7 September 2018, the day the 2018 sea ice anomaly starts to form. All times are reported as UTC. The sea ice anomaly extent (ASIT$< 30$ cm) 7 September 2018 is shown as a white dashed reference oval. All plots cover the area of interest outlined in Fig. 1.

The notion that strong winds are responsible for the Weddell Sea polynya is not a new one (Martinson et al., 1981; McPhee et al., 1996), but it is still generally classified as an open-ocean polynya (Morales Maqueda et al., 2004). Explanations that tie together the atmospheric and oceanic processes that cause the Weddell Sea polynya generally do not include direct atmospheric forcing, rather indirect large-scale atmospheric involvement is mentioned in the form of the negative wind stress curl intensifying the Weddell Gyre (e.g., Cheon et al., 2014b, a, 2018; Cheon and Gordon, 2019; Campbell et al., 2019). In addition to the 2017 polynya (Fig. 5), we observe wind activity influencing the sea ice anomaly of 2018 (Fig. 6). Without in-situ oceanographic data, it is difficult to access what caused the anomaly of 2018, however the possibility of wind-driven preconditioning is not unlikely based on the analysed reanalysis data as well susceptibility of the Maud Rise region to wind-induced

perturbations (McPhee et al., 1996). Wind-induced ice-ocean shear can deepen the mixed layer via turbulent mixing resulting in the entrainment of warm saline water from below, which as shown in Campbell et al. (2019) can lead to a polynya. However, it is also known that impact of strong winds is heavily determined by the regional stratification of the ocean (Wilson et al., 2019). Thus we can speculate that there was wind stress being applied to the ice above Maud Rise in 2018, which lead to ice melting but was insufficient in deepening and thereby warming the mixed layer to the point where large areas of ice could have completely melted.

Important to note is that for this study also other parameters were calculated from the base ERA5 data products like atmospheric divergence and curl (not shown), which have also been used to identify direct atmospheric forcing (Heuzé et al., 2021). In the end, strength of the wind magnitude present above the region is most directly connected with drops in ASIT and SIC. We observe the wind magnitude to be more influential than wind direction, which although sporadic, is generally towards the East as the area is dominated by westerlies. This is likely due to the scale of the area analyzed, as upon investigating the relation between wind and polynya formation on larger scales as done in Francis et al. (2019), diverging winds facilitated by cyclones have been shown to aid polynya formation. Also worth mentioning is the work done by Francis et al. (2020) that demonstrate the impact of moisture-carrying atmospheric rivers during polynya years which in addition to increasing snow fall, which effectively decouples the sea ice from the cold atmosphere once precipitated, brings clouds that trap the outgoing long-wave radiation locally resulting in further ice melting. As such, atmospheric rivers are yet another process that aid in the formation of the Weddell Sea polynya. With so many processes driving the formation of the polynya it is thus no surprise to see more regularity in sea ice anomalies in the region, as the 11-year ASIT time series has shown (Fig. 2). However, often melting of ice produces a strong negative feedback that suppresses further entrainment of deep ocean heat thereby halting convection and polynya formation, hence the rarity of this occurrence. Thus in most of the years the forcing was not strong enough to open the polynya and only the ASIT record shows the imprint of the sea ice anomaly. With strong storm activity, the previously mentioned negative feedback can be overridden by entraining enough Warm Deep Water such that the ice is fully melted (Francis et al., 2019; Campbell et al., 2019; Wilson et al., 2019). Moreover, it is the combination of these polynya-favourable forcings that cause the Weddell Sea polynya but each have their own effect on the sea ice cover. We cannot fully quantify these effects with the data presented in this manuscript. Given the full ASIT record (Fig. 2), we may speculate that other than the mean-state factors, other processes that lead to the polynya like deep convection (Martinson et al., 1981), wind-driven turbulent mixing (Wilson et al., 2019; Campbell et al., 2019) and sea ice divergence (Francis et al., 2019) as well as the influence of SAM (Cheon and Gordon, 2019) tend not to occur simultaneously. Otherwise we would expect more fully open polynya events, instead we see in our 11-year record that the thin sea ice anomaly is a more frequent occurrence indicative of some but not all of these processes taking place separately.

As for the effectiveness of ASIT analysis, we have demonstrated that it offers information that is unique as compared to standard SIC-based analysis of the region. While influenced by SIC, the SMOS-SMAP ASIT retrieval has demonstrated itself as an additional source of information that provides reasonable data about the ice conditions above Maud Rise. Most impressive are periods of near-100% SIC and low ASIT, e.g., during periods leading up to polynya. For example when the polynya opens, the large heat loss from the ocean often causes thin sea ice to grow, which soon shows up as 100% SIC but is correctly shown as

large-scale thin ice area in the SMOS-SMAP dataset. Based on our limited sample size of two polynya (2016 & 2017) within the 11-year SMOS-SMAP ASIT record (Figs. 5 and A1), austral winters that featured the Weddell Sea polynya are easily distinguishable from those that did not. Looking at Fig. 2, while we identify sea ice anomalies in years other than in 2016 and 2017, the two polynya years seem to feature large areas of anomalously thin ice prior to the occurrence of both the Weddell Sea polynya of 2016 and 2017. 2017 in particular features prolonged wide-scaled thinning that is corroborated by the weak ocean stability indicative of cycles of melt and refreeze presented in Campbell et al. (2019). Here we identify the potential of using the retrieval to predict Weddell Sea polynya but acknowledge the fact that more research needs to be conducted in this direction to validate this hypothesis and cannot comment on the robustness of this method in relation to other early detection criteria (e.g., Heuzé et al., 2021). When the polynya is open, the ASIT signal from the retrieval is unlikely to provide accurate ice thickness data due to large areas of open water influencing the signal. As mentioned before, low SIC affects the ASIT record. Thus we would like stress once again that due to the potential uncertainties in this study the ASIT record serves mainly as an indicator of anomalous sea ice activity rather than a means by which to quantify the exact degree of thinning or calculate ice volume change in the region.

In 2018, a polynya-free year, ASIT retrieval has shown that the beginning and end of a sea ice anomaly that, at its peak (18 Sep: $< 50\,\mathrm{cm}$ sea ice region with an area of $300 \cdot 10^3$ km$^2$), reached an estimated area larger than the United Kingdom. It is apparent that the SIT anomaly covered a much wider area than the area where low SIC (most likely minor lead openings) is recorded. This type of analysis, able to detect anomalous activity above Maud Rise, paves the way for a better understanding of the underlying processes that not only drive the polynya but are in fact affecting the sea ice more often than previously thought possible. The ASIT retrieval would benefit most if evaluated with direct atmospheric and oceanographic measurements, and while ERA5 atmospheric reanalysis data partly accommodates for the atmospheric component, comparisons with in-situ oceanographic measurements or model-generated best fits, like the Southern Ocean State Estimate (Mazloff et al., 2009), to better understand coincident ocean properties is highly encouraged and needed. An extension of the 11-year SMOS time series is needed to better quantify the regularity and how often such polynya-type ice anomaly events occur. As both SMOS and SMAP are science missions with no planned follow ups there is a chance that we will have a gap in the current L-band radiometry capability in space. However, with the future, operational Copernicus CIMR mission (planned launch 2028; https://cimr.eu/) some continuation of the ASIT time series will be possible.

In conclusion, through comparisons between ASIT and ERA5 data we corroborate the idea that the Weddell Sea polynya is not purely ocean-driven and instead also facilitated by direct atmospheric forcing. As for ASIT retrieval from L-band microwave radiometers like SMOS ad SMAP: it is an effective tool at monitoring sea ice conditions above Maud Rise and capable of collecting more substantial information than its SIC counterpart. Rather than substitute SIC retrieval though, the two should be used in conjunction with one another to aid the scientific understanding of the processes taking place and it should be added as yet another tool at trying to understand the unique and complex processes present in the Maud Rise region.

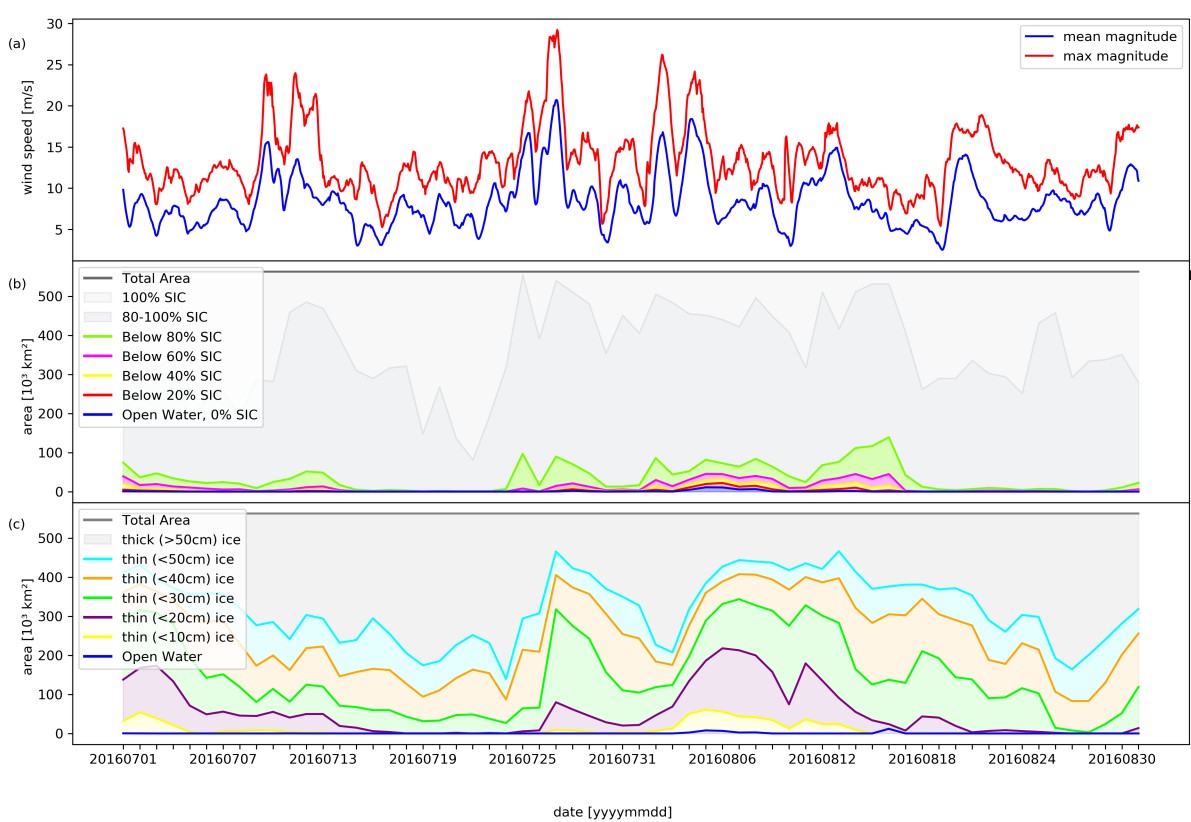

**Figure A1.** Same format as Fig. 5 for July–August 2016. See Fig. 5 caption for specifications.

*Data availability.* The SMOS–SMAP SIT and ASI SIC daily data are available at https://seaice.uni-bremen.de/databrowser/. In addition, we
acknowledge the provision of SMOS data by ESA (https://earth.esa.int/eogateway/missions/smos), SMAP data (https://smap.jpl.nasa.gov),
MODIS imagery (Worldview application, https://worldview.earthdata.nasa.gov/) by NASA, AMSR-E/2 data by JAXA (https://suzaku.eorc.
jaxa.jp/GCOM_W/) and ERA5: Fifth generation of ECMWF atmospheric reanalyses of the global climate (https://doi.org/10.24381/cds.
bd0915c6) by Copernicus Climate Change Service.

# Appendix A

## A1 The 2016 Polynya Event

In Fig. A1 we show the 2016 polynya time series in the same format as the 2017 polynya and 2018 ice thinning anomaly.

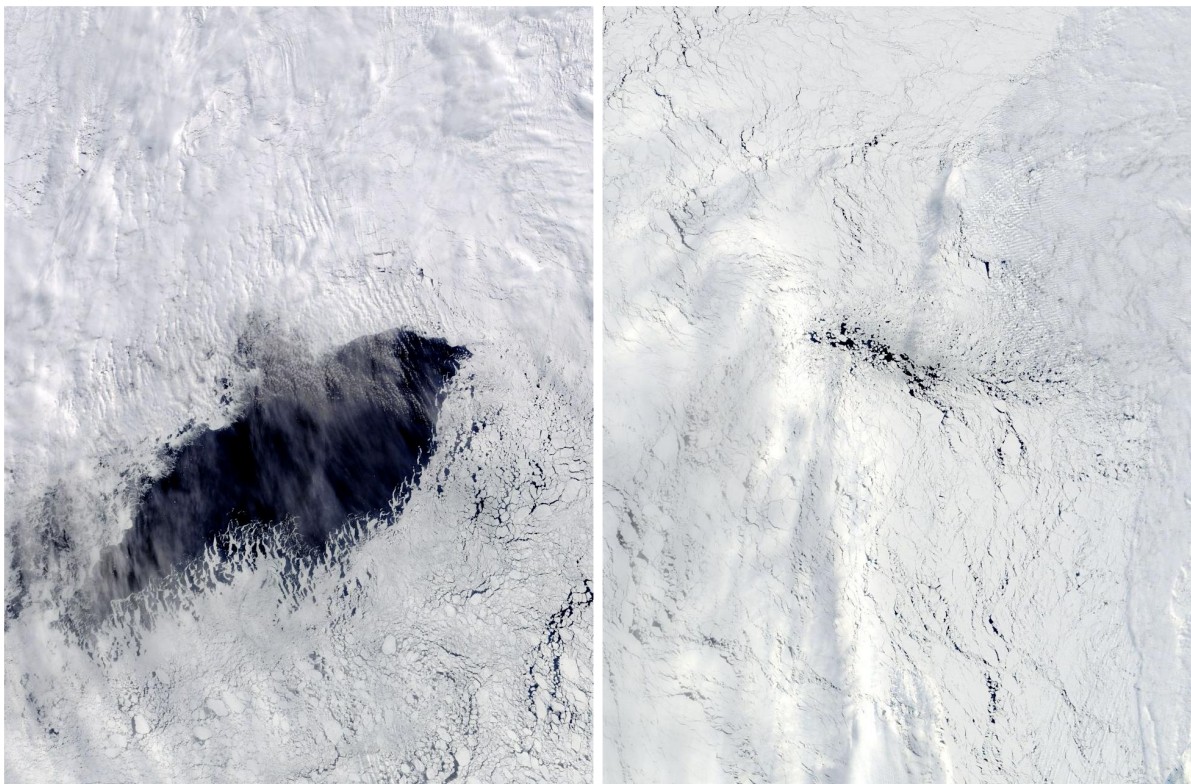

**Figure A2.** Left: MODIS image of the Weddell Sea polynya (25 September 2017). Right: MODIS image of the sea ice anomaly of 2018 (8 September 2018). Area viewed in both images is the same and chosen by assigning the bottom left corner and top right corner to chosen coordinates upon selection: 67°S, 1°E and 61°S, 8°W, respectively (images from NASA Worldview, https://worldview.earthdata.nasa.gov/).

## A2 MODIS comparison

In Fig. A2 we show the Weddell Sea polynya of 2017 and the sea ice thinning anomaly of 2018 as seen by the MODIS instrument onboard the TERRA satellite (processed and made available through NASA Worldview, https://worldview.earthdata. nasa.gov/).

*Author contributions.* Alexander Mchedlishvili wrote the paper and all co-authors contributed to the discussion and interpretation of the results. Gunnar Spreen and Christian Melsheimer provided the general structure of the paper and supervised the research that went into the project. Marcus Huntemann contributed to the methods, writing and editing of the paper.

*Competing interests.* The authors declare that they have no conflict of interest.

*Acknowledgements.* We gratefully acknowledge the work done by Cătălin Paţilea in developing the SMOS-SMAP SIT retrieval used heavily in this research. This work was support by the Deutsche Forschungsgemeinschaft (DFG) in the framework of the priority programme 'Antarctic research with comparative investigations in Arctic ice areas' SPP 1158 through grant SITAnt (project 365778379; CM, GS), the Transregional Collaborative Research Center TRR 172 "Arctic Amplification: Climate Relevant Atmospheric and Surface Processes, and Feedback Mechanisms (AC)3." (project 268020496; AM, GS), and the MOSAiCmicrowaveRS project (420499875; MH, GS). We thank the three reviewers for their very helpful comments, especially for the oceanographic aspects, which certainly helped to improve the manuscript.

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
