# Peer review of "Weddell Sea Polynya Analysis Using SMOS-SMAP Apparent Sea Ice Thickness Retrieval"

_The Cryosphere, 2021_

## Referee Comment (RC1)

**Summary:**
This manuscript presents an analysis of winter sea ice thickness in the Weddell Sea using retrievals from the space-borne SMOS and SMAP passive microwave sensors. The study period spans 2010-2020 and focuses on sea ice variability in the vicinity of the Maud Rise seamount, located in the eastern Weddell Sea. A key finding is that this region features substantial sub-seasonal variability in sea ice thickness that is poorly captured by traditional sea ice area/concentration products. The authors also showed that in 2018, there was a period of large-scale sea ice thinning near Maud Rise that was reminiscent of the lead-up to the major polynya event in 2017. For both 2017 and 2018, they show that the initiation of strong basal melting coincided with strong surface wind events.

**General thoughts:**
This paper presents groundbreaking results that reveal the nuances of sea ice thickness variability in the Weddell Sea. The Southern Ocean community has been eagerly awaiting reliable estimates of Antarctic sea ice thickness and the dataset featured here is a very promising step towards that goal. This work also provides closure to the 2016 and 2017 Maud Rise polynya events, which seemingly came to an abrupt end in 2018. Many oceanographers (myself included) expected the polynya to return in 2018 but were taken by surprise when the polynya did not reappear despite favorable upper ocean conditions. The authors showed that these events did not come to an abrupt end and that the sea ice field was in fact on the cusp of producing another polynya in 2018. Overall, it was quite satisfying to see these results as they confirm my long-held suspicion that the sea ice field over Maud Rise undergoes cycles of substantial thinning and re-growth in the winter that are not captured by estimates of sea ice area.

Though the scientific significance of this manuscript is high, the paper has several shortcomings that I believe need to be addressed before publication can be recommended. Below, I try to summarize my two major concerns:

i) Acknowledgment of relevant past studies is inadequate: One of the main conclusions from this work is that winter storms in the Weddell Sea can drive substantial basal melting, which may or may not manifest itself as a polynya (i.e. a complete melting of the sea ice cover). While the authors do present fresh evidence in support of this hypothesis, they fail to acknowledge the long history of this idea in the literature. A well-known example is the field report by McPhee et al. 1996, which documents the ANZFLUX field campaign near Maud Rise during the austral winter of 1994. In this report, they describe how the passage of a series of storms led to so much basal ice melting that they abandon their field site on the ice. A more detailed assessment of the results from that research cruise is provided by McPhee et al. (1999). Additionally, a few modeling and theoretical studies have highlighted the importance of wind-driven mixing in generating basal melting in the Weddell sea (e.g. Goosse and Fichefet 2000 and Wilson et al. 2019). With regard to the 2017 polynya event, Francis et al. (2019) and Campbell et al. (2019) both provide detailed accounts of the polar cyclones and atmospheric conditions leading up to the polynya event. While the authors credited Campbell et al. (2019) for highlighting the impact of storms, they did not cite the equally

relevant study by Francis et al. (2019). I will restate that this work does provide new and valuable validation of the notion that storms are an important ingredient for the generation of open-ocean polynyas. I also agree with the authors that importance of storms in the generation of polynyas are not acknowledged to the same extent as ocean preconditioning and large scale Ekman pumping. That said, many studies have argued in favor of these storms (i.e. McPhee et al. 1996, McPhee 1999, Goosse and Fichefet 2000, Wilson et al. 2019, and Francis et al. 2019) and they deserve to be acknowledged here.

ii) Description of SMOS-SMAP sea ice thickness retrieval is a bit lacking: While I appreciate the technical discussion provided in the data and methods section, I am still left with an unclear understanding of the sea ice thickness (SIT) retrieval process, its limitations, and the measurement uncertainties associated with the data product. Since similar SIT data have been available in the Arctic for several years now, the big question in my mind is why did it take so long to apply this technique to Antarctic sea ice. My understanding is that Antarctic sea ice has the added complication of snow-loading, where the weight of snow depresses the sea ice surface below sea level, leading to the formation of a snow-ice slush that has poorly defined densities. However, this issue is only briefly mentioned and immediately cast aside in the discussion of the results (line 275). I also don't fully comprehend the connection between the errors in SIT and sea ice concentration (SIC). In lines 100-111, the authors mention that the SMOS-SMAP retrieval algorithm assumes near 100% SIC and that they need to "correct" for this assumption using predefined SIT/SIC rations. For example, the authors acknowledge that 50 cm thick ice that is retrieved at 90% SIC is adjusted down to 28 cm (line 107). I am surprised by the magnitude of this adjustment and makes me question the validity of these results. What's even more puzzling is the assertion that these SIT values should not be combined with SIC to estimate ice volume. Overall, I think the methods section needs one or two additional paragraphs that give a more general (i.e. less technical) overview of the SIT retrieval process and the limitations and uncertainties associated with these estimates. I anticipate most readers will not be familiar with the details of these satellite retrievals and data processing, so this is an opportunity for the authors to educate the community on how best to use and interpret this valuable dataset.

Besides these two major concerns, I have less serious concerns regarding the presentation of data and the writing itself. After these revisions are implemented, I believe this paper will be a widely-read and significant contribution to the Southern Ocean literature.

**More detailed comments:**

- Line 1: This is minor point but there is some debate in the literature concerning what definition of a Weddell Sea Polynya (WSP). Some authors (e.g. Cheon and Gordon 2019 and Kurtakoti et al. 2020) argue that a WSP is strictly one that grows beyond Maud Rise and occupies much of the Weddell Sea, such as those that occurred in the

late 1970s. They distinguish these polynyas from the smaller Maud Rise Polynyas (MRP), which are largely confined to the vicinity of Maud Rise. By this definition, the 2017 event was not a WSP but a large MRP. While I think the distinction between MRP and WSP is rather superficial, the authors should make a deliberate decision with their naming convention. If they deem the 2017 event a WSP, I think it should follow that the 2016 event and other smaller events should be considered WSPs. Alternatively, the authors my sidestep this issue all together and just refer to these events generically as open-ocean or offshore polynyas in the Weddell Sea.

- Lines 20-21: This relates to my previous point about the naming of polynyas. I don't think it's true that "no sizable opening" in the sea ice occurred between 1970s and 2016. There were substantial polynyas openings in the mid 1990s and early 2000s (see figure 5 of Campbell et al. 2019).

- Line 30: "...convection pushes up Warm Deep Water..." I think "pushes up" is an awkward word choice. I would replace with "mixes" or "entrains".

- Lines 30-35: See my major points at the beginning of this review. This section introduces the potential for storms to initiate polynyas but only cites Campbell et al. (2019). Several other studies need to be acknowledged here (e.g. McPhee et al. 1996, McPhee 1999, Goosse and Fichefet 2000, Wilson et al. 2019, and Francis et al. 2019). Francis et al. 2019 in particular gives a fairly detailed account of atmospheric conditions leading up to the opening of the 2017 polynya.

- Lines 39-50: In this discussion about why Maud Rise is so favorable for polynya formation, I would cite Martinson and Iannuzzi 1998 and Wilson et al. 2019. Both use in situ ocean data to demonstrate that the Maud Rise region has a unique combination of weak stratification and high sub-mixed layer heat content that primes the region for strong wintertime basal melting and deep convection.

- Lines 43-44: Unclear description. Is the intent here to convey that the seamount affects ocean heat fluxes over an area that is twice the footprint of the seamount itself?

- Line 45: The core of the ACC, as defined by the polar front, is much further north than Maud Rise. This major current here is simply the southeastern limb of the Weddell Gyre.

- Line 73: Is there a simple explanation for why this 50 cm thickness limit exist? Is this limit affected by the presence of snow?

- Line 92: First sentence is unclear. Were these algorithms were derived from past studies that have focused estimating Arctic sea ice thickness?

- Line 97: I think these "subtle differences" between the two polar regions are important and should be described in more detail.

- Lines 100-110: As I mentioned earlier, these relatively large adjustments associated with lower than 100% SIC is rather surprising. I think more justification is needed here.

- Lines 112-120: Is SIC derived from the SMOS or SMAP sensors?

- Lines 147-148: It might help to label the subplots that referenced here.

- Line 152: After "satellite imagery" add "and in situ ocean data".

- Line 155: Here are elsewhere, the patches of low sea-ice thickness are referred to as "polynyas". Strictly-speaking a polynya describes ice-free conditions. Since the overlying ice did not completely melt, a polynya did not form. Another word or phrase is needed to describe these "polynya-like" events.

- Line 163: See previous comment. "polynya thin ice" is a bit contradictory.

- Lines 165-168: Campbell et al. (2019) and Francis (2019) both report these wind anomalies.

- Line 183: I would replace "more plausible scenario" with something like "a more complete perspective". The 2D versus 3D views of sea ice cover are not contradictory. It remains true that no polynya appeared over Maud Rise in 2018. So in that sense, the shift from record breaking winter polynya in 2017 to no polynya in 2018 was rather abrupt. With this new SIT data, we now see that 2018 represented a waning of polynya favorable conditions rather than an abrupt end. Furthermore, nothing in this study discredits the hypothesis there may have been local freshening of the surface layers in this region.

- Line 197: The discussion here is incomplete. While it is true that the loss of heat would destabilize the water column, the thinning of the sea ice implies melt, which strengthens the stratification. At these low temperatures, the stability of the water column is set by salinity. My suspicion is that this partial melting of the sea ice created a thin halocline that stabilized the water column and suppressed further exchange with the Warm Deep Water layer (Martinson 1990 and Wilson et al. 2019 describe this negative feedback in more detail). The suppression of vertical mixing effectively protected the sea ice from further basal melting, which caused no polynya to occur. I would further speculate that if a stronger storm had passed through the region, we would have seen another polynya reappearance in 2018.

- Line 198: It's not clear to me how the destablization isolates the water mass above Maud Rise. Please elaborate.

- Line 199: Up until this point, the discussion has been entirely focused on the area above Maud Rise. We have no information about the stability of the water column across the entire Antarctic sea ice zone.

- Line 203: This presence of this convection cell is speculative. This heat may have been brought up by strong wind-driven vertical mixing or Ekman divergence.

- Lines 210-212: This sentence is hard to follow. I would rephrase for clarity.

- Line 214: Campbell et al. 2019 and Francis et al. (2019) discuss wind-driven ice divergence.

- Lines 220-221: I don't think by comparing wind conditions in 2017 and 2018, one can make a general conclusion about when winds can or cannot generate a polynya. This evidence is rather anecdotal.

- Lines 237-238: Please cite the relevant studies here.

- Line 244: I would replace "anticipate" with "suggested by SIC data"

- Line 251-252: As I mentioned before, numerous studies have "rigorously" explored the idea that wind-driven mixing may drive substantial melting.

- Line 265-266: I very much agree with this statement. At some point, perhaps not at this exact line, the authors should stress that the melting of sea ice produces a strong negative feedback that suppresses further entrainment of deep ocean heat. This effect discourages deep convection, which is required to sustain a polynya. A sequence of sufficiently strong storms may override that feedback by entraining enough WDW that completely melts the ice cover and eliminates the pycnocline.

- Lines 275-278: These uncertainties should be discussed in more detail in the methods section.

- Line 290: I don't understand what is meant by "purely-open ocean polynya".

**Figures:**

Figure 1: These plots are a bit counter-intuitive. At first glance, it would appear that upticks in thin sea ice represents new growth from open-ocean conditions, when in fact they represent thinning from thicker sea ice. I would suggest filling in the space between each line with a given color as well as the top region above the green line, which represents "thick ice". Also, why is 30cm instead 50 cm used as the threshold for thin/thick ice? One might also consider normalizing the area such that the lines represent the fractional area covered by sea ice below a certain thickness.

Figure 2: A few things here. I would make the zoomed out map of the Antarctic sea ice field a standalone figure and have that be the first figure of the paper. This would help orient the reader before they examine the line plots shown in the current Figure 1. By

removing that zoomed out map, the remaining figure can then be arranged in the same format as Figure 3 to facilitate better comparison. Lastly, all these plots should have higher resolution.

Figure 4: As in Figure 2,  I think plots (b) and (c) would be better visualized if the area above and below these lines were filled with color. The time axis is also difficult read. I would suggest labelling fewer tick marks (e.g. every 5 days) and making the labels horizontal.

Figure 5: See above.

**Typos and other miscellaneous formatting issues:**

- Line 65: "for" should be capitalized.

- Line 67: "sea ice thickness of ice" is a redundant phrase.

- Line 70: SMOS and SMAP are already defined

- Line 71: "which allows to provide" I think this is a typo.

- Line 81: "...SMOS-SMAP thin ice sea ice thickness..." Redundant. Delete first "ice".

- Line 87: typo. "Because of to the 12 hour..."

- Line 143: "the" is repeated.

- Line 203: "the" is repeated.

- In the references, many of the DOI links begin with "https://doi.org//https://doi.org".

**References:**

Francis et al. (2019): Polar Cyclones at the Origin of the Reoccurrence of the Maud Rise Polynya in Austral Winter 2017. DOI: https://doi.org/10.1029/2019JD030618

Goosse and Fichefet (2000): Open-ocean convection and polynya formation in a large-scale ice–ocean model. DOI: https://doi.org/10.1034/j.1600-0870.2001.01061.x

Kurtakoti et al. (2020): On the Generation of Weddell Sea Polynyas in a High-Resolution Earth System Model. DOI: https://doi.org/10.1175/JCLI-D-20-0229.1

McPhee et al. (1996): The Antarctic Zone Flux Experiment. DOI: https://doi.org/

10.1175/1520-0477(1996)077<1221:TAZFE>2.0.CO;2

McPhee et al. (1999): Ocean Heat Flux in the Central Weddell Sea during Winter. DOI: https://doi.org/10.1175/1520-0485(1999)029<1166:OHFITC>2.0.CO;2

Wilson et al. (2019): Winter Upper-Ocean Stability and Ice–Ocean Feedbacks in the Sea Ice–Covered Southern Ocean. DOI: https://doi.org/10.1175/JPO-D-18-0184.1

---

## Referee Comment (RC3)

**Review of "Weddell Sea Polynya analysis using SMOS-SMAP sea ice thickness retrieval"**

Submitted to *The Cryosphere* by Alexander Mchedlishvili et al.

**General thoughts / major scientific comments:**

**Title, Page 1:** I would suggest considering a more descriptive title that reflects your paper's broader scope beyond the 2017 polynya. As an example, something like: "Sea ice thinning at Maud Rise identified in SMOS-SMAP record". In any case, please standardize the capitalization of the title that you choose, e.g., for the existing title: "Weddell Sea Polynya analysis using SMOS-SMAP sea ice thickness retrieval".

**Introduction, Pages 1-3:** I am concerned that the Introduction section's summary of Weddell polynya formation begins and ends with papers published in 2019 and 2020. As currently written, it neglects the four-decade-long body of literature on the Weddell polynya phenomenon, particularly on the polynya's relationship with preconditioning, stratification, convection, and eddies at Maud Rise. I recommend, at a minimum, consulting Gordon (1978), Martinson et al. (1981), Motoi et al. (1987), Comiso and Gordon (1987), Gordon and Huber (1990), Martinson (1990), Holland (2001), and de Steur et al. (2007) – none of which are cited in this manuscript – and expanding the Introduction by briefly summarizing at least a few of these foundational papers. Please also take some time to think about how these previous studies relate to your findings and cite some of them throughout your Discussion and Conclusions sections where appropriate. I make this request as I feel it is critical to reference and build on past work, especially when new science challenges long-held paradigms, as yours does. Furthermore, in addition to being incomplete, I found the Introduction difficult to follow, as it skips around between unrelated points. It would help to organize it by common themes, rather than just summarizing one paper after another. I recommend reorganizing the entire Introduction. A logical order to address topics (~1 paragraph each) might be:

(1) Past observed Weddell polynyas

(2) How Weddell polynyas are formed by destratification and maintained by convection

(3) Mean-state factors that favor/precondition polynyas at Maud Rise (low stratification, high heat fluxes, Taylor column, eddies, etc.)

(4) Role of interannual variability (SAM) in preconditioning polynyas, including in 2016-2017

(5) Role of subseasonal synoptic variability in triggering polynyas (storms, possibly eddies)

**Introduction, Page 2, Lines 28-29:** Cheon and Gordon (2019) are not the first to attribute Weddell polynyas to weak stratification, but one would not be aware from reading this passage. This explanation goes back to Martinson et al. (1981) and the other papers I've mentioned above.

**Introduction, Page 2, Lines 30-38:** Please follow the suggestions of Reviewer #1 regarding this passage. Others besides Cheon and Gordon (2019) and Campbell et al. (2019) have made the case for storms being important – or even critical – for polynya formation near Maud Rise, and those studies should be acknowledged. Additionally, the phrasing "admit to" feels overly disparaging towards Cheon and Gordon (2019), who discuss "atmospheric influences" in considerable detail. Please change this phrasing; you could be more specific and say that their study discusses large-scale, climate-related atmospheric effects, but not synoptic scale meteorological variability.

**Introduction, Page 2, Lines 37-38:** This sentence is almost entirely copied verbatim from two sentences on p. 320 of Campbell et al. (2019), to the extent that this could be considered plagiarism. You must rephrase this. You've also combined two unrelated ideas (divergence of ice preventing stabilization by ice melt, and turbulent mixing leading to heat/salt entrainment) from their study and so the sentence is not coherent as written.

**Introduction, Page 2, Line 39:** Saying "thus far all discussed preconditioning is... [not] exclusive to... the region of interest" is not accurate: (1) Preconditioning mechanisms have barely been discussed in the preceding paragraphs. (2) All of the preconditioning mechanisms are, in fact, exclusive to the region of interest. The reason that positive SAM fluctuations and an enhanced Weddell gyre circulation result in preconditioning is that they increase the doming of isopycnals in the center of the Weddell gyre, *where Maud Rise is located*, thus uplifting warm and salty Weddell Deep Water.

**Data/Methods, Page 4, Lines 92-99 (also Lines 275-277):** The lack of published calibration and validation of the SMOS and SMOS-SMAP thin ice thickness retrievals in the Antarctic is hugely concerning to me. Your paper's analysis and conclusions rest on a highly uncertain empirical satellite retrieval for which no published validation has been conducted for sea ice in the Southern Hemisphere. I request that several actions be taken to mitigate this issue:

1. Please be explicit in Line 92 that the retrieval is empirical and that it was developed through comparison with a simple Cumulative Freezing Degree Days (CFDD) model of Arctic sea ice growth, then calibrated and validated using other Arctic sea ice estimates (Huntemann et al. 2014).

2. Make clear throughout Section 2.1 that the lack of Antarctic validation is a limitation and discuss the reasons why this is the case. For example, discuss the sensitivity of the SMOS/SMOS-SMAP retrieval to overlying snow cover, sea ice salinity, and other ice/snow parameters, and mention the degree to which these parameters differ between the Arctic and Antarctic. Additionally, the retrieval was trained on a CFDD model that applies well to the Arctic but probably not the Antarctic, where ocean heat fluxes are stronger and significantly influence sea ice growth in weakly-stratified regions such as near Maud Rise (see, e.g., Wilson et al. 2019). Please discuss the potential impact of different rates of ice growth in these regions.

3. Be explicit that the "minor evaluation tests" mentioned in the Antarctic (Lines 92-96) are unpublished. Mention what region the EM-bird validation tests occurred, over what time period(s), and over what ranges of sea ice thickness. Give more detail about the degree of agreement between the EM-bird and SMOS-SMAP data at different sea ice thicknesses within 0-50 cm. Please consider including a figure showing this validation exercise. Importantly, this will help provide a reference for the community as others publish studies using the SMOS-SMAP data for Antarctic research, which is inevitable given that you have publicly released the Antarctic data and it is increasingly being used.

4. Add a straightforward figure to this paper, perhaps in the Appendix, showing the uncertainty of the SMOS-SMAP retrieval at sea ice thicknesses spanning 0-50 cm, perhaps modeled after Figure 9a in Paţilea et al. (2019). However, note that their Figure 9a (which already shows substantial error of ~30 cm for a thickness of 50 cm) is based on a SIC uncertainty of 5%. Given the 'halo' of low SIC (85-90%) known to be present around Maud Rise (Lindsay et al. 2004), the deviation of SIC from 100% is much larger than 5% in this region. Please account for this larger SIC uncertainty when computing SMOS-SMAP uncertainties for this figure. This will help to address the concerns of both Reviewer #1 and myself.

5. Clarify in Lines 105-107 whether the SMOS-SMAP SIT data you show in this study are biased high or low at SICs less than 100%. The language you use (e.g., "retrieved sea ice thickness," "50 cm... is just 28 cm") is confusing.

6. Please do not frame the three recent papers that have used SMOS SIT estimates to answer scientific questions in the Antarctic as mitigating the uncertainties in this retrieval (as you imply in Lines 97-99). SMOS SIT data were not the focus of any of those papers, which all referenced a wider variety of data sources and thus were less dependent on the accuracy of the SMOS retrieval.

**Data/Methods, Page 5, Lines 131-133:** This sentence is copied nearly verbatim from Campbell et al. (2019). This is not permissible and could be perceived as plagiarism, like the other instance I mention above. I understand the wish to provide these specific details, but the sentence must be at least rephrased.

**Results, Page 8, Lines 165-167:** This was a major finding of both Francis et al. (2019) [see their Fig. 5] and Campbell et al. (2019) [see their Extended Data Fig. 4], as Reviewer #1 mentions. Those papers should be cited.

**Discussion, Pages 10-11, Lines 195-205:** As Reviewer #1 points out, these few sentences about the role of the ocean contain incoherent reasoning and baseless speculation. My best advice would be to read and digest more of the oceanographic literature on Weddell polynyas – which, as I mention in previous comments, has almost entirely been neglected in the references of this manuscript – and then to completely reconsider the relevance of ocean processes to the data presented in this study. I will emphasize two key ideas to consider, though I would encourage the authors to build on these by assembling their own reasoning and references:

1. The sea ice thinning observed to precede the 2017 polynya is not necessarily (and, in my judgment, is almost certainly not) associated with ocean "destabilization" or deep convection. Note that sea ice growth in the Antarctic is (approximately) governed by the balance of ocean heat input and atmospheric heat extraction. If ocean heat fluxes become greater, sea ice will cease to grow and may start to melt. A variety of factors can influence ocean heat fluxes:

   - Turbulent mixing due to ice-ocean shear (which itself is greater at high wind speeds) can deepen the mixed layer and entrain warm pycnocline waters.

   - Brine rejection from ice growth densifies and deepens the mixed layer, also resulting in entrainment of warm pycnocline waters. In contrast, freshwater input from ice melt can rapidly shoal the mixed layer, limiting heat fluxes from warm waters below.

   - Upwelling due to large-scale cyclonic winds (Ekman upwelling) or smaller-scale eddies (which may produce "doming" of isopycnals) can uplift warm pycnocline waters into the surface mixed layer.

   Without a basic understanding of these three processes, I believe it is not responsible to speculate on the causes of the thin sea ice anomalies that you observe. To better understand these processes, I would recommend carefully reading the very relevant study by Wilson et al. (2019). Note that atmospheric variability may also play a role in modifying the energy balance of sea ice; for example, see the study on the 2017 polynya published in *Science Advances* by Francis et al. (2020).

2. To explain the time evolution and interannual variability of sea ice thickness at Maud Rise, it is important to consider the role of ocean stratification and storms. For example, Campbell et al. (2019) show reduced upper-ocean stratification at Maud Rise in 2016 and 2017 due to a saltier surface layer. Lower stratification means lower resistance to turbulent mixing-driven or brine rejection-driven entrainment of warm pycnocline waters. This could easily explain the thin sea ice anomalies observed preceding the 2017 polynya (more entrainment = larger heat fluxes = thinner ice). However, I hope that you can go further and think about why thin sea ice anomalies are also seen in other years, as well as the sub-seasonal time evolution of these anomalies and their relation to storm perturbations. As McPhee et al. (1996) discovered as the sea ice melted beneath their ice camp during the ANZFLUX experiment, passing storms can strongly affect both the ocean and sea ice over Maud Rise.

**Discussion, Page 11, Lines 215-216:** No, low SIC and strong winds do not directly cause upwelling of warm water. Upwelling is a distinct process from mixing or entrainment. I believe what you are referring to is either turbulent mixing of warm pycnocline water related to high wind speeds, or entrainment of those same warm pycnocline waters related to densification of the mixed layer from cooling or brine rejection. Please see my previous comment on this topic.

**Discussion, Page 11, Lines 226-227:** Please cite Cheon and Gordon (2019) and Campbell et al. (2019) for this finding, which is not shown in your study. Change "upwelling" to "upwelling and/or mixing"; note my comments above regarding this distinction.

**Conclusions, Page 14, Lines 251-253:** As Reviewer #1 points out and as my comments throughout have indicated, previous studies have, in fact, put forth "rigorous" explanations regarding the influence of atmospheric perturbations (e.g., storms) on ice melt and polynya formation at Maud Rise. They should be cited here. Also, Cheon and Gordon (2019) is not the only study that has discussed the role of large-scale negative wind stress curl: Campbell et al. (2019) and Cheon et al. (2014, 2015, 2018) have also discussed this.

**Conclusions, Page 15, Lines 257-258:** I disagree wholeheartedly. I don't see how ocean heat loss during the 2016-2017 polynyas would preclude the possibility of anomalously large ocean heat fluxes (rather than atmospheric forcing) producing thinner sea ice in 2018. See my comments above about the factors that control ocean heat fluxes, such as upper ocean stratification. Additionally, you do not assess wind speeds in most years of your 2010-2020 record, and so you have not demonstrated that wind speeds preceding/during the 2016-2018 polynyas or ice thinning events were particularly anomalous. Intense storms pass by Maud Rise every year; you have not shown that the atmospheric perturbations in 2016-2018 were more severe than in other years.

**Specific comments / minor scientific comments:**

**Abstract, Page 1, Line 2:** I'd echo the point made by Reviewer #1 regarding the somewhat controversial nomenclature of "Weddell Sea Polynyas". As they mention, some recent literature makes a distinction between "Maud Rise polynyas" and "Weddell Sea polynyas" (e.g., Kurtakoti et al. 2018, 2021) despite providing no objective quantitative or mechanistic criteria (e.g., size or geographic thresholds, multi-year persistence, distinct formation mechanisms, etc.) to definitively sort a polynya into one category or the other. On the other hand, your use of "Weddell Sea Polynya" as a catch-all designation raises the question of how large or persistent an opening must be to merit this formal, capitalized title. The 2017 polynya reached ~50,000 km$^2$ in size. Could a much smaller opening only 500 km$^2$ (22x22 km) also be considered a "Weddell Sea Polynya"? Many might disagree. How about one that is 5,000 km$^2$? These questions are not hypothetical, as we know this phenomenon occurs over a broad spectrum of size and duration (see, e.g., Campbell et al. 2019; Heuzé et al. 2021). If you want to avoid addressing this issue, one option might be to change "Polynya" to a lowercase "polynya" throughout your manuscript and mention (perhaps in Lines 22-23) that "Weddell Sea polynya" simply refers to any sea ice opening near Maud Rise.

**Abstract, Page 1, Lines 2-3:** There are a few problems with this sentence:

- It is inaccurate to state that the WSP has been absent for 40 years. This also undermines your paper's argument that a moderate thinning in non-polynya years is notable. As Reviewer #2 notes, other smaller polynyas have appeared over Maud Rise, e.g., in 1980, 1994, 2005, 2016, and arguably in a significant fraction of other years (see Comiso and Gordon 1987; Holland 2001; Muench et al. 2001; Venegas and Drinkwater 2001; de Steur et al. 2007; Campbell et al. 2019; Heuzé et al. 2021). Please consider acknowledging this, e.g., by stating: "After 40 years of intermittent, smaller openings, a larger, longer-lasting polynya appeared..."

- It is not clear how to precisely and objectively define when the 2017 polynya first appeared or what "fully opened" means. Campbell et al. (2019), for example, trace its origin to two openings that were actually first seen on September 3 and coalesced and grew in size from September 13-18 (see their Extended Data Fig. 8). You could say: "...appeared in early September, 2017..."

- To be more specific than "melt," use "spring ice melt season" or similar. As Reviewer #2's comment indicates, a polynya is already open and so it cannot "melt."

- Consistent with my earlier comment, change "a total of 80 days" to "approximately 80 days."

- In summary, here's what I would suggest: "After 40 years of intermittent, smaller openings, a larger, longer-lasting polynya appeared in early September, 2017, and remained open for approximately 80 days until spring ice melt."

**Abstract, Page 1, Lines 8-9:** This phrasing ("we present the strong impact storm activity has on sea ice") ignores that Campbell et al. (2019) and Francis et al. (2019) have already made a strong case for storm activity impacting the evolution of the 2016 and 2017 Maud Rise polynyas using similar or identical atmospheric reanalysis data sets, as Reviewer #1 also mentions. Consider changing this to: "... we corroborate previous findings on the strong impact that storm activity can have on sea ice at Maud Rise" (note that this is also avoids implying that your results can be generalized to other sea ice-covered regions, where the impact of storms may be quite different).

**Abstract, Page 1, Lines 9-10:** First, it is unclear what you mean by "direct atmospheric forcing." Second, the grammar of the phrase set off by commas ("... in addition to oceanographic effects") is not correct. This should be changed to something like, "... help consolidate the theory that the evolution of Weddell Sea Polynyas is controlled by atmospheric as well as oceanographic variability" [or 'forcing' or 'effects', whichever is most appropriate].

**Introduction, Page 1, Line 20:** Cheon and Gordon (2019) is not the appropriate citation for the 1970s polynya. Please cite Carsey (1980).

**Introduction, Page 1, Line 20:** See my comment above on Lines 2-3 and the similar note from Reviewer #1. Polynyas have appeared in many years since the 1970s, and there is a substantial amount of literature discussing their past occurrence that is being neglected here.

**Introduction, Page 1, Line 22:** "Anything comparable" feels very arbitrary. The 2017 polynya was not much larger than the 2016 polynya, which itself was not much larger than the 1994 polynya, which was not much larger than the 2005 polynya, and so on (see Fig. 5 in Campbell et al. 2019). These events exist on a continuum of size and duration, most likely stemming from similar physical processes, and so arbitrary cutoffs make little sense. It would be more accurate to just state that the 2016 and 2017 events were the largest and longest-lived since 1976. Here, Swart et al. (2018), Cheon and Gordon (2019), Campbell et al. (2019), and Jena et al. (2019) should be cited; note that Swart et al. should be included as they were first to report on the 2017 event.

**Introduction, Page 1, Line 25:** The standard citation for classification of the WSP as an open-ocean or "sensible heat" polynya is Morales Maqueda et al. (2004), not these three papers from 2019.

**Introduction, Page 2, Line 35:** Replace "contributes" with "may contribute", as storms will not necessarily lead to ice divergence over Maud Rise (it depends on the particular storm), and both the divergence and mixing were speculative rather than shown directly by Campbell et al. (2019). However, McPhee et al. (1999), Sirevaag et al. (2010), and others have directly measured turbulent mixing over Maud Rise.

**Introduction, Page 2, Lines 49-51:** Since the 23-year time series referenced here ends in 2001, this wording is confusing. You can just say that "... the mean sea ice concentration (SIC) for the months of July through November shows... (Lindsay et al. 2004)." Please also omit the "... lacks the open water expanse indicative of a polynya" part. A literal "open water expanse" (i.e., 0% SIC) would only occur in a mean SIC field if a polynya occurred 100% of the time during these 23 years, so your wording does not make sense.

**Data/Methods, Page 3, Line 73:** Please be more explicit here that the SMOS and SMOS-SMAP retrievals cannot estimate sea ice thicknesses greater than 50 cm.

**Data/Methods, Page 4, Lines 107-108:** This is the incorrect citation. Pațilea et al. (2019) do not show this; this result was found by Heygster et al. (2014).

**Data/Methods, Page 4, Section 2.2 (Lines 113-120):** Please clarify which satellite mission's data the ASI algorithm is applied to, both here and above on Line 67. Otherwise, the reader will assume ASI is being applied to SMOS or SMAP. I am only familiar with ASI being applied to the SSM/I and AMSR-E/2 sensors, not SMOS or SMAP (Spreen et al. 2008; Beitsch et al. 2014). Provide the appropriate citations for the data.

**Data/Methods, Page 4, Line 123:** The findings of your study are not the final word on this interesting and complex question that many have tried to address, so please omit "conclusively" and consider changing "answer" to "investigate".

**Data/Methods, Page 5, Lines 135-136:** This claim about ERA5 improving on ERA-Interim requires a citation.

**Results, Page 5, Line 143:** Sea ice thinning does not constitute a polynya. A polynya is, by definition, an area of open water. Please change "the polynya is visible ... but does not open completely" to something along the lines of "sea ice thinning is observed over multiple weeks". Please also rephrase this in Lines 155 and 163 below.

**Results, Page 5, Lines 143-144:** Your Fig. 1 does not compare the SMOS-SMAP retrieval directly to SIC data, so this sentence is not justified in this section. Please delete.

**Results, Page 5, Line 147:** For reproducibility, it is important to describe and show precisely the "area of interest" that constitutes your averaging region. The coordinates that you cite do not match with the box you drew in Fig. 2 (left) or the subplots in Fig. 2 (right), which are regions irregular in latitude and longitude. Which is your actual averaging region? Please make sure the black box and subplots correspond precisely to the region you use. If it is indeed irregular in lat/lon, you could, for example, give the coordinates of the northwest and southeast corners. Also, please change negative coordinates (e.g., -3.5°E) to the proper values (e.g., 3.5°W).

**Results, Page 5, Line 153:** Could you be more descriptive in your summary of Fig. 3? For example, you could mention that it shows a broad gradient of SIT encompassing a larger area on all sides of the polynya than shown in the SIC data, which exhibits a sharper gradient of ice concentrations.

**Results, Page 8, Line 164:** This is not the correct usage of "preconditioning". With polynyas, preconditioning refers to ocean processes that reduce stratification and/or increase subsurface heat content. This can happen over weeks, months, or years. More accurate here would be to say: "Fig. 5 depicts the Weddell Sea polynya of 2017 as well as the weeks leading up to the event."

**Discussion, Page 9, Line 180:** Since it is not established in the literature that the small 1973 polynya directly produced the 1974-76 event, please say "preceded" instead of "resulted in a much larger iteration of". Also cite Martinson et al. (1981) and Comiso and Gordon (1987), who mention the 1973 polynya.

**Discussion, Page 10, Line 189:** Neither low SIC area nor thin SIT area peaked on 4-5 August of 2016, from looking at your Fig. A1. Please fix or omit this.

**Discussion, Page 10, Line 191:** What do you mean by "shows some variability"? This is quite vague. Be specific, if not quantitative. Additionally, I agree with Reviewer #2 that a correlation coefficient should be presented if you are going to state that two time series are "not very correlated". Lastly, please fix the grammar in this sentence.

**Discussion, Page 11, Lines 195-196:** Fix the grammar in this sentence, e.g., "... we see that this opening in sea ice, at first minor, eventually paved the way for the Weddell Sea polynya." Also, do you mean to refer to a small opening that preceded the polynya, or a thinning of sea ice preceding the polynya? I am guessing the latter, but if you intended the former (a small opening preceding the larger opening), please reference Campbell et al. (2019) who also demonstrate this.

**Discussion, Page 11, Lines 212-215:** Ice may be advected without creating any new open water, i.e., ice advection does not necessarily imply ice divergence. You have not shown or calculated ice divergence, and so what you are claiming (pack ice "broken apart by wind") is unfounded speculation. However, you could reference other studies that have calculated or discussed ice divergence for the 2016 and 2017 polynyas: Campbell et al. (2019) [see their Methods section "Atmospheric reanalysis" and Extended Data Fig. 5] and Francis et al. (2019) [see their p. 10-11].

**Discussion, Page 11, Lines 220-221:** Reviewers #1 and #2 point out that this statement is not justified by your analysis. I agree.

**Discussion, Page 11, Lines 221-222:** For your statement regarding 13 September 2017, please cite the prior studies, e.g., "..., corroborating the findings of Campbell et al. (2019) and Francis et al. (2019)."

**Discussion, Page 11, Lines 222-224:** Do you have a figure showing this? If not, please note in parentheses: "(not shown)".

**Discussion, Page 11, Lines 228-230:** This wording ("instead of") implies that you use SMOS before 2015 and SMOS-SMAP from 2015 onwards in Fig. 1, which contradicts your Fig. 1 caption. Please clarify your wording to make clear which is the case. Also, this is a run-on sentence. Fix the grammar, e.g., "... that fully includes the freezing periods...".

**Fig. 8 caption, Page 13:** See my comment above on the Abstract regarding the earlier sea ice opening on September 3, 2017. I would say that the polynya "rapidly expanded" on September 13, rather than opened for the first time.

**Conclusions, Pages 14-15:** In this section, I suggest that you consider discussing how your findings relate to those of Heuzé et al. (2021), whose recent analysis of past polynya events at Maud Rise is highly relevant. Could the SIT data you present offer an "early detection" system for polynyas, as their study aims to develop? In other words, do you find that SIT anomalies consistently precede sea ice openings at Maud Rise? How much lead time prior to a polynya opening could a SIT-based detection system offer?

**Conclusions, Page 14, Line 241:** Reference my earlier comments regarding the opening date of the 2017 polynya.

**Conclusions, Page 14, Line 255-256:** Campbell et al. (2019) also show this using in situ ocean observations, as well as quantifying the rate of heat loss during the 2016 polynya (see their Fig. 3 and Extended Data Fig. 7).

**Conclusions, Page 14, Line 261:** Since you do not quantitatively test correlations, please change "has the most direct correlation" to "is most directly connected" or something similar.

**Conclusions, Page 14, Lines 264-265:** This is not a good summary of the processes that Francis et al. (2020) argue contributed to formation of the 2017 polynya. Please fix. Their study focuses on how atmospheric rivers changed the energy balance of sea ice and the overlying snow cover to favor surface ice melt and thinning.

**Conclusions, Page 14, Lines 271-272:** The fact that the SMOS-SMAP retrieval is influenced by SIC means that, by definition, it is not independent of SIC. Please change to something along the lines of "an additional source of information".

**Conclusions, Page 15, Lines 282-283:** I don't see how daily SMOS-SMAP snapshots allow monitoring "on a more frequent basis" than existing daily SIC data. Delete.

**Conclusions, Page 15, Line 290:** I agree with Reviewer #1—the phrase "purely-open ocean polynya" is confusing and, in any case, neglects past work that has discussed atmospheric influences on the polynya.

**Data Availability / Acknowledgements, Pages 15 and 17:** Please move the appropriate URLs, DOIs, and information about other data sources (e.g., ERA5) to your Data Availability statement; they do not belong in the Acknowledgments.

**Appendix A2 / Fig. A2, Pages 16-17:** I'm not sure what you are aiming to illustrate with Fig. A2. It is not referenced in the text of your paper. If you want to include it, please discuss and reference it somewhere in the text; if not, Appendix A2 can be removed. Note that "snippets" is not a formal term; change to "images" or similar.

**Technical corrections / typographical comments:**

**Abstract, Page 1, Lines 3-4:** Awkward and ambiguous wording in this sentence. I'd suggest: "We find that 2017, however, is far from the only year that the imprint of a polynya can be identified at this location."

**Abstract, Page 1, Lines 4-5:** Clarify that this is a "thin sea ice thickness (SIT)" product, and make a similar change to Lines 54 and 56.

**Abstract, Page 1, Line 5:** Perhaps "estimates" would be more appropriate than just the general word "data" given the uncertainties, limitations, and lack of validation of this product in the Antarctic.

**Abstract, Page 1, Line 6:** Instead of "have isolated," either "identified", "found", or "discovered" (without "have") would be a more fitting word choice and verb tense.

**Abstract, Page 1, Line 8:** Clarify: "... in the 11-year SMOS-SMAP time series." Omit "the" in "Using the ERA5 surface wind..."

**Abstract, Page 1, Line 10:** Add a comma after "results presented".

**Abstract, Page 1, Line 11:** Instead of "a binary system," "a binary phenomenon" would be more clear. "System" doesn't make sense here. Please also change this below in Line 57.

**Abstract, Page 1, Line 12:** Avoid using "occur" twice. Perhaps the second instance could be changed to something like: "... for the polynya to appear and persist."

**Abstract, Page 1, Line 13:** "Atop" would make sense for activity happening directly on a mountain top on land. For a sea ice opening that manifests >1000 m above a seamount, avoid "atop" and use "above," "at," or "near" instead here and also below at Line 56.

**Abstract, Page 1, Line 14:** Insufficient for what? I assume you mean "insufficient to trigger a polynya" – please change to something along these lines. Also, avoid future tense in "as will be shown"—instead: "as we show" or "as we demonstrate".

**Abstract, Page 1, Line 17:** This wording is awkward. Instead: "... in comparison to the use of satellite sea ice concentration products alone."

**Introduction, Page 2, Line 34:** In the Southern Ocean, storms do not "occur locally" at a certain location – they are fast-moving and long-lived and thus, more accurately, "pass over" a location. Please replace with "... proof of how the passage of intense storms near Maud Rise aids..."

**Introduction, Page 2, Line 45:** Change "the Taylor column" to "a Taylor column" since this has not been previously mentioned in the paper.

**Introduction, Page 2, Lines 53-54:** The preprint you cite is now published as Heuzé et al. (2021). Please see the References section in this review for the new citation, and update it throughout your paper.

**Introduction, Page 2, Line 56:** Instead of "reverse the notion," how about "challenge the notion"?

**Data/Methods, Page 3, Lines 66-67:** Clarify: "... preceding the installment of SMAP in 2015".

**Data/Methods, Page 3, Lines 70-72:** Fix the grammar and extra period in this sentence ("are working", "which allows to provide", "thin sea ice SIT). (refs)").

**Data/Methods, Page 3, Lines 81-82:** Fix the grammar in the phrase "by adapting it to SMAP by modifying it...". Also, "average" should be "the average".

**Data/Methods, Page 3, Line 87:** "Radio" should be lowercase. The abbreviation (RFI) does not need to be included because you do not use it elsewhere.

**Data/Methods, Page 3, Line 88:** Delete the unnecessary comma after "temperatures".

**Data/Methods, Page 3, Line 89:** Fix grammar – do you mean "more of the brightness temperature variations"?

**Data/Methods, Page 4, Line 100:** In "further validate and distinguish SIT data," what do you mean by "distinguish"? This word choice is confusing.

**Data/Methods, Page 4, Line 101:** "Polynyas" is the proper plural form, not "polynya". Please also fix this below on Line 236.

**Data/Methods, Page 4, Line 122:** "Climate Reanalysis" is not an official title associated with ERA5, so it should not be capitalized. That said, a more accurate description is "ERA5 atmospheric reanalysis data".

**Data/Methods, Page 5, Line 125:** "Embodies" doesn't make sense. Consider "provides" or "represents".

**Data/Methods, Page 5, Lines 129-130:** Since you do not use the ERA5 ensemble uncertainty estimates, that information can be omitted.

**Results, Page 5, Lines 140-141:** More accurately – "For a detailed analysis of the Weddell Sea polynya region, ..." – since a polynya did not appear in 2018. Also, "two years" should be "two months".

**Results, Page 5, Line 142:** "Results section" (with the section name capitalized), not "result section".

**Results, Page 5, Line 143:** Change comma to a semicolon; otherwise, the grammar is not correct here.

**Results, Page 5, Lines 146-147 (and Fig. 2):** Label the subplots of Fig. 2 and then specify here that you're referring to Fig. 2a. Clarify that this is not just "a standard... retrieval" but rather an image of the sea ice thinning event of interest in September 2018. You can omit "... with a total of 664 rows and 632 columns." It is unclear whether this refers to just the sea ice-covered region or the entire plot, and, in any case, the dimensions of the raw data array is not useful information here.

**Results, Page 5, Lines 148-149:** This sentence ("Accompanying the SMOS-SMAP 2017 maps...") is referring to Fig. 3, not Fig. 2. Please note this. Also, change "segment" to "region".

**Fig. 1 caption, Page 6:** Please clarify: "As with the SMOS-SMAP SIT time series in Figs. 4-5, ...". Also, please follow the suggestions of the editor regarding adding units to the y-axis and noting that what is shown is the area of open water or thin sea ice.

**Fig. 2 caption, Page 7:** The word "spanning" is not used correctly here. Instead, you could say: "SMOS-SMAP SIT retrieval for the seasonal ice zone around Antarctica." Additionally, the word "segment" is not used correctly here. Please replace both instances with "region".

**Results, Page 7, Line 159:** Switch around the words: "No area below 40% SIC is detected..."

**Fig. 4 caption, Page 8:** Specify that wind speed is "10-m wind speed". Clarify that the "daily maximum" derives from the ERA5 grid cell with the maximum wind speed and that both "daily maximum" and "daily mean" are computed for (not just "cover") the bounding box shown in Fig. 2. Also add these in the Fig. 5 and Fig. A1 captions.

**Results, Page 9, Line 171:** Change "For the highlighted regions..." to something like, "For the time periods highlighted with colored frames...". "Region" generally refers to a geographical region, so your wording is confusing.

**Discussion, Page 9, Lines 176-177:** "... comparing the SIT retrieval..."

**Discussion, Page 10, Lines 184, 188, 195:** The phrase "we have" is too colloquial and a bit confusing. Consider replacing these occurrences with "we observe", "we identify", "we see", or equivalent phrases.

**Discussion, Page 10, Lines 187-188:** This sentence is phrased awkwardly. Please fix.

**Discussion, Page 10, Line 188:** "... in early August..."

**Discussion, Page 10, Line 189:** You do not provide objective, quantitative criteria to sort polynya events into "major" and "minor," so please change "minor" here to "smaller".

**Discussion, Page 10, Line 189:** "Fig. 5b", not just "5b".

**Discussion, Page 10, Line 194:** Change "less than 50 cm thick ice" to "sea ice thinner than 50 cm" ("less than" is confusing here).

**Discussion, Page 11, Line 208:** Change "event" to "events". Add comma after "seen in Fig. 4b".

**Fig. 7 caption, Page 12:** Specify the time zone of the timestamps (is it UTC?). Add a quiver key to the plot showing the scale of the wind vectors. Do both for Fig. 8 as well.

**Conclusions, Page 14, Line 240:** In "... known that a Weddell Sea Polynya events...", delete "a".

**Conclusions, Page 14, Line 241:** "the anomalous episode"

**Conclusions, Page 14, Lines 261-262:** The grammar of this sentence is not correct.

**Conclusions, Page 14, Line 270:** "the effectiveness"

**Conclusions, Page 15, Line 283:** "paves the way for..."

**Conclusions, Page 15, Line 285:** "11-year"

**Conclusions, Page 15, Line 291:** "Polynya conditioning" is not a standard term. Neither is "preconditioning" appropriate here. How about "polynya formation"?

**References:**

Beitsch, A., L. Kaleschke, and S. Kern, 2014: Investigating high-resolution AMSR2 sea ice concentrations during the February 2013 fracture event in the Beaufort Sea. *Remote Sens.*, **6**, 3841–3856, doi:10.3390/rs6053841.

Campbell, E. C., E. A. Wilson, G. W. K. Moore, S. C. Riser, C. E. Brayton, M. R. Mazloff, and L. D. Talley, 2019: Antarctic offshore polynyas linked to Southern Hemisphere climate anomalies. *Nature*, **570**, 319–325, doi:10.1038/s41586-019-1294-0.

Carsey, F. D., 1980: Microwave observation of the Weddell Polynya. *Mon. Weather Rev.*, **108**, 2032–2044, doi:10.1175/1520-0493(1980)108<2032:MOOTWP>2.0.CO;2.

Cheon, W. G., and A. L. Gordon, 2019: Open-ocean polynyas and deep convection in the Southern Ocean. *Sci. Rep.*, **9**, 6935, doi:10.1038/s41598-019-43466-2.

——, Y.-G. Park, J. R. Toggweiler, and S.-K. Lee, 2014: The relationship of Weddell Polynya and open-ocean deep convection to the Southern Hemisphere westerlies. *J. Phys. Oceanogr.*, **44**, 694–713, doi:10.1175/JPO-D-13-0112.1.

——, S.-K. Lee, A. L. Gordon, Y. Liu, C.-B. Cho, and J. J. Park, 2015: Replicating the 1970s' Weddell Polynya using a coupled ocean-sea ice model with reanalysis surface flux fields. *Geophys. Res. Lett.*, **42**, 5411–5418, doi:10.1002/2015GL064364.

——, C.-B. Cho, A. L. Gordon, Y. H. Kim, and Y.-G. Park, 2018: The role of oscillating Southern Hemisphere westerly winds: Southern Ocean coastal and open-ocean polynyas. *J. Clim.*, **31**, 1053–1073, doi:10.1175/JCLI-D-17-0237.1.

Comiso, J. C., and A. L. Gordon, 1987: Recurring polynyas over the Cosmonaut Sea and the Maud Rise. *J. Geophys. Res.*, **92**, 2819–2833, doi:10.1029/JC092iC03p02819.

Francis, D., C. Eayrs, J. Cuesta, and D. M. Holland, 2019: Polar cyclones at the origin of the reoccurrence of the Maud Rise Polynya in austral winter 2017. *J. Geophys. Res. Atmos.*, **124**, 5251–5267, doi:10.1029/2019JD030618.

——, K. S. Mattingly, M. Temimi, R. Massom, and P. Heil, 2020: On the crucial role of atmospheric rivers in the two major Weddell Polynya events in 1973 and 2017 in Antarctica. *Sci. Adv.*, **6**, eabc2695, doi:10.1126/sciadv.abc2695.

Gordon, A. L., 1978: Deep Antarctic convection west of Maud Rise. *J. Phys. Oceanogr.*, **8**, 600–612, doi:10.1175/1520-0485(1978)008<0600:DACWOM>2.0.CO;2.

——, and B. A. Huber, 1990: Southern Ocean winter mixed layer. *J. Geophys. Res.*, **95**, 11655–11672, doi:10.1029/JC095iC07p11655.

Heuzé, C., L. Zhou, M. Mohrmann, and A. Lemos, 2021: Spaceborne infrared imagery for early detection of Weddell Polynya opening. *Cryosph.*, **15**, 3401–3421, doi:10.5194/tc-15-3401-2021.

Heygster, G., M. Huntemann, N. Ivanova, R. Saldo, and L. T. Pedersen, 2014: Response of passive microwave sea ice concentration algorithms to thin ice. *2014 IEEE Geoscience and Remote Sensing Symposium*, 3618–3621.

Holland, D. M., 2001: Explaining the Weddell Polynya—a large ocean eddy shed at Maud Rise. *Science*, **292**, 1697–1700, doi:10.1126/science.1059322.

Huntemann, M., G. Heygster, L. Kaleschke, T. Krumpen, M. Mäkynen, and M. Drusch, 2014: Empirical sea ice thickness retrieval during the freeze-up period from SMOS high incident angle observations. *Cryosph.*, **8**, 439–451, doi:10.5194/tc-8-439-2014.

Jena, B., M. Ravichandran, and J. Turner, 2019: Recent reoccurrence of large open-ocean polynya on the Maud Rise polynya. *Geophys. Res. Lett.*, **46**, 4320–4329, doi:10.1029/2018GL081482.

Kurtakoti, P., M. Veneziani, A. Stössel, and W. Weijer, 2018: Preconditioning and formation of Maud Rise polynyas in a high-resolution earth system model. *J. Clim.*, **31**, 9659–9678, doi:10.1175/JCLI-D-18-0392.1.

——, ——, ——, ——, and M. Maltrud, 2021: On the generation of Weddell Sea polynyas in a high-resolution earth system model. *J. Clim.*, **34**, 2491–2510, doi:10.1175/JCLI-D-20-0229.1.

Lindsay, R. W., D. M. Holland, and R. A. Woodgate, 2004: Halo of low ice concentration observed over the Maud Rise seamount. *Geophys. Res. Lett.*, **31**, L13302, doi:10.1029/2004GL019831.

Martinson, D. G., 1990: Evolution of the Southern Ocean winter mixed layer and sea ice: Open ocean deepwater formation and ventilation. *J. Geophys. Res.*, **95**, 11641–11654, doi:10.1029/JC095iC07p11641.

——, P. D. Killworth, and A. L. Gordon, 1981: A convective model for the Weddell Polynya. *J. Phys. Oceanogr.*, **11**, 466–488, doi:10.1175/1520-0485(1981)011<0466:ACMFTW>2.0.CO;2.

McPhee, M. G., and Coauthors, 1996: The Antarctic Zone Flux Experiment. *Bull. Am. Meteorol. Soc.*, **77**, 1221–1232, doi:10.1175/1520-0477(1996)077<1221:TAZFE>2.0.CO;2.

——, C. Kottmeier, and J. H. Morison, 1999: Ocean heat flux in the central Weddell Sea during winter. *J. Phys. Oceanogr.*, **29**, 1166–1179, doi:10.1175/1520-0485(1999)029<1166:OHFITC>2.0.CO;2.

Morales Maqueda, M. A., A. J. Willmott, and N. R. T. Biggs, 2004: Polynya dynamics: a review of observations and modeling. *Rev. Geophys.*, **42**, RG1004, doi:10.1029/2002RG000116.

Motoi, T., N. Ono, and M. Wakatsuchi, 1987: A mechanism for the formation of the Weddell Polynya in 1974. *J. Phys. Oceanogr.*, **17**, 2241–2247, doi:10.1175/1520-0485(1987)017<2241:AMFTFO>2.0.CO;2.

Muench, R. D., J. H. Morison, L. Padman, D. G. Martinson, P. Schlosser, B. A. Huber, and R. Hohmann, 2001: Maud Rise revisited. *J. Geophys. Res.*, **106**, 2423–2440, doi:10.1029/2000JC000531.

Paţilea, C., G. Heygster, M. Huntemann, and G. Spreen, 2019: Combined SMAP–SMOS thin sea ice thickness retrieval. *Cryosph.*, **13**, 675–691, doi:10.5194/tc-13-675-2019.

Sirevaag, A., M. G. McPhee, J. H. Morison, W. J. Shaw, and T. P. Stanton, 2010: Wintertime mixed layer measurements at Maud Rise, Weddell Sea. *J. Geophys. Res.*, **115**, C02009, 1–15, doi:10.1029/2008JC005141.

Spreen, G., L. Kaleschke, and G. Heygster, 2008: Sea ice remote sensing using AMSR-E 89-GHz channels. *J. Geophys. Res. Ocean.*, **113**, 1–14, doi:10.1029/2005JC003384.

de Steur, L., D. M. Holland, R. D. Muench, and M. G. McPhee, 2007: The warm-water "halo" around Maud Rise: Properties, dynamics and impact. *Deep Sea Res. Part I Oceanogr. Res. Pap.*, **54**, 871–896, doi:10.1016/j.dsr.2007.03.009.

Swart, S., and Coauthors, 2018: Return of the Maud Rise polynya: Climate litmus or sea ice anomaly? [in "State of the Climate in 2017"]. *Bull. Am. Meteorol. Soc.*, **99**, S188–S189, doi:10.1175/2018BAMSStateoftheClimate.1.

Venegas, S. A., and M. R. Drinkwater, 2001: Sea ice, atmosphere and upper ocean variability in the Weddell Sea, Antarctica. *J. Geophys. Res.*, **106**, 16747–16765, doi:10.1029/2000JC000594.

Wilson, E. A., S. C. Riser, E. C. Campbell, and A. P. S. Wong, 2019: Winter upper-ocean stability and ice-ocean feedbacks in the sea ice-covered Southern Ocean. *J. Phys. Oceanogr.*, **49**, 1099–1117, doi:10.1175/JPO-D-18-0184.1.

---

## Author Comment (AC1)

In the following reply, you may find your own comments marked in **gray** and the replies by the authors of the manuscript indented and in **black**.

**General Reply:**

> We would like to thank you for your evaluation and insightful comments. We are pleased to know that this work has the potential to benefit the field of oceanography through remote sensing. This manuscript was written first and foremost to identify what the SMOS and SMOS-SMAP sea ice thickness retrievals pick up when looking at the polynya-prone region. That said, ERA5 atmospheric reanalysis data was also analyzed in parallel leading to the secondary conclusion that corroborates past studies on atmospheric influences over the Maud Rise region.

i) Acknowledgment of relevant past studies is inadequate: One of the main conclusions from this work is that winter storms in the Weddell Sea can drive substantial basal melting, which may or may not manifest itself as a polynya (i.e. a complete melting of the sea ice cover). While the authors do present fresh evidence in support of this hypothesis, they fail to acknowledge the long history of this idea in the literature. A well known example is the field report by McPhee et al. 1996, which documents the ANZFLUX field campaign near Maud Rise during the austral winter of 1994. In this report, they describe how the passage of a series of storms led to so much basal ice melting that they abandon their field site on the ice. A more detailed assessment of the results from that research cruise is provided by McPhee et al. (1999). Additionally, a few modeling and theoretical studies have highlighted the importance of wind-driven mixing in generating basal melting in the Weddell sea (e.g. Goosse and Fichefet 2000 and Wilson et al. 2019). With regard to the 2017 polynya event, Francis et al. (2019) and Campbell et al. (2019) both provide detailed accounts of the polar cyclones and atmospheric conditions leading up to the polynya event. While the authors credited Campbell et al. (2019) for highlighting the impact of storms, they did not cite the equally relevant study by Francis et al. (2019). I will restate that this work does provide new and valuable validation of the notion that storms are an important ingredient for the generation of open-ocean polynyas. I also agree with the authors that importance of storms in the generation of polynyas are not acknowledged to the same extent as ocean preconditioning and large scale Ekman pumping. That said, many studies have argued in favor of these storms (i.e. McPhee et al. 1996, McPhee 1999, Goosse and Fichefet 2000, Wilson et al. 2019, and Francis et al. 2019) and they deserve to be acknowledged here.

> We would like to thank you for providing us with more fundamental literature on this topic that has broadened our understanding and helped formulate a better

manuscript. More importantly, we are pleased that this will lead to acknowledging the researchers that have studied this effect extensively.

ii) Description of SMOS-SMAP sea ice thickness retrieval is a bit lacking: While I appreciate the technical discussion provided in the data and methods section, I am still left with an unclear understanding of the sea ice thickness (SIT) retrieval process, its limitations, and the measurement uncertainties associated with the data product. Since similar SIT data have been available in the Arctic for several years now, the big question in my mind is why did it take so long to apply this technique to Antarctic sea ice. My understanding is that Antarctic sea ice has the added complication of snow-loading, where the weight of snow depresses the sea ice surface below sea level, leading to the formation of a snow-ice slush that has poorly defined densities. However, this issue is only briefly mentioned and immediately cast aside in the discussion of the results (line 275). I also don't fully comprehend the connection between the errors in SIT and sea ice concentration (SIC). In lines 100-111, the authors mention that the SMOS-SMAP retrieval algorithm assumes near 100% SIC and that they need to "correct" for this assumption using predefined SIT/SIC rations. For example, the authors acknowledge that 50 cm thick ice that is retrieved at 90% SIC is adjusted down to 28 cm (line 107). I am surprised by the magnitude of this adjustment and makes me question the validity of these results. What's even more puzzling is the assertion that these SIT values should not be combined with SIC to estimate ice volume. Overall, I think the methods section needs one or two additional paragraphs that give a more general (i.e. less technical) overview of the SIT retrieval process and the limitations and uncertainties associated with these estimates. I anticipate most readers will not be familiar with the details of these satellite retrievals and data processing, so this is an opportunity for the authors to educate the community on how best to use and interpret this valuable dataset.

With regard to the sea ice thickness (SIT) retrieval, the data product was developed primarily with the Arctic in mind and so most of the validation tests presented in the relevant papers (Huntemann et al. 2014, Paţilea et al. 2019), both ship-based and airborne data, have been done in Arctic. However, it is expected that the retrieval will work the same in the Antarctic. There are no fundamental differences for thin ice between the Arctic and Antarctic. Yes, flooding due to high snow load is more common in the Antarctic but it is also observed widespread in the Arctic (see e.g. publications about the N-ICE2015 campaign). Such conditions would hinder the retrieval in both hemispheres. However, for thin ice in the ice growth season such high snow loads are not typical. The retrieval was originally developed to assess sea ice thicknesses during freeze-up in near-100% sea ice concentrations. After some discussion prompted by concerns presented from all 3 reviewers, we decided for the sake of this manuscript to treat the SMOS and SMOS-SMAP SIT as apparent sea ice

thickness. We think it's necessary to clarify that since we are using this retrieval for a region in which it is not optimized for, the exact values of sea ice thickness are not absolute. Despite the inaccuracy, this retrieval is based on the fact that at L band (1.4 GHz) the radiation is sensitive to SIT up to 50 cm (Kaleschke et al., 2010, 2012). Thus we believe, although not wholly accurate in terms of the exact values presented, it is still worth reporting wide-scale ice thinning near and above Maud Rise. However, computing ice volume from SIT maps that most likely have an inhomogeneous uncertainty distribution would likely result in erroneous results. As such, we want to keep this manuscript focused on the perceived signal and its areal as well as temporal distribution rather than taking the SIT retrieval at face value. We like to emphasize again that these disadvantages of the SIT retrieval are exactly the same for the Arctic and Antarctic. Also in the Arctic one can have little confidence in the absolute SIT values for SIC < 100% and under melting conditions. This is also independent of the particular retrieval used.

The ASI sea ice concentration (SIC) product is not meant to directly validate or correct the SIT product, but instead be used for comparison. This decision originates from previous analysis and comparison of passive microwave SIC algorithms to thin SIT (Ivanova et al. 2015, Heygster et al. 2014). Also from past studies (Huntemann et al. 2014, Paţilea et al. 2019), we know that both SIT retrievals are influenced by varying sea ice concentration, thus if we observed thinning on the same scale and continuously by the same amount for given sea ice concentrations, it would not be very promising. Despite this we have shown that during these thinning events and even polynya events, areas impacted by sea ice thinning are much larger than low sea ice concentration areas. As such, we believe it necessary to show this discrepancy as well report both data products over the span of the 2017 polynya and 2018 sea ice thickness anomaly. In addition, it helps to compare the two to study these events from different perspectives.

**Specific Comments Reply:**

- Line 1: This is minor point but there is some debate in the literature concerning what definition of a Weddell Sea Polynya (WSP). Some authors (e.g. Cheon and Gordon 2019 and Kurtakoti et al. 2020) argue that a WSP is strictly one that grows beyond Maud Rise and occupies much of the Weddell Sea, such as those that occurred in thelate 1970s. They distinguish these polynyas from the smaller Maud Rise Polynyas (MRP), which are largely confined to the vicinity of Maud Rise. By this definition, the 2017 event was not a WSP but a large MRP. While I think the distinction between MRP

and WSP is rather superficial, the authors should make a deliberate decision with their naming convention. If they deem the 2017 event a WSP, I think it should follow that the 2016 event and other smaller events should be considered WSPs. Alternatively, the authors my sidestep this issue all together and just refer to these events generically as open-ocean or offshore polynyas in the Weddell Sea.

> We went with the latter option as it was also suggested by the third reviewer. It is now referred to as the Weddell Sea polynya where "polynya" is lower case. Despite this change, we refer to it as "the Weddell Sea polynya" after clarifying the one we mean rather than specifying location at each instance.

- Lines 20-21: This relates to my previous point about the naming of polynyas. I don't think it's true that "no sizable opening" in the sea ice occurred between 1970s and 2016. There were substantial polynyas openings in the mid 1990s and early 2000s (see figure 5 of Campbell et al. 2019).

> Statements will be refined to acknowledge the presence of openings in the intermediate years.

- Line 30: "...convection pushes up Warm Deep Water..." I think "pushes up" is an awkward word choice. I would replace with "mixes" or "entrains".

> Changed to "entrains".

Lines 30-35: See my major points at the beginning of this review. This section introduces the potential for storms to initiate polynyas but only cites Campbell et al. (2019). Several other studies need to be acknowledged here (e.g. McPhee et al. 1996, McPhee 1999, Goosse and Fichefet 2000, Wilson et al. 2019, and Francis et al. 2019). Francis et al. 2019 in particular gives a fairly detailed account of atmospheric conditions leading up to the opening of the 2017 polynya.

> References will be added.

- Lines 39-50: In this discussion about why Maud Rise is so favorable for polynya formation, I would cite Martinson and Iannuzzi 1998 and Wilson et al. 2019. Both use in situ ocean data to demonstrate that the Maud Rise region has a unique combination of weak stratification and high sub-mixed layer heat content that primes the region for strong wintertime basal melting and deep convection.

> References will be added.

- Lines 43-44: Unclear description. Is the intent here to convey that the seamount affects ocean heat fluxes over an area that is twice the footprint of the seamount itself?

> Yes, that is correct. We will reformulate to clarify this part.

- Line 45: The core of the ACC, as defined by the polar front, is much further north than Maud Rise. This major current here is simply the southeastern limb of the Weddell Gyre.

> Noted, the text you are referring to will be corrected accordingly.

- Line 73: Is there a simple explanation for why this 50 cm thickness limit exist? Is this limit affected by the presence of snow?

See explanation in the answer for (ii). We will add a sentence to clarify the limit at this location in the paper.

- Line 92: First sentence is unclear. Were these algorithms were derived from past studies that have focused estimating Arctic sea ice thickness?

Yes this is correct. Wording will be made clearer.

Line 97: I think these "subtle differences" between the two polar regions are important and should be described in more detail.- Lines 100-110: As I mentioned earlier, these relatively large adjustments associated

with lower than 100% SIC is rather surprising. I think more justification is needed here.

Comments from you and reviewer 3 were noted, and the whole Data/Methods section will be revised to convey our change in approach from SIT to apparent SIT which should make everything clearer hopefully.

- Lines 112-120: Is SIC derived from the SMOS or SMAP sensors?

No it's from the AMSR2 satellite onboard JAXA's GCOM-W1 spacecraft - further clarified in the manuscript.

- Lines 147-148: It might help to label the subplots that referenced here.

For the figure we're referring to here, we decided to split it into 1 figure that simply shows where the location of interest is and another that shows the 2018 maps as you also suggest in your comments about figures. Thus labelling should not be necessary.

- Line 152: After "satellite imagery" add "and in situ ocean data".

The text you proposed has been added.

- Line 155: Here are elsewhere, the patches of low sea-ice thickness are referred to as "polynyas". Strictly-speaking a polynya describes ice-free conditions. Since the overlying ice did not completely melt, a polynya did not form. Another word or phrase is needed to describe these "polynya-like" events.

Sentence changed to: "August and September 2018, shown in the time series plots of Fig. 4, is the time period of interest for this research, where the area that featured a polynya the year prior shows a low SIT anomaly".

- Line 163: See previous comment. "polynya thin ice" is a bit contradictory.

Sentence omitted as the point being made in it was self-explanatory.

- Lines 165-168: Campbell et al. (2019) and Francis (2019) both report these wind Anomalies.

References have been added.

- Line 183: I would replace "more plausible scenario" with something like "a more complete perspective". The 2D versus 3D views of sea ice cover are not contradictory. It remains true that no polynya appeared over Maud Rise in 2018. So in that sense, the shift from record breaking winter polynya in 2017 to no polynya in 2018 was rather abrupt. With this new SIT data, we now see that 2018 represented a waning of polynya favorable conditions rather than an abrupt end. Furthermore, nothing in this study

discredits the hypothesis there may have been local freshening of the surface layers in this region.

Wording changed as suggested.

- Line 197: The discussion here is incomplete. While it is true that the loss of heat would destabilize the water column, the thinning of the sea ice implies melt, which strengthens the stratification. At these low temperatures, the stability of the water column is set by salinity. My suspicion is that this partial melting of the sea ice created a thin halocline that stabilized the water column and suppressed further exchange with the Warm Deep Water layer (Martinson 1990 and Wilson et al. 2019 describe this negative feedback in more detail). The suppression of vertical mixing effectively protected the sea ice from further basal melting, which caused no polynya to occur. I would further speculate that if a stronger storm had passed through the region, we would have seen another polynya reappearance in 2018.

The line indicated was referring to the 2017 polynya, not the 2018 anomaly. In any case, this entire section was re-written on account of reviewer 3 who proposed reading up on the oceanographic literature which identified the inaccuracies in the text.

- Line 198: It's not clear to me how the destablization isolates the water mass above Maud Rise. Please elaborate.
this region.

Rewritten, as mentioned above.

- Line 199: Up until this point, the discussion has been entirely focused on the area above Maud Rise. We have no information about the stability of the water column across the entire Antarctic sea ice zone.

Rewritten, as mentioned above.

- Line 203: This presence of this convection cell is speculative. This heat may have been brought up by strong wind-driven vertical mixing or Ekman divergence.

Rewritten, as mentioned above.

- Lines 210-212: This sentence is hard to follow. I would rephrase for clarity.

Rephrased.

- Line 214: Campbell et al. 2019 and Francis et al. (2019) discuss wind-driven ice divergence.

References have been added.

- Lines 220-221: I don't think by comparing wind conditions in 2017 and 2018, one can make a general conclusion about when winds can or cannot generate a polynya. This evidence is rather anecdotal.

Reworded appropriately. Now written as: "Through the comparison of our SIC data with ERA5 atmospheric data we can speculate what wind conditions are favourable for polynya formation."

- Lines 237-238: Please cite the relevant studies here.

References have been added.
- Line 244: I would replace "anticipate" with "suggested by SIC data"
    The change in wording has been implemented.
- Line 251-252: As I mentioned before, numerous studies have "rigorously" explored the idea that wind-driven mixing may drive substantial melting.
    These studies were mentioned and referenced.
- Line 265-266: I very much agree with this statement. At some point, perhaps not at this
exact line, the authors should stress that the melting of sea ice produces a strong negative feedback that suppresses further entrainment of deep ocean heat. This effect discourages deep convection, which is required to sustain a polynya. A sequence of sufficiently strong storms may override that feedback by entraining enough WDW that completely melts the ice cover and eliminates the pycnocline.
    Information on these processes will be included and mentioned in our Discussion section. Thank you for pointing them out to us.
- Lines 275-278: These uncertainties should be discussed in more detail in the methods section.
    Yes, this will be addressed in the revised Data/Methods section.
- Line 290: I don't understand what is meant by "purely-open ocean polynya".
    Clarification added. Changed to "purely ocean-driven".

Figures:
Figure 1: These plots are a bit counter-intuitive. At first glance, it would appear that upticks in thin sea ice represents new growth from open-ocean conditions, when in fact they represent thinning from thicker sea ice. I would suggest filling in the space between each line with a given color as well as the top region above the green line, which represents "thick ice". Also, why is 30cm instead 50 cm used as the threshold for thin/thick ice? One might also consider normalizing the area such that the lines represent the fractional area covered by sea ice below a certain thickness.
    Excellent suggestion which made things clearer. The figures now follow this basic layout.
Figure 2: A few things here. I would make the zoomed out map of the Antarctic sea ice field a standalone figure and have that be the first figure of the paper. This would help orient the reader before they examine the line plots shown in the current Figure 1. By removing that zoomed out map, the remaining figure can then be arranged in the same format as Figure 3 to facilitate better comparison. Lastly, all these plots should have higher resolution.
    . Resolution has been increased. 2018 side-by-side will also be added.
Figure 4: As in Figure 2, I think plots (b) and (c) would be better visualized if the area above and below these lines were filled with color. The time axis is also difficult read. I

would suggest labelling fewer tick marks (e.g. every 5 days) and making the labels horizontal.

Figure 5: See above.

Here again, the suggested changes really improved the output. Thank you for that.

**References:**

G. Heygster, M. Huntemann, N. Ivanova, R. Saldo and L. T. Pedersen, "Response of passive microwave sea ice concentration algorithms to thin ice," *2014 IEEE Geoscience and Remote Sensing Symposium*, 2014, pp. 3618-3621, doi: 10.1109/IGARSS.2014.6947266.

Ivanova, N., Pedersen, L. T., Tonboe, R. T., Kern, S., Heygster, G., Lavergne, T., Sørensen, A., Saldo, R., Dybkjær, G., Brucker, L., & Shokr, M. (2015). Inter-comparison and evaluation of sea ice algorithms: towards further identification of challenges and optimal approach using passive microwave observations. *The Cryosphere*, *9*(5), 1797–1817. https://doi.org/10.5194/tc-9-1797-2015

Huntemann, M., Heygster, G., Kaleschke, L., Krumpen, T., Mäkynen, M., and Drusch, M.: Empirical sea ice thickness retrieval during the freeze-up period from SMOS high incident angle observations, The Cryosphere, 8, 439–451, https://doi.org/10.5194/tc-8-439-2014, 2014.

Paţilea, C., Heygster, G., Huntemann, M., and Spreen, G.: Combined SMAP–SMOS thin sea ice thickness retrieval, The Cryosphere, 13, 675–691, https://doi.org/10.5194/tc-13-675-2019, 2019.

Kaleschke, L., Maaß, N., Haas, C., Hendricks, S., Heygster, G., and Tonboe, R. T.: A sea-ice thickness retrieval model for 1.4 GHz radiometry and application to airborne measurements over low salinity sea-ice, The Cryosphere, 4, 583–592, https://doi.org/10.5194/tc-4-583-2010, 2010

Kaleschke, L., Tian-Kunze, X., Maaß, N., Mäkynen, M., and Drusch, M. (2012), Sea ice thickness retrieval from SMOS brightness temperatures during the Arctic freeze-up period, *Geophys. Res. Lett.*, 39, L05501, doi:10.1029/2012GL050916.

---

## Author Comment (AC2)

In the following reply, you may find your own comments marked in **gray** and the replies by the authors of the manuscript indented and in **black**.

**General Reply:**

Based on the definition of this type of polynya, open water is obvious indicator of the polynya. Thin ice is an indirect indicator that shows open happened here but now thin ice and will soon become thick ice if the warm water upward does not appear again in a later date. In my opinion, the open water is better mapped by the ASI ice concentration map (6.5km resolution) as indicated in the paper and prior studies. The new thin ice thickness product is 45km and is not a good indicator of the small opening; also there is publication about thin ice thickness retrieval from passive microwave remote sensing such as the AMSE-E/2, that could be up to 6.5km in spatial resolution (see paper Dai et al., 2020, Remote Sens. 2020, 12(9), 1484; https://doi.org/10.3390/rs12091484). However, this paper did not mention this method in their paper at all, indicating some lack in literature review. To confirm and validate their thin ice method for mapping polynya, Sentinel-1 SAR image is a much better approach than the ASI concentration since the Sentinel-1 has much higher spatial resolution (also in Dai et al., 2020 paper and other papers).

> Thank you very much for your comments and clarifications which have led to the improvement of the manuscript. Your concerns on the specific data product chosen for this research are noted. Reading up on the suggested literature on AMSR-E/2 sea ice thickness (SIT) products (Nihashi et al. 2017, Dai et al. 2020) was informative and as a result, it will definitely be mentioned in the finalized manuscript. This manuscript, like the Master's thesis that inspired it, began first and foremost as an application of SMOS sea ice thickness: a retrieval which was mainly developed at the Institute of Environmental Physics (IUP) in Bremen. All the authors of this manuscript are affiliated with IUP and thus SMOS and SMOS-SMAP SIT data products were the most logical choice. In addition, the mentioned retrievals have publicly available datasets with manuscripts dedicated solely to their validation (Huntemann et al. 2014, Paţilea et al. 2019). The theory behind both retrievals is that at L band (1.4 GHz) the radiation is sensitive to SIT up to 50 cm which is reported on and discussed in Kaleschke et al., 2010, 2012 prior to the development of the SIT data products. As such, although this method has plenty of its own uncertainties, these uncertainties have been studied and we can trust in the validity of the retrieved signal.

> In terms of quality and validity of the retrieved SIT, it is difficult to compare the SMOS and SMOS-SMAP SIT data products with the AMSR-E/2 SIT algorithm developed by Nihashi et al. 2017 and used by Dai et al. 2020. However, due to

more extensive literature on the former as well as the above-mentioned reasons, we opt to stay with SMOS and SMOS-SMAP SIT analysis. Keeping in line with the topic of the manuscript, it is also beneficial to be able to discern sea ice thicknesses beyond just 20 cm like in the AMSR-E/2 SIT algorithm (Nihashi et al. 2017, Hwang et al. 2007) as during polynya years, ice thicknesses up to 20 cm are closely confined to the rim of the polynya. Simply put, the laid-out goal of this research was more to apply SMOS and  SMOS-SMAP SIT retrieval to the Maud Rise region rather than the thin SIT study of the Maud Rise Region for which the data product was chosen after the formulation of the hypothesis. Accordingly, no effort was made to develop our own AMSR-E/2 SIT retrieval algorithm for the purposes of this study, and we decided to stick with established data products.

**Specific Comments Reply:**

In the paper, authors made it clear that their SMOS-SMAP retrieval algorithm assumes ~100% SIC, while there is low SIC with polynya. This causes a concern on their results.For example, in their text line 108-110, "Thin ice thickness is … a combined ice area and thickness anomaly and not be used to calculate... ice volume…".

> Due to concerns expressed by all 3 reviewers, we have opted to switch the approach of our SIT analysis to apparent SIT. While we know that the SMOS-signal is associated with thin sea ice, due to the conditions posed by the dynamically changing polynya and Maud Rise anomalies as a mix of older ice with newly formed ice, we cannot take the SIT retrieval at face value. As such computing any ice volume from it would include high levels of spatially inhomogeneous uncertainties that cannot be quantified.

This paper claimed that it is the first time to confirm that wind is a major factor for the Weddell Sea polynya, although I am not sure if they have enough data to confirm this finding from these polynya events 4o years ago. Otherwise, are you so confident that the results from these two years of data can apply to other times?

> With regard to the polynya events, we simply corroborate past studies on the topic. Thanks to reviewers 2 and 3, more of the appropriate studies (i.e. McPhee et al. 1996, McPhee 1999, Goosse and Fichefet 2000) could be rightfully attributed with first insights into wind-driven openings above Maud Rise. As for the 2018 sea ice anomaly, we build off of those studies and speculate based on ERA5 atmospheric reanalysis whether wind-driven forcing could have played a part in the anomaly. Wording that implies all anomalies are wind-driven will be corrected.

they conclude various different factors must occur simultaneously for the polynya to occur. But this statement is not an approved statement. the paper writing needs to improve, I have listed a few in the details below, but many can be found throughout the

paper. One big comment is the section 4 (conclusions). Most of the content in this section should be in the discussion not in the conclusions

> Noted. We will see what we can transfer from the Conclusions to the Discussion.

L2, "fully opened again on …2107", but figure 1 and text shows 2016 opened.

> Noted and corrected.

L3, "lasted until melt" is not clear and confused. Maybe change to "lasted until the summer melting season"? "80 days, 2017," should be "80 days in 2017.", right? are you sure it reached early December before all surrounding ice melted?

> Changed to approximately 80 days (in reality it's more like 81 days before the polynya open water area is integrated into the Southern Ocean). "Lasted until melt" has been clarified as well.

L4, "actually was not the…", what is the subject of this sentence? You missed it.

> The subject was meant to be 2017 itself. We will clarify this part of the abstract in accordance with some suggestions from reviewer3.

L59-61, I have question for this purpose of the study: "we aim to ….", why you want to using the thin ice thickness which is not already existing and will be much coarse resolution as compared with the exiting ASI ice concentration data. AMSR-E/2 can also be used to derive thin ice thickness. Is this your thickness compatible with the AMSR-E/2 Derived?

> To our knowledge there was no comparison study between the AMSR-E/2 derived ice thicknesses and SMOS/SMAP sea ice thickness product. We also could not find the said AMSR-E/2 sea ice thickness product online, so that no comparison could be performed.

L85: a root mean square difference (RMSD)

> Corrected.

L92, "growing sea ice", do you mean "sea ice growing season"?

> Yes, we will modify the text accordingly.

L110-111, I am not sure how long this thin ice would last once the upward of warm deep water stops or weaken, since most of the time the Weddell Sea polynya is an open water area as indicated (for 80 days in 2017). Once the upward of warm deep water stops, thin ice would form and would thicker, also thicker ice from surrounding would come to fill the open and thin ice area soon as I can imagine.

> Noted. Will elaborate on this further.

L141: Can you explain why 2017 and 2018 are chosen?

> Reasons behind the choice were clarified. Sentence was changed to: "For a detailed analysis of the Weddell Sea polynya, 2017, the year in which the largest Weddell Sea polynya of this century occurred, and 2018, the year that followed which as will be shown exhibits anomalous thin ice behaviour, have been chosen."

L143-144: Can you clarify the strength of SMOS-SMAP SIT compared to the SIC datasets?

> Changed to: "Here the advantage of the SMOS-SMAP ice thickness retrieval shows its strength by detecting anomalous sea ice behaviour where traditional sea ice concentration datasets cannot"

L149: Can you mention this spatial resolution of ASI SIC in section 2.2?

> Yes, the spatial resolution as well as more information on the ASI SIC data product will be included.

L169, "….area that is classified as open water". I have question. this means it was 0% open water before Sept 13? if this is the case, then 100% ice covered, then ice thickness should be higher but why it was actually lower compared with after Sept 13? this is not possible. am i wrong?

> On September 12 there was a bit of open water area left (below 80% class in Fig. 4b) while the entire area is ice covered on Sep 13. Technically this means that this area could be refrozen, i.e., it should be detected in the thin ice class (<10cm). However, because of the difference in Resolution of the two products this is not an enforced causality. The reduction of open water area in the footprint of SMOS/SMAP could just increase the retrieved ice thickness. The latter is consistent with what is shown in Fig. 4c as reduction of low SIT area across all SIT classes.

L170, 0% open water should come with high SIC and SIT, right?

> Yes, this is correct, see also the previous answer.

L175: There is no section 3.2. Please check it. Should it be "4. Discussion"?

> Corrected.

L180: Can you add any references here?

> We added the following references…

L191: Can you present any statistical parameters, such as correlation coefficient (R)?
    We will add the following statistical parameters: …

L193-194: I guess the higher resolution of ASI SIC can affect this result. In Figure 3 and Figure 6, it seems that ASI SIC (6.25 km) has a much finer spatial resolution than SMOS SIT (45 km?). If so, the ASI SIC data should underestimate the thin-ice (or open water) area compared to the SMOS SIT data. Maybe you need to discuss the effect of different spatial resolution.
    Since ASI retrieval is estimating open water as a percentage on a sub-footprint scale, as is SMOS and SMAP-SMOS but with SIT, we do not think any sizeable under(or over)-estimation will occur in either data product.

L201-203, sentence "we see… freeze up", please break up…
    This sentence as well as paragraph will be rewritten also due to concerns from reviewer 3.

L213, "east and southeast directions", but the figure 4 does not show these directions.
    Figure 7 referenced for the direction specifications.

L216-219, Figure 7: Can you mark the extent of sea ice anomaly area in Figure 7?
    Thank you for the suggestion. While the exact shape of the polynya, and the anomaly even more so, is difficult to establish. A reference cylinder will be included in the wind charts to guide the viewer.

L226-227, "therefore also…", please cite papers here since it is your finding.
    We will add the following references.

L 237-238: I just wonder if there is any possibility of sea ice advection (e.g. drift of thin sea ice from other regions?). By seeing the time-series video of SIT, you may be able to confirm if those thin ice events are all really "polynya-type" events or they are advection of thin ice from surrounding area.
    More generally, literature on the topic attributes this thinning to melting from below. This is partly confirmed for the 2017 polynya (e.g. Campbell et al. 2019) with in-situ data and the situation can be assumed to be similar for 2018 since the region is prone to such events (Wilson et al. 2019, Holland 2001). The maps chosen were those that best depicted the initial expansion of 2017 as well the wide-scale thinning in 2018, however longer time periods were studied and the videos you are referring to were analysed. As such we can confirm that this is not simply the advected thin ice.

L268-269, the last sentence "Moreover, it is the combination….". this sentence is suspicious, since your paper did not approve it.

> Here we try to corroborate past studies on the topic rather than claim it as our finding, the wording will be fixed and references will be added.

L281, "…low SIC (most likely minor lead openings) is recorded". I really hope the paper use the Sentinel-1 SAR data to validate or confirm…

> In the end, we decided this is beyond the scope of this manuscript and we'll stick with MODIS blue marble imaging for high-resolution comparisons. However we will mention this point in the outlook.

Figure 1: Can you briefly explain how to define/distinguish "polynya events" and "ice thinning anomalies"?

> Done.

Figure 3. what are the resolutions for them? it is really not easy to match the two sets.

> The SMOS/SMAP product is oversampled in a resolution of 12.5km while the size of footprint is about 45km on average (hence the soft contours). The ASI resolution is about 5km and it is resampled into a 6.25km grid.

Figure 4 and Figure 5: I am just curious why you mention 2017 (Figure 5) first, and then 2018 (Figure 4). Would it better to mention 2017 first prior to 2018? In Discussion, you describe 2017 first and then 2018, so it is somehow confusing to read the text and figure together. And same for Figure 7 (2018) and Figure 8 (2017). Also Figure 4b, if 0% SIC for open water, should be 100% sea ice, but figure 4c shows 0% ice from August 11-Sept 4, please explain?

> First point will be addressed, and the flow of the text will be switched to consistently be 2017 discussed first and then 2018. Thank you for that insight; it should improve the structure of the manuscript.
>
> Fig4c shows the area of the different sea ice thickness classes where the thick ice class (>50 cm) is not shown in the plot. This means that entire area was covered by thick ice beyond the thickness sensitivity of the SMOS/SMAP retrieval.

Figure 6, really the SIC and SIT do not match much, except the 2016 and 2017.

> No and they are not meant to, that is partly what this manuscript aims to show that areas of thin ice, in this setting where a lot of the melting is from below, need not be low in terms of sea ice concentration.

Figure 8: Same to Figure 7, can you mark the extent of sea ice anomaly area in Figure 8?

A reference cylinder indicating the location of the polynya will be included.

**References:**

Dai L, Xie H, Ackley SF, Mestas-Nuñez AM. Ice Production in Ross Ice Shelf Polynyas during 2017–2018 from Sentinel–1 SAR Images. Remote Sensing. 2020; 12(9):1484. https://doi.org/10.3390/rs12091484

S. Nihashi, K. I. Ohshima and T. Tamura, "Sea-Ice Production in Antarctic Coastal Polynyas Estimated From AMSR2 Data and Its Validation Using AMSR-E and SSM/I-SSMIS Data," in IEEE Journal of Selected Topics in Applied Earth Observations and Remote Sensing, vol. 10, no. 9, pp. 3912-3922, Sept. 2017, doi: 10.1109/JSTARS.2017.2731995.

Huntemann, M., Heygster, G., Kaleschke, L., Krumpen, T., Mäkynen, M., and Drusch, M.: Empirical sea ice thickness retrieval during the freeze-up period from SMOS high incident angle observations, The Cryosphere, 8, 439–451, https://doi.org/10.5194/tc-8-439-2014, 2014.

Paţilea, C., Heygster, G., Huntemann, M., and Spreen, G.: Combined SMAP–SMOS thin sea ice thickness retrieval, The Cryosphere, 13, 675–691, https://doi.org/10.5194/tc-13-675-2019, 2019.

Kaleschke, L., Maaß, N., Haas, C., Hendricks, S., Heygster, G., and Tonboe, R. T.: A sea-ice thickness retrieval model for 1.4 GHz radiometry and application to airborne measurements over low salinity sea-ice, The Cryosphere, 4, 583–592, https://doi.org/10.5194/tc-4-583-2010, 2010

Kaleschke, L., Tian-Kunze, X., Maaß, N., Mäkynen, M., and Drusch, M. (2012), Sea ice thickness retrieval from SMOS brightness temperatures during the Arctic freeze-up period, *Geophys. Res. Lett.*, 39, L05501, doi:10.1029/2012GL050916.

B. J. Hwang, J. K. Ehn, D. G. Barber, R. Galley and T. C. Grenfel, "Investigations of newly formed sea ice in the Cape Bathurst polynya: 2. Microwave emission", J. Geophys. Res., vol. 112, 2007.

Goosse and Fichefet (2000): Open-ocean convection and polynya formation in a large-scale ice–ocean model. DOI: https://doi.org/10.1034/j.1600-0870.2001.01061.x

McPhee et al. (1996): The Antarctic Zone Flux Experiment. DOI: https://doi.org/10.1175/1520-0477(1996)077<1221:TAZFE>2.0.CO;2

McPhee et al. (1999): Ocean Heat Flux in the Central Weddell Sea during Winter. DOI: https://doi.org/10.1175/1520-0485(1999)029<1166:OHFITC>2.0.CO;2

Campbell, E. C., E. A. Wilson, G. W. K. Moore, S. C. Riser, C. E. Brayton, M. R. Mazloff, and L. D. Talley, 2019: Antarctic offshore polynyas linked to Southern Hemisphere climate anomalies. Nature, 570, 319–325, doi:10.1038/s41586-019-1294-0.

Wilson, E. A., S. C. Riser, E. C. Campbell, and A. P. S. Wong, 2019: Winter upper-ocean stability and ice-ocean feedbacks in the sea ice-covered Southern Ocean. J. Phys. Oceanogr., 49, 1099–1117, doi:10.1175/JPO-D-18- 0184.1.

Holland, D. M., 2001: Explaining the Weddell Polynya—a large ocean eddy shed at Maud Rise. Science, 292, 1697–1700, doi:10.1126/science.1059322.

---

## Author Response (AR1)

In the following reply, you may find your own comments marked in **black** and the replies by the authors of the manuscript indented and in **red**.

**General Reply (to Reviewer 1):**

i) Acknowledgment of relevant past studies is inadequate: One of the main conclusions from this work is that winter storms in the Weddell Sea can drive substantial basal melting, which may or may not manifest itself as a polynya (i.e. a complete melting of the sea ice cover). While the authors do present fresh evidence in support of this hypothesis, they fail to acknowledge the long history of this idea in the literature. A well known example is the field report by McPhee et al. 1996, which documents the ANZFLUX field campaign near Maud Rise during the austral winter of 1994. In this report, they describe how the passage of a series of storms led to so much basal ice melting that they abandon their field site on the ice. A more detailed assessment of the results from that research cruise is provided by McPhee et al. (1999). Additionally, a few modeling and theoretical studies have highlighted the importance of wind-driven mixing in generating basal melting in the Weddell sea (e.g. Goosse and Fichefet 2000 and Wilson et al. 2019). With regard to the 2017 polynya event, Francis et al. (2019) and Campbell et al. (2019) both provide detailed accounts of the polar cyclones and atmospheric conditions leading up to the polynya event. While the authors credited Campbell et al. (2019) for highlighting the impact of storms, they did not cite the equally relevant study by Francis et al. (2019). I will restate that this work does provide new and valuable validation of the notion that storms are an important ingredient for the generation of open-ocean polynyas. I also agree with the authors that importance of storms in the generation of polynyas are not acknowledged to the same extent as ocean preconditioning and large scale Ekman pumping. That said, many studies have argued in favor of these storms (i.e. McPhee et al. 1996, McPhee 1999, Goosse and Fichefet 2000, Wilson et al. 2019, and Francis et al. 2019) and they deserve to be acknowledged here.

- McPhee et al 1996, Goosse and Fichefet 2000, Wilson et al. 2019 and Francis et al. 2019 have been cited and the ANZFLUX field campaign as well as their works mentioned, respectively.
- Thanks to yours and reviewer 3's comments, the wording has been corrected so as not to imply our results are unique and are now instead corroborating these past studies.

ii) Description of SMOS-SMAP sea ice thickness retrieval is a bit lacking: While I appreciate the technical discussion provided in the data and methods section, I am still left with an unclear understanding of the sea ice thickness (SIT) retrieval process, its limitations, and the measurement uncertainties associated with the data product. Since

similar SIT data have been available in the Arctic for several years now, the big question in my mind is why did it take so long to apply this technique to Antarctic sea ice. My understanding is that Antarctic sea ice has the added complication of snow-loading, where the weight of snow depresses the sea ice surface below sea level, leading to the formation of a snow-ice slush that has poorly defined densities. However, this issue is only briefly mentioned and immediately cast aside in the discussion of the results (line 275). I also don't fully comprehend the connection between the errors in SIT and sea ice concentration (SIC). In lines 100-111, the authors mention that the SMOS-SMAP retrieval algorithm assumes near 100% SIC and that they need to "correct" for this assumption using predefined SIT/SIC rations. For example, the authors acknowledge that 50 cm thick ice that is retrieved at 90% SIC is adjusted down to 28 cm (line 107). I am surprised by the magnitude of this adjustment and makes me question the validity of these results. What's even more puzzling is the assertion that these SIT values should not be combined with SIC to estimate ice volume. Overall, I think the methods section needs one or two additional paragraphs that give a more general (i.e. less technical) overview of the SIT retrieval process and the limitations and uncertainties associated with these estimates. I anticipate most readers will not be familiar with the details of these satellite retrievals and data processing, so this is an opportunity for the authors to educate the community on how best to use and interpret this valuable dataset.

- The Data and Methods section has been rewritten with a key difference being that we redefine the sea ice thickness (SIT) data as apparent SIT (ASIT) to convey the inaccuracies that come with applying SMOS-SMAP SIT analysis to a region like Maud Rise that is susceptible to low sea ice concentrations on a regular basis.
  - With this change, we aim to focus more on the distribution as well as recurrence of low SIT anomalies rather than the exact thinning observed at each pixel (which was the original aim of the manuscript).
- The limitations and uncertainties of the retrieval with regard to decreasing SIC and near-50 cm SIT is now covered and made clearer.
- The lack of sea ice volume calculations has been justified.

**Specific Comments Reply (to Reviewer 1):**

- Line 1: This is minor point but there is some debate in the literature concerning what definition of a Weddell Sea Polynya (WSP). Some authors (e.g. Cheon and Gordon 2019 and Kurtakoti et al. 2020) argue that a WSP is strictly one that grows beyond Maud Rise and occupies much of the Weddell Sea, such as those that occurred in thelate 1970s. They distinguish these polynyas from the smaller Maud Rise Polynyas (MRP), which are largely confined to the vicinity of Maud Rise. By this definition, the

2017 event was not a WSP but a large MRP. While I think the distinction between MRP and WSP is rather superficial, the authors should make a deliberate decision with their naming convention. If they deem the 2017 event a WSP, I think it should follow that the 2016 event and other smaller events should be considered WSPs. Alternatively, the authors my sidestep this issue all together and just refer to these events generically as open-ocean or offshore polynyas in the Weddell Sea.

We went with the latter option as it was also suggested by the third reviewer. It is now referred to as the Weddell Sea polynya where "polynya" is lower case. Despite this change, we refer to it as "the Weddell Sea polynya" after clarifying the one we mean rather than specifying location at each instance.

- Lines 20-21: This relates to my previous point about the naming of polynyas. I don't think it's true that "no sizable opening" in the sea ice occurred between 1970s and 2016. There were substantial polynyas openings in the mid 1990s and early 2000s (see figure 5 of Campbell et al. 2019).

Sentence changed to "For the next 40 years the few polynya events are comparatively smaller (Campbell et al., 2019) and often in the form of a low sea ice concentration (SIC) halo around Maud Rise (Lindsay et al., 2004).".

- Line 30: "...convection pushes up Warm Deep Water..." I think "pushes up" is an awkward word choice. I would replace with "mixes" or "entrains".

This sentence was omitted and ocean-driven polynya formation is now covered in detail in paragraph 1 of the introduction.

Lines 30-35: See my major points at the beginning of this review. This section introduces the potential for storms to initiate polynyas but only cites Campbell et al. (2019). Several other studies need to be acknowledged here (e.g. McPhee et al. 1996, McPhee 1999, Goosse and Fichefet 2000, Wilson et al. 2019, and Francis et al. 2019). Francis et al. 2019 in particular gives a fairly detailed account of atmospheric conditions leading up to the opening of the 2017 polynya.

All indicated references were added in the 4th paragraph of the rewritten introduction section that discusses the impact on wind at Maud Rise in more detail.

- Lines 39-50: In this discussion about why Maud Rise is so favorable for polynya formation, I would cite Martinson and Iannuzzi 1998 and Wilson et al. 2019. Both use in situ ocean data to demonstrate that the Maud Rise region has a unique combination of weak stratification and high sub-mixed layer heat content that primes the region for strong wintertime basal melting and deep convection.

Both references were added (line 41-42); this discussion is now the third paragraph of the introduction.

- Lines 43-44: Unclear description. Is the intent here to convey that the seamount affects ocean heat fluxes over an area that is twice the footprint of the seamount itself?

> Yes, that is correct. The sentence was changed to "Muench et al. (2001) go further to state that Maud Rise facilitates an upward transport of Warm Deep Water that affects a sea ice area that is roughly twice the size of the seamount" for clarity.

- Line 45: The core of the ACC, as defined by the polar front, is much further north than Maud Rise. This major current here is simply the southeastern limb of the Weddell Gyre.

> ACC changed to the southeastern limb of the Weddell Gyre.

- Line 73: Is there a simple explanation for why this 50 cm thickness limit exist? Is this limit affected by the presence of snow?

> Explanation for the limit added in paragraph 3 of section 2.1. While the presence of snow is a factor, the main contributor for the cutoff is the penetration depth of the 1.4 GHz frequency used.

- Line 92: First sentence is unclear. Were these algorithms were derived from past studies that have focused estimating Arctic sea ice thickness?

> - Lines 121-123 now briefly summarizes the development process of the precursor SMOS algorithm which was derived through comparison with a Cumulative Freezing Degree Days (CFDD) model as opposed to other SIT products.
> - The line in question was left as is, as the paragraph it begins is simply meant to briefly discuss the improvement of the retrieval by also considering the SMAP satellite.

Line 97: I think these "subtle differences" between the two polar regions are important and should be described in more detail.- Lines 100-110: As I mentioned earlier, these relatively large adjustments associated
with lower than 100% SIC is rather surprising. I think more justification is needed here.

> - Our switch from SIT to ASIT is meant to justify the use of this data product for the region despite the differences mentioned (elaborated on in the re-written section 2.1).
>   - Due to the lack of validation studies that quantify the retrieval uncertainty associated with sea ice differences in the Arctic and the Antarctic, we thought it necessary to make this change.

- Lines 112-120: Is SIC derived from the SMOS or SMAP sensors?

> No, it's from the AMSR2 satellite onboard JAXA's GCOM-W1 spacecraft - further clarified in the manuscript (now mentioned in lines 136-138).

- Lines 147-148: It might help to label the subplots that referenced here.

> For the figure we're referring to here, we decided to split it into 1 figure that simply shows where the location of interest is (now Fig. 1 as you suggested) and another that shows the 2018 maps (now Fig. 4).

- Line 152: After "satellite imagery" add "and in situ ocean data".

> The text you proposed has been added.

- Line 155: Here are elsewhere, the patches of low sea-ice thickness are referred to as

"polynyas". Strictly-speaking a polynya describes ice-free conditions. Since the overlying ice did not completely melt, a polynya did not form. Another word or phrase is needed to describe these "polynya-like" events.

> Sentence changed to: "August and September 2018, shown in the time series plots of Fig. 4, is the time period of interest for this research, where the area that featured a polynya the year prior shows a low SIT anomaly".

- Line 163: See previous comment. "polynya thin ice" is a bit contradictory.

> Sentence omitted as the point being made in it was self-explanatory.

- Lines 165-168: Campbell et al. (2019) and Francis (2019) both report these wind Anomalies.

> References have been added.

- Line 183: I would replace "more plausible scenario" with something like "a more complete perspective". The 2D versus 3D views of sea ice cover are not contradictory. It remains true that no polynya appeared over Maud Rise in 2018. So in that sense, the shift from record breaking winter polynya in 2017 to no polynya in 2018 was rather abrupt. With this new SIT data, we now see that 2018 represented a waning of polynya favorable conditions rather than an abrupt end. Furthermore, nothing in this study discredits the hypothesis there may have been local freshening of the surface layers in this region.

> Wording changed as suggested.

- Line 197: The discussion here is incomplete. While it is true that the loss of heat would destabilize the water column, the thinning of the sea ice implies melt, which strengthens the stratification. At these low temperatures, the stability of the water column is set by salinity. My suspicion is that this partial melting of the sea ice created a thin halocline that stabilized the water column and suppressed further exchange with the Warm Deep Water layer (Martinson 1990 and Wilson et al. 2019 describe this negative feedback in more detail). The suppression of vertical mixing effectively protected the sea ice from further basal melting, which caused no polynya to occur. I would further speculate that if a stronger storm had passed through the region, we would have seen another polynya reappearance in 2018.

> All your speculations are reflected in the re-written discussion section.

- Line 198: It's not clear to me how the destablization isolates the water mass above Maud Rise. Please elaborate.

> The sentence had flaws and upon reading more literature on the topic, was omitted altogether. Thanks in part to you and reviewer 3.

- Line 199: Up until this point, the discussion has been entirely focused on the area above Maud Rise. We have no information about the stability of the water column across the entire Antarctic sea ice zone.

> The sentence was also omitted for the same reasons.

- Line 203: This presence of this convection cell is speculative. This heat may have

been brought up by strong wind-driven vertical mixing or Ekman divergence.

> The concept you mention is now reflected in the re-written discussion section.

- Lines 210-212: This sentence is hard to follow. I would rephrase for clarity.

> This sentence has been omitted in the revision process.

- Line 214: Campbell et al. 2019 and Francis et al. (2019) discuss wind-driven ice divergence.

> References have been added.

- Lines 220-221: I don't think by comparing wind conditions in 2017 and 2018, one can make a general conclusion about when winds can or cannot generate a polynya. This evidence is rather anecdotal.

> Reworded appropriately. Now written as: "Through the comparison of our SIC data with ERA5 atmospheric data we can speculate what wind conditions are favourable for polynya formation."

- Lines 237-238: Please cite the relevant studies here.

> References have been added.

- Line 244: I would replace "anticipate" with "suggested by SIC data"

> The change in wording has been implemented.

- Line 251-252: As I mentioned before, numerous studies have "rigorously" explored the idea that wind-driven mixing may drive substantial melting.

> These studies were mentioned and referenced.

- Line 265-266: I very much agree with this statement. At some point, perhaps not at this exact line, the authors should stress that the melting of sea ice produces a strong negative feedback that suppresses further entrainment of deep ocean heat. This effect discourages deep convection, which is required to sustain a polynya. A sequence of sufficiently strong storms may override that feedback by entraining enough WDW that completely melts the ice cover and eliminates the pycnocline.

> Both your points were mentioned in the revised conclusion section.

- Lines 275-278: These uncertainties should be discussed in more detail in the methods section.

> - The sentence has been moved to data and methods (section 2.1), however no further elaboration on the degree of uncertainty was given.
>   - This is because of the impossibility to definitively assess the spatial and temporal distribution of flooding via remote sensing.
>     - This limitation is now mentioned in section 2.1 as well.

- Line 290: I don't understand what is meant by "purely-open ocean polynya".

> Clarification added. Changed to "purely ocean-driven".

Figures:

Figure 1: These plots are a bit counter-intuitive. At first glance, it would appear that upticks in thin sea ice represents new growth from open-ocean conditions, when in fact they represent thinning from thicker sea ice. I would suggest filling in the space between each line with a given color as well as the top region above the green line, which represents "thick ice". Also, why is 30cm instead 50 cm used as the threshold for thin/thick ice? One might also consider normalizing the area such that the lines represent the fractional area covered by sea ice below a certain thickness.

Excellent suggestion which made things clearer. The figures now follow this basic layout that you suggested.

Figure 2: A few things here. I would make the zoomed out map of the Antarctic sea ice field a standalone figure and have that be the first figure of the paper. This would help orient the reader before they examine the line plots shown in the current Figure 1. By removing that zoomed out map, the remaining figure can then be arranged in the same format as Figure 3 to facilitate better comparison. Lastly, all these plots should have higher resolution.

The resolution was increased. 2018 side-by-side has also been added (now Fig. 4).

Figure 4: As in Figure 2, I think plots (b) and (c) would be better visualized if the area above and below these lines were filled with color. The time axis is also difficult read. I would suggest labelling fewer tick marks (e.g. every 5 days) and making the labels horizontal. Figure 5: See above.

Here too, the figures have been changed as you suggested.

**General Reply (to Reviewer 2):**

Based on the definition of this type of polynya, open water is obvious indicator of the polynya. Thin ice is an indirect indicator that shows open happened here but now thin ice and will soon become thick ice if the warm water upward does not appear again in a later date. In my opinion, the open water is better mapped by the ASI ice concentration map (6.5km resolution) as indicated in the paper and prior studies. The new thin ice thickness product is 45km and is not a good indicator of the small opening; also there is publication about thin ice thickness retrieval from passive microwave remote sensing such as the AMSE-E/2, that could be up to 6.5km in spatial resolution (see paper Dai et al., 2020, Remote Sens. 2020, 12(9), 1484; https://doi.org/10.3390/rs12091484). However, this paper did not mention this method in their paper at all, indicating some lack in literature review. To confirm and validate their thin ice method for mapping polynya, Sentinel-1 SAR image is a much better approach than the ASI concentration since the Sentinel-1 has much higher spatial resolution (also in Dai et al., 2020 paper and other papers).

- We have opted not to cite papers: Nihashi et al. 2017 and Dai et al. 2020

- ○ We believe due to the lack of studies done on the method developed by Nihashi et al. 2017, the mentioning of this paper would lead to more questions rather than answers that will not benefit the scientific discussion at hand
  - ○ A lot of observed thinning was in the 30-50 cm range, which the algorithm from Nihashi et al. 2017 cannot pick up on.
- We have opted to stay with ASI SIC and MODIS for comparison with our ASIT record as using Sentinel-1 SAR images would mostly serve as a visual guide (what we already partly achieve with MODIS)
  - ○ We believe the AMSR-E/2 ASI SIC product can more reliably report on the ice concentration in the area of interest

**Specific Comments Reply (to Reviewer 2):**

In the paper, authors made it clear that their SMOS-SMAP retrieval algorithm assumes ~100% SIC, while there is low SIC with polynya. This causes a concern on their results.For example, in their text line 108-110, "Thin ice thickness is … a combined ice area and thickness anomaly and not be used to calculate... ice volume…".

- Data and Methods (section 2.1) has been rewritten to specify that we care about apparent SIT (ASIT) to observe the spatial and temporal distribution of thinning rather than the degree of thinning itself.

This paper claimed that it is the first time to confirm that wind is a major factor for the Weddell Sea polynya, although I am not sure if they have enough data to confirm this finding from these polynya events 4o years ago. Otherwise, are you so confident that the results from these two years of data can apply to other times?

- The wording that implies we were the first to discover this has been correctly adjusted.
- We now explicitly state that we simply corroborate past studies with the short time period that we ourselves analysed

they conclude various different factors must occur simultaneously for the polynya to occur. But this statement is not an approved statement. the paper writing needs to improve, I have listed a few in the details below, but many can be found throughout the paper. One big comment is the section 4 (conclusions). Most of the content in this section should be in the discussion not in the conclusions

   The conclusions section is written in a way that reflects what we conclude based on the discussion section and the plots and maps shown in the results section. As such we opted not to move over anything so as not to interrupt the flow we established when we first wrote up the manuscript.

L2, "fully opened again on …2107", but figure 1 and text shows 2016 opened.

L3, "lasted until melt" is not clear and confused. Maybe change to "lasted until the summer melting season"? "80 days, 2017," should be "80 days in 2017.", right? are you sure it reached early December before all surrounding ice melted?

> The indicated passage has been rewritten in light of reviewer comments and is now as follows: "After 40 years of intermittent, smaller openings, a larger, more persistent polynya appeared in early September, 2017, and remained open for approximately 80 days until spring ice melt."

L4, "actually was not the…", what is the subject of this sentence? You missed it.

> The subject is the year 2017, the full sentence is now: "2017, however, actually was not the only year the imprint of the polynya could be identified."

L59-61, I have question for this purpose of the study: "we aim to ….", why you want to using the thin ice thickness which is not already existing and will be much coarse resolution as compared with the exiting ASI ice concentration data. AMSR-E/2 can also be used to derive thin ice thickness. Is this your thickness compatible with the AMSR-E/2 Derived?

> - The SMOS-SMAP SIT product was developed in 2019 (Paţilea et al. 2019) and its precursor SMOS SIT retrieval was developed in 2014 (Huntemann et al. 2014), so at the time of writing this manuscript, both data products were already in existence and made available.
> - To our knowledge there was no AMSR-E/2 sea ice thickness product that was available at the time of writing the manuscript.

L85: a root mean square difference (RMSD)

> Now written out.

L92, "growing sea ice", do you mean "sea ice growing season"?

> The indicated sentence now reads: "Both the SMOS-SMAP and SMOS are empirical retrievals that were initially developed for monitoring the sea ice thickness of growing sea ice in the Arctic during freeze-up through comparison with a Cumulative Freezing Degree Days (CFDD) model and thereafter calibration and validation using observations (Huntemann et al., 2014)"

L110-111, I am not sure how long this thin ice would last once the upward of warm deep water stops or weaken, since most of the time the Weddell Sea polynya is an open water area as indicated (for 80 days in 2017). Once the upward of warm deep water

stops, thin ice would form and would thicker, also thicker ice from surrounding would come to fill the open and thin ice area soon as I can imagine.

> The indicated sentence has been omitted.

L141: Can you explain why 2017 and 2018 are chosen?

> Reasons behind the choice were clarified. Sentence was changed to: "For a detailed analysis of the Weddell Sea polynya, 2017, the year in which the largest Weddell Sea polynya of this century occurred, and 2018, the year that followed which as will be shown exhibits anomalous thin ice behaviour, have been chosen."

L143-144: Can you clarify the strength of SMOS-SMAP SIT compared to the SIC datasets?

> Indicated sentence changed to: "Here the advantage of the SMOS-SMAP ice thickness retrieval shows its strength by detecting anomalous sea ice behaviour where traditional sea ice concentration datasets cannot"

L149: Can you mention this spatial resolution of ASI SIC in section 2.2?

> The spatial resolution has been included at the end of the paragraph contained within section 2.2.

L169, "….area that is classified as open water". I have question. this means it was 0% open water before Sept 13? if this is the case, then 100% ice covered, then ice thickness should be higher but why it was actually lower compared with after Sept 13? this is not possible. am i wrong?

> On September 12 there was a bit of open water area left (below 80% class in Fig. 4b) while the entire area is ice covered on Sep 13. Technically this means that this area could be refrozen, i.e., it should be detected in the thin ice class (<10cm). However, because of the difference in Resolution of the two products this is not an enforced causality. The reduction of open water area in the footprint of SMOS/SMAP could just increase the retrieved ice thickness. The latter is consistent with what is shown in Fig. 4c as reduction of low SIT area across all SIT classes.

L170, 0% open water should come with high SIC and SIT, right?

> Yes, this is correct, see also the previous answer.

L175: There is no section 3.2. Please check it. Should it be "4. Discussion"?

> You are correct. This had been corrected.

L180: Can you add any references here?

Appropriate references have been added to the sentence in question.

L191: Can you present any statistical parameters, such as correlation coefficient (R)?

- No, no correlation studies were done nor were they the focus of the manuscript.
- In principle, correlation studies between SIC and SIT would only serve to distinguish the two datasets which we already sufficiently show in the maps and time series.
- Wording that implied that correlation studies were conducted has been removed

L193-194: I guess the higher resolution of ASI SIC can affect this result. In Figure 3 and Figure 6, it seems that ASI SIC (6.25 km) has a much finer spatial resolution than SMOS SIT (45 km?). If so, the ASI SIC data should underestimate the thin-ice (or open water) area compared to the SMOS SIT data. Maybe you need to discuss the effect of different spatial resolution.

Since ASI retrieval is estimating open water as a percentage on a sub-footprint scale, as is SMOS and SMAP-SMOS but with SIT, we do not think any sizeable under(or over)-estimation will occur in either data product.

L201-203, sentence "we see… freeze up", please break up…

The indicated sentence has been omitted and large parts of the section completely rewritten.

L213, "east and southeast directions", but the figure 4 does not show these directions.

Figure 7 referenced for the direction specifications.

L216-219, Figure 7: Can you mark the extent of sea ice anomaly area in Figure 7?

Thank you for the suggestion. While the exact shape of the polynya, and the anomaly even more so, is difficult to establish. A reference oval has been included in the wind charts to guide the viewer.

L226-227, "therefore also…", please cite papers here since it is your finding.

The indicated sentence has been changed to: "Thereafter also turbulent mixing of warm salty water plays an increasing role (Campbell et al., 2019)."

L 237-238: I just wonder if there is any possibility of sea ice advection (e.g. drift of thin sea ice from other regions?). By seeing the time-series video of SIT, you may be able to confirm if those thin ice events are all really "polynya-type" events or they are advection of thin ice from surrounding area.

More generally, literature on the topic attributes this thinning to melting from below. This is partly confirmed for the 2017 polynya (e.g. Campbell et al. 2019) with in-situ data and the situation can be assumed to be similar for 2018 since the region is prone to such events (Wilson et al. 2019, Holland 2001). The maps chosen were those that best depicted the initial expansion of 2017 as well the wide-scale thinning in 2018, however longer time periods were studied and the videos you are referring to were analysed. As such we can confirm that this is not simply the advected thin ice.

L268-269, the last sentence "Moreover, it is the combination….". this sentence is suspicious, since your paper did not approve it.

Here we try to corroborate past studies on the topic rather than claim it as our finding. The inclusion of relevant papers and the buildup to that sentence, as well as the better wording throughout the manuscript, hopefully makes that clear now.

L281, "…low SIC (most likely minor lead openings) is recorded". I really hope the paper use the Sentinel-1 SAR data to validate or confirm…

In the end, we decided this is beyond the scope of this manuscript and we'll stick with MODIS blue marble imaging for high-resolution comparisons. We do not mention Sentinel-1 SAR specifically in the outlook portion of the Conclusions section, but do encourage additional validation and thereby improvement of our SIT retrieval.

Figure 1: Can you briefly explain how to define/distinguish "polynya events" and "ice thinning anomalies"?

Already in the introduction we now have: "...thin ice area anomalies, i.e., thinning of ice on the same scale as the polynya that is subject to similar underlying causes."

Figure 3. what are the resolutions for them? it is really not easy to match the two sets.

The SMOS/SMAP product is oversampled in a resolution of 12.5km while the size of footprint is about 45km on average (hence the soft contours). The ASI resolution is about 5km and it is resampled into a 6.25km grid.

Figure 4 and Figure 5: I am just curious why you mention 2017 (Figure 5) first, and then 2018 (Figure 4). Would it better to mention 2017 first prior to 2018? In Discussion, you describe 2017 first and then 2018, so it is somehow confusing to read the text and figure together. And same for Figure 7 (2018) and Figure 8 (2017). Also Figure 4b, if 0%

SIC for open water, should be 100% sea ice, but figure 4c shows 0% ice from August 11-Sept 4, please explain?

- The flow of the text has been switched to consistently be 2017 discussed first and then 2018. Thank you for that insight; it has added clarity to the structure of the manuscript.
- The figure you are referring to (now Fig. 6) shows the area of the different sea ice thickness classes where the thick ice class (>50 cm) is not shown in the plot. This means that the entire area was covered by thick ice beyond the thickness sensitivity of the SMOS/SMAP retrieval.

Figure 6, really the SIC and SIT do not match much, except the 2016 and 2017.

No and they are not meant to, that is partly what this manuscript aims to show that areas of thin ice, in this setting where a lot of the melting is from below, need not be low in terms of sea ice concentration.

Figure 8: Same to Figure 7, can you mark the extent of sea ice anomaly area in Figure 8?

A reference oval indicating the location of the polynya has been included.

**General Reply (to Reviewer 3):**

Title, Page 1: I would suggest considering a more descriptive title that reflects your paper's broader scope beyond the 2017 polynya. As an example, something like: "Sea ice thinning at Maud Rise identified in SMOS-SMAP record". In any case, please standardize the capitalization of the title that you choose, e.g., for the existing title: "Weddell Sea Polynya analysis using SMOS-SMAP sea ice thickness retrieval".

We went with the latter option and standardized the capitalization for our existing title.

Introduction, Pages 1-3: I am concerned that the Introduction section's summary of Weddell polynya formation begins and ends with papers published in 2019 and 2020. As currently written, it neglects the four-decade-long body of literature on the Weddell polynya phenomenon, particularly on the polynya's relationship with preconditioning, stratification, convection, and eddies at Maud Rise. I recommend, at a minimum, consulting Gordon (1978), Martinson et al. (1981), Motoi et al. (1987), Comiso and Gordon (1987), Gordon and Huber (1990), Martinson (1990), Holland (2001), and de Steur et al. (2007) – none of which are cited in this manuscript – and expanding the Introduction by briefly summarizing at least a few of these foundational papers. Please also take some time to think about how these previous studies relate to your findings

and cite some of them throughout your Discussion and Conclusions sections where appropriate. I make this request as I feel it is critical to reference and build on past work, especially when new science challenges long-held paradigms, as yours does. Furthermore, in addition to being incomplete, I found the Introduction difficult to follow, as it skips around between unrelated points. It would help to organize it by common themes, rather than just summarizing one paper after another. I recommend reorganizing the entire Introduction. A logical order to address topics (~1 paragraph each) might be:

1. Past observed Weddell polynyas
2. How Weddell polynyas are formed by destratification and maintained by convection
3. Mean-state factors that favor/precondition polynyas at Maud Rise (low stratification, high heat fluxes,
4. Taylor column, eddies, etc.)
5. Role of interannual variability (SAM) in preconditioning polynyas, including in 2016-2017
6. Role of subseasonal synoptic variability in triggering polynyas (storms, possibly eddies)

- We admit to the lack of literature review that you pointed out and have spent time reviewing the suggested literature.
- Previously erroneous speculations presented in the manuscript have been rewritten and improved (namely the whole introduction section, the 2017 and 2018 paragraphs in the discussion section and large portions of the conclusions).
- The introduction has been reorganized to a large extent as you had suggested.

Introduction, Page 2, Lines 28-29: Cheon and Gordon (2019) are not the first to attribute Weddell polynyas to weak stratification, but one would not be aware from reading this passage. This explanation goes back to Martinson et al. (1981) and the other papers I've mentioned above.

> The paragraph within the revised introduction section that addresses that topic (point 2 in your laid out plan for the logical order of the introduction section), now cites all the following works: Martinson et al., 1981; de Steur et al., 2007; Wilson et al., 2019; Cheon and Gordon, 2019.

Introduction, Page 2, Lines 30-38: Please follow the suggestions of Reviewer #1 regarding this passage. Others besides Cheon and Gordon (2019) and Campbell et al. (2019) have made the case for storms being important – or even critical – for polynya

formation near Maud Rise, and those studies should be acknowledged. Additionally, the phrasing "admit to" feels overly disparaging towards Cheon and Gordon (2019), who discuss "atmospheric influences" in considerable detail. Please change this phrasing; you could be more specific and say that their study discusses large-scale, climate-related atmospheric effects, but not synoptic scale meteorological variability.

- The paragraph within the revised introduction section that addresses that topic (point 6 in your laid out plan for the logical order of the introduction section), now cites all the following works: e.g., McPhee et al., 1996; Goosse and Fichefet, 2000; Francis et al., 2019; Campbell et al., 2019; Wilson et al., 2019; Heuzé et al., 2021.
- The sentence you are referring to that previously cited Cheon and Gordon (2019) has been removed and no such disparaging terminology is used.

Results, Page 8, Lines 165-167: This was a major finding of both Francis et al. (2019) [see their Fig. 5] and Campbell et al. (2019) [see their Extended Data Fig. 4], as Reviewer #1 mentions. Those papers should be cited.

Both the studies you mention are now briefly summarized and cited in the revised introduction section.

Conclusions, Page 14, Lines 251-253: As Reviewer #1 points out and as my comments throughout have indicated, previous studies have, in fact, put forth "rigorous" explanations regarding the influence of atmospheric perturbations (e.g., storms) on ice melt and polynya formation at Maud Rise. They should be cited here. Also, Cheon and Gordon (2019) is not the only study that has discussed the role of large-scale negative wind stress curl: Campbell et al. (2019) and Cheon et al. (2014, 2015, 2018) have also discussed this.

- researchers that have studied the influence of atmospheric perturbations on ice melt and polynya formation at Maud Rise have been cited.
- We did not mean to claim that our findings are the first in that respect, but admit to how the phrasing in the manuscript could have implied it - said phrasing has now been corrected.

Introduction, Page 2, Lines 37-38: This sentence is almost entirely copied verbatim from two sentences on p. 320 of Campbell et al. (2019), to the extent that this could be considered plagiarism. You must rephrase this. You've also combined two unrelated ideas (divergence of ice preventing stabilization by ice melt, and turbulent mixing leading to heat/salt entrainment) from their study and so the sentence is not coherent as written.

The sentence now reads: "Ice divergence due to strong winds enables rapid ice production and brine rejection preventing stabilization from ice melt as

wind-driven turbulent mixing entrains warm and saline water into the surface mixed layer (Campbell et al., 2019)."

Data/Methods, Page 5, Lines 131-133: This sentence is copied nearly verbatim from Campbell et al. (2019). This is not permissible and could be perceived as plagiarism, like the other instance I mention above. I understand the wish to provide these specific details, but the sentence must be at least rephrased.

The sentence now reads: "Campbell et al. (2019) report that there was sufficient agreement between mean sea level pressure (MSLP) data obtained from the SANAE-AWS weather station and the nearest ERA-Interim grid cell (1979-) for ERA-I to be used in gathering signs of storm activity as it skillfully represented MSLP variability near Maud Rise."

Introduction, Page 2, Line 39: Saying "thus far all discussed preconditioning is... [not] exclusive to... the region of interest" is not accurate: (1) Preconditioning mechanisms have barely been discussed in the preceding paragraphs. (2) All of the preconditioning mechanisms are, in fact, exclusive to the region of interest. The reason that positive SAM fluctuations and an enhanced Weddell gyre circulation result in preconditioning is that they increase the doming of isopycnals in the center of the Weddell gyre, where Maud Rise is located, thus uplifting warm and salty Weddell Deep Water.

The paragraph within the revised introduction section that addresses that topic (point 3 in your laid out plan for the logical order of the introduction section), now makes no such claims.

Data/Methods, Page 4, Lines 92-99 (also Lines 275-277): The lack of published calibration and validation of the SMOS and SMOS-SMAP thin ice thickness retrievals in the Antarctic is hugely concerning to me. Your paper's analysis and conclusions rest on a highly uncertain empirical satellite retrieval for which no published validation has been conducted for sea ice in the Southern Hemisphere. I request that several actions be taken to mitigate this issue:

1. Please be explicit in Line 92 that the retrieval is empirical and that it was developed through comparison with a simple Cumulative Freezing Degree Days (CFDD) model of Arctic sea ice growth, then calibrated and validated using other Arctic sea ice estimates (Huntemann et al. 2014).

This has been briefly summarized in the following sentence from section 2.1: "Both the SMOS-SMAP and SMOS are empirical retrievals that were initially developed for monitoring the sea ice thickness of growing sea ice in the Arctic during freeze-up through comparison with a Cumulative Freezing Degree Days (CFDD) model and thereafter calibration and validation using observations (Huntemann et al., 2014)."

2. Make clear throughout Section 2.1 that the lack of Antarctic validation is a limitation and discuss the reasons why this is the case. For example, discuss the sensitivity of the SMOS/SMOS-SMAP retrieval to overlying snow cover, sea ice salinity, and other ice/snow parameters, and mention the degree to which these parameters differ between the Arctic and Antarctic. Additionally, the retrieval was trained on a CFDD model that applies well to the Arctic but probably not the Antarctic, where ocean heat fluxes are stronger and significantly influence sea ice growth in weakly-stratified regions such as near Maud Rise (see, e.g., Wilson et al. 2019). Please discuss the potential impact of different rates of ice growth in these regions.
   - We have made it clear that we do not claim our SIT algorithm retrieves exact values of thickness.
     - We now instead focus on the temporal and spatial distribution of thinning and refer to our product, for this study, as apparent SIT (ASIT)
   - The effect these differences can have on the product are purely speculative due to the lack of validation studies done in the Antarctic; we can only infer that the uncertainty goes up but do not know the degree to which it does so - as a result we opt to mention these sources of uncertainty but in view of our shift in focus as well as lack of validation choose not to discuss them individually

3. Be explicit that the "minor evaluation tests" mentioned in the Antarctic (Lines 92-96) are unpublished. Mention what region the EM-bird validation tests occurred, over what time period(s), and over what ranges of sea ice thickness. Give more detail about the degree of agreement between the EM-bird and SMOSSMAP data at different sea ice thicknesses within 0-50 cm. Please consider including a figure showing this validation exercise. Importantly, this will help provide a reference for the community as others publish studies using the SMOS-SMAP data for Antarctic research, which is inevitable given that you have publicly released the Antarctic data and it is increasingly being used.

   This manuscript, first and foremost, was meant to be an application of the SMOS and SMOS-SMAP sea ice thickness (SIT) retrieval to the Weddell Sea polynya over the available time frame. As you have already informed yourself from this manuscript and more so from Huntemann et al. 2014, Paţilea et al. 2019 and Kaleschke et al. 2012, this goal is hindered by the fact that we are applying a method that was validated and developed for high ice concentration locations undergoing freeze-up, to a dynamic area that is subject to cycles of melt and refreeze as well as exposing large

areas of open water. Due to concerns expressed by all reviewers, we have now shifted the focus from taking the SIT retrieval at face value, and analyzed the retrieved signal as ASIT. Unfortunately, the inclusion of the EM-bird validation will not aid this research as it was done in areas with thicker ice and was only processed with a preliminary calibration. What is more, most of the evaluated sea ice thicknesses are above 50 cm, which due to the high levels of uncertainty at those thicknesses, were cut out from the finalized and published dataset. For these reasons, we have opted not to include the validation but nevertheless mention it.

4. Add a straightforward figure to this paper, perhaps in the Appendix, showing the uncertainty of the SMOSSMAP retrieval at sea ice thicknesses spanning 0-50 cm, perhaps modeled after Figure 9a in Paţilea et al. (2019). However, note that their Figure 9a (which already shows substantial error of ~30 cm for a thickness of 50 cm) is based on a SIC uncertainty of 5%. Given the 'halo' of low SIC (85-90%) known to be present around Maud Rise (Lindsay et al. 2004), the deviation of SIC from 100% is much larger than 5% in this region. Please account for this larger SIC uncertainty when computing SMOS-SMAP uncertainties for this figure. This will help to address the concerns of both Reviewer #1 and myself.

> We decided low-SIC based SIT uncertainty derivation to be unnecessary in view of our decision to only focus on ASIT. The uncertainties are undoubtedly high due to reasons mentioned above, as such the exact values of sea ice thickness are likely not accurate, especially near the low SIC halo (Lindsay et al. 2004). Nevertheless, it has been shown that at L band (1.4 GHz) the radiation is sensitive to SIT up to 50 cm (Kaleschke et al., 2010, 2012). By comparing to SIC data, we show that the used SIT retrieval isn't simply retrieving low SIT values where the halo is; rather the SIT area is generally larger and with less steep gradients than where the polynya or sea ice anomaly are located in 2017 and 2018, respectively. We also show that the two datasets evolve differently through time. Now as we shift our approach to simply depicting the sea ice thinning area through time rather than trying to assign exact values to said thinning, we believe any further uncertainty analysis is not needed. In conclusion, the Data/Methods section has been rewritten and revised in accordance with the shift from SIT to ASIT and as such does not include any study that quantifies related uncertainties.

5. Clarify in Lines 105-107 whether the SMOS-SMAP SIT data you show in this study are biased high or low at SICs less than 100%. The language you use (e.g., "retrieved sea ice thickness," "50 cm... is just 28 cm") is confusing.

6. Please do not frame the three recent papers that have used SMOS SIT estimates to answer scientific questions in the Antarctic as mitigating the uncertainties in this retrieval (as you imply in Lines 97-99). SMOS SIT data were not the focus of any of those papers, which all referenced a wider variety of data sources and thus were less dependent on the accuracy of the SMOS retrieval.

These papers are no longer mentioned.

Discussion, Pages 10-11, Lines 195-205: As Reviewer #1 points out, these few sentences about the role of the ocean contain incoherent reasoning and baseless speculation. My best advice would be to read and digest more of the oceanographic literature on Weddell polynyas – which, as I mention in previous comments, has almost entirely been neglected in the references of this manuscript – and then to completely reconsider the relevance of ocean processes to the data presented in this study. I will emphasize two key ideas to consider, though I would encourage the authors to build on these by assembling their own reasoning and references:

1. The sea ice thinning observed to precede the 2017 polynya is not necessarily (and, in my judgment, is almost certainly not) associated with ocean "destabilization" or deep convection. Note that sea ice growth in the Antarctic is (approximately) governed by the balance of ocean heat input and atmospheric heat extraction. If ocean heat fluxes become greater, sea ice will cease to grow and may start to melt. A variety of factors can influence ocean heat fluxes:
   - Turbulent mixing due to ice-ocean shear (which itself is greater at high wind speeds) can deepen the mixed layer and entrain warm pycnocline waters.
   - Brine rejection from ice growth densifies and deepens the mixed layer, also resulting in
   - entrainment of warm pycnocline waters. In contrast, freshwater input from ice melt can rapidly shoal the mixed layer, limiting heat fluxes from warm waters below.
   - Upwelling due to large-scale cyclonic winds (Ekman upwelling) or smaller-scale eddies (which may produce "doming" of isopycnals) can uplift warm pycnocline waters into the surface mixed layer.

Without a basic understanding of these three processes, I believe it is not responsible to speculate on the causes of the thin sea ice anomalies that you observe. To better understand these processes, I would recommend carefully reading the very relevant study by Wilson et al. (2019). Note that atmospheric variability may also play a role in modifying the energy balance of sea ice; for

example, see the study on the 2017 polynya published in Science Advances by Francis et al. (2020).

2.  To explain the time evolution and interannual variability of sea ice thickness at Maud Rise, it is important to consider the role of ocean stratification and storms. For example, Campbell et al. (2019) show reduced upper-ocean stratification at Maud Rise in 2016 and 2017 due to a saltier surface layer. Lower stratification means lower resistance to turbulent mixing-driven or brine rejection-driven entrainment of warm pycnocline waters. This could easily explain the thin sea ice anomalies observed preceding the 2017 polynya (more entrainment = larger heat fluxes = thinner ice). However, I hope that you can go further and think about why thin sea ice anomalies are also seen in other years, as well as the sub-seasonal time evolution of these anomalies and their relation to storm perturbations. As McPhee et al. (1996) discovered as the sea ice melted beneath their ice camp during the ANZFLUX experiment, passing storms can strongly affect both the ocean and sea ice over Maud Rise.

The Discussion as well as Conclusions sections have been, for the most part, rewritten to reflect the above-mentioned points after reading through all the suggested literature as well as re-reading through manuscripts that were cited initially.

Discussion, Page 11, Lines 215-216: No, low SIC and strong winds do not directly cause upwelling of warm water. Upwelling is a distinct process from mixing or entrainment. I believe what you are referring to is either turbulent mixing of warm pycnocline water related to high wind speeds, or entrainment of those same warm pycnocline waters related to densification of the mixed layer from cooling or brine rejection. Please see my previous comment on this topic.

This has been corrected and upwelling is no longer erroneously mentioned in the text. This particular sentence has been omitted, and the containing paragraph has been rewritten.

Discussion, Page 11, Lines 226-227: Please cite Cheon and Gordon (2019) and Campbell et al. (2019) for this finding, which is not shown in your study. Change "upwelling" to "upwelling and/or mixing"; note my comments above regarding this distinction.

References have been added and the text updated according to your request.

Conclusions, Page 15, Lines 257-258: I disagree wholeheartedly. I don't see how ocean heat loss during the 2016- 2017 polynyas would preclude the possibility of anomalously large ocean heat fluxes (rather than atmospheric forcing) producing thinner sea ice in

2018. See my comments above about the factors that control ocean heat fluxes, such as upper ocean stratification. Additionally, you do not assess wind speeds in most years of your 2010-2020 record, and so you have not demonstrated that wind speeds preceding/during the 2016-2018 polynyas or ice thinning events were particularly anomalous. Intense storms pass by Maud Rise every year; you have not shown that the atmospheric perturbations in 2016-2018 were more severe than in other years.

As with the Discussion section, the Conclusions will be re-written, this time with the knowledge gathered from all the suggested literature from you. We now see the error in our past speculative discussion from which we have drawn our conclusions, and we will correct them such that they are in line with past research on this topic. Similarly, all cases where we imply that we can generalize our results for all polynya/ice-thinning cases have been deleted or reworded as necessary.

**Specific Comments Reply (to Reviewer 3):**

Abstract, Page 1, Line 2: I'd echo the point made by Reviewer #1 regarding the somewhat controversial nomenclature of "Weddell Sea Polynyas". As they mention, some recent literature makes a distinction between "Maud Rise polynyas" and "Weddell Sea polynyas" (e.g., Kurtakoti et al. 2018, 2021) despite providing no objective quantitative or mechanistic criteria (e.g., size or geographic thresholds, multi-year persistence, distinct formation mechanisms, etc.) to definitively sort a polynya into one category or the other. On the other hand, your use of "Weddell Sea Polynya" as a catch-all designation raises the question of how large or persistent an opening must be to merit this formal, capitalized title. The 2017 polynya reached ~50,000 km2 in size. Could a much smaller opening only 500 km2 (22x22 km) also be considered a "Weddell Sea Polynya"? Many might disagree. How about one that is 5,000 km2? These questions are not hypothetical, as we know this phenomenon occurs over a broad spectrum of size and duration (see, e.g., Campbell et al. 2019; Heuzé et al. 2021). If you want to avoid addressing this issue, one option might be to change "Polynya" to a lowercase "polynya" throughout your manuscript and mention (perhaps in Lines 22-23) that "Weddell Sea polynya" simply refers to any sea ice opening near Maud Rise.

The nomenclature has been changed as you suggested i.e. all instances of Weddell Sea Polynya now have a lowercase polynya. Despite this, we still refer to it as the Weddell Sea polynya, as specifying it as a Weddell Sea polynya near Maud Rise each time seems redundant.

Abstract, Page 1, Lines 2-3: There are a few problems with this sentence:

- It is inaccurate to state that the WSP has been absent for 40 years. This also undermines your paper's argument that a moderate thinning in non-polynya years is notable. As Reviewer #2 notes, other smaller polynyas have appeared over Maud Rise, e.g., in 1980, 1994, 2005, 2016, and arguably in a significant fraction of other years (see Comiso and Gordon 1987; Holland 2001; Muench et al. 2001; Venegas and Drinkwater 2001; de Steur et al. 2007; Campbell et al. 2019; Heuzé et al. 2021). Please consider acknowledging this, e.g., by stating: "After 40 years of intermittent, smaller openings, a larger, longer lasting polynya appeared..."

  Phrasing changed as suggested.
- It is not clear how to precisely and objectively define when the 2017 polynya first appeared or what "fully opened" means. Campbell et al. (2019), for example, trace its origin to two openings that were actually first seen on September 3 and coalesced and grew in size from September 13-18 (see their Extended Data Fig. 8). You could say: "...appeared in early September, 2017..."

  Noted. This was also pointed out by other reviewers; will be clarified.
- To be more specific than "melt," use "spring ice melt season" or similar. As Reviewer #2's comment indicates, a polynya is already open and so it cannot "melt."

  Phrasing changed as requested.
- Consistent with my earlier comment, change "a total of 80 days" to "approximately 80 days."

  Phrasing changed as requested.
- In summary, here's what I would suggest: "After 40 years of intermittent, smaller openings, a larger, longer-lasting polynya appeared in early September, 2017, and remained open for approximately 80 days until spring ice melt."

  Phrasing changed as suggested.

Abstract, Page 1, Lines 8-9: This phrasing ("we present the strong impact storm activity has on sea ice") ignores that Campbell et al. (2019) and Francis et al. (2019) have already made a strong case for storm activity impacting the evolution of the 2016 and 2017 Maud Rise polynyas using similar or identical atmospheric reanalysis data sets, as Reviewer #1 also mentions. Consider changing this to: "... we corroborate previous findings on the strong impact that storm activity can have on sea ice at Maud Rise" (note that this is also avoids implying that your results can be generalized to other sea ice-covered regions, where the impact of storms may be quite different).

  Phrasing changed as suggested.

Abstract, Page 1, Lines 9-10: First, it is unclear what you mean by "direct atmospheric forcing." Second, the grammar of the phrase set off by commas ("... in addition to

oceanographic effects") is not correct. This should be changed to something like, "...
help consolidate the theory that the evolution of Weddell Sea Polynyas is controlled by
atmospheric as well as oceanographic variability" [or 'forcing' or 'effects', whichever is
most appropriate].

> Phrasing changed as suggested.

Introduction, Page 1, Line 20: Cheon and Gordon (2019) is not the appropriate citation
for the 1970s polynya. Please cite Carsey (1980).

> Correct citation used as indicated.

Introduction, Page 1, Line 20: See my comment above on Lines 2-3 and the similar note
from Reviewer #1. Polynyas have appeared in many years since the 1970s, and there is
a substantial amount of literature discussing their past occurrence that is being
neglected here.

> Sentence changed to: " For the next 40 years the few polynya events are
> comparatively smaller (Campbell et al., 2019) and often in the form of a low
> sea ice concentration (SIC) halo around Maud Rise (Lindsay
> et al., 2004)."

Introduction, Page 1, Line 22: "Anything comparable" feels very arbitrary. The 2017
polynya was not much larger than the 2016 polynya, which itself was not much larger
than the 1994 polynya, which was not much larger than the 2005 polynya, and so on
(see Fig. 5 in Campbell et al. 2019). These events exist on a continuum of size and
duration, most likely stemming from similar physical processes, and so arbitrary cutoffs
make little sense. It would be more accurate to just state that the 2016 and 2017 events
were the largest and longest-lived since 1976. Here, Swart et al. (2018), Cheon and
Gordon (2019), Campbell et al. (2019), and Jena et al. (2019) should be cited; note
that Swart et al. should be included as they were first to report on the 2017 event.

> Phrased as suggested; citations included.

Introduction, Page 1, Line 25: The standard citation for classification of the WSP as an
open-ocean or "sensible heat" polynya is Morales Maqueda et al. (2004), not these
three papers from 2019.

> Correct citation used as indicated.

Introduction, Page 2, Line 35: Replace "contributes" with "may contribute", as storms will
not necessarily lead to ice divergence over Maud Rise (it depends on the particular
storm), and both the divergence and mixing were speculative rather than shown directly
by Campbell et al. (2019). However, McPhee et al. (1999), Sirevaag et al. (2010), and
others have directly measured turbulent mixing over Maud Rise.

Replaced as suggested.

Introduction, Page 2, Lines 49-51: Since the 23-year time series referenced here ends in 2001, this wording is confusing. You can just say that "... the mean sea ice concentration (SIC) for the months of July through November shows... (Lindsay et al. 2004)." Please also omit the "... lacks the open water expanse indicative of a polynya" part. A literal "open water expanse" (i.e., 0% SIC) would only occur in a mean SIC field if a polynya occurred 100% of the time during these 23 years, so your wording does not make sense.

Wording corrected as indicated.

Data/Methods, Page 3, Line 73: Please be more explicit here that the SMOS and SMOS-SMAP retrievals cannot estimate sea ice thicknesses greater than 50 cm.

Section 2.1 paragraph 3 now includes a discussion leading up to why this is the case.

Data/Methods, Page 4, Lines 107-108: This is the incorrect citation. Paţilea et al. (2019) do not show this; this result was found by Heygster et al. (2014).

Correct citation used as indicated.

Data/Methods, Page 4, Section 2.2 (Lines 113-120): Please clarify which satellite mission's data the ASI algorithm is applied to, both here and above on Line 67. Otherwise, the reader will assume ASI is being applied to SMOS or SMAP. I am only familiar with ASI being applied to the SSM/I and AMSR-E/2 sensors, not SMOS or SMAP (Spreen et al. 2008; Beitsch et al. 2014). Provide the appropriate citations for the data.

A clarification, in the first sentence of section 2.2, has been added and states the following: "The ARTIST Sea Ice (ASI) algorithm calculates SIC from the difference between brightness temperatures at 89 GHz at vertical and horizontal polarizations which are retrieved by the Advanced Microwave Scanning Radiometer 2 (AMSR2) onboard theGlobal Change Observation Mission-Water (GCOM-W1) satellite."

Data/Methods, Page 4, Line 123: The findings of your study are not the final word on this interesting and complex question that many have tried to address, so please omit "conclusively" and consider changing "answer" to "investigate".

Changed as suggested.

Data/Methods, Page 5, Lines 135-136: This claim about ERA5 improving on ERA-Interim requires a citation.

Citation added.

Results, Page 5, Line 143: Sea ice thinning does not constitute a polynya. A polynya is, by definition, an area of open water. Please change "the polynya is visible ... but does not open completely" to something along the lines of "sea ice thinning is observed over multiple weeks". Please also rephrase this in Lines 155 and 163 below.

Phrasing has been corrected as indicated.

Results, Page 5, Lines 143-144: Your Fig. 1 does not compare the SMOS-SMAP retrieval directly to SIC data, so this sentence is not justified in this section. Please delete.

Removed the sentence after the Fig. 1 reference as suggested.

Results, Page 5, Line 147: For reproducibility, it is important to describe and show precisely the "area of interest" that constitutes your averaging region. The coordinates that you cite do not match with the box you drew in Fig. 2 (left) or the subplots in Fig. 2 (right), which are regions irregular in latitude and longitude. Which is your actual averaging region? Please make sure the black box and subplots correspond precisely to the region you use. If it is indeed irregular in lat/lon, you could, for example, give the coordinates of the northwest and southeast corners. Also, please change negative coordinates (e.g., -3.5°E) to the proper values (e.g., 3.5°W).

Northwest and Southeast corner coordinates given.

Results, Page 5, Line 153: Could you be more descriptive in your summary of Fig. 3? For example, you could mention that it shows a broad gradient of SIT encompassing a larger area on all sides of the polynya than shown in the SIC data, which exhibits a sharper gradient of ice concentrations.

The requested description has been added to the main text and the Fig. 3 description has been extended.

Results, Page 8, Line 164: This is not the correct usage of "preconditioning". With polynyas, preconditioning refers to ocean processes that reduce stratification and/or increase subsurface heat content. This can happen over weeks, months, or years. More accurate here would be to say: "Fig. 5 depicts the Weddell Sea polynya of 2017 as well as the weeks leading up to the event."

Phrasing has been corrected as suggested.

Discussion, Page 9, Line 180: Since it is not established in the literature that the small 1973 polynya directly produced the 1974-76 event, please say "preceded" instead of

"resulted in a much larger iteration of". Also cite Martinson et al. (1981) and Comiso and Gordon (1987), who mention the 1973 polynya.

Text has been corrected and the citations have been added as follows: "This is similar to the 1970s polynya cases, where the 1973 smaller polynya preceded the larger Weddell Sea polynya visible from 1974 to 1976 (e.g., Martinson et al., 1981; Motoi et al., 1987; Comiso and Gordon, 1998; Cheon and Gordon, 2019)." - the inclusion of words "smaller" and "larger" is meant to inform the reader that the 1973 polynya was much more localized than the 1974-1976 polynya.

Discussion, Page 10, Line 189: Neither low SIC area nor thin SIT area peaked on 4-5 August of 2016, from looking at your Fig. A1. Please fix or omit this.

Erroneous statement omitted.

Discussion, Page 10, Line 191: What do you mean by "shows some variability"? This is quite vague. Be specific, if not quantitative. Additionally, I agree with Reviewer #2 that a correlation coefficient should be presented if you are going to state that two time series are "not very correlated". Lastly, please fix the grammar in this sentence.

The word "correlation" has been omitted and the vague phrasing has been removed.

Discussion, Page 11, Lines 195-196: Fix the grammar in this sentence, e.g., "... we see that this opening in sea ice, at first minor, eventually paved the way for the Weddell Sea polynya." Also, do you mean to refer to a small opening that preceded the polynya, or a thinning of sea ice preceding the polynya? I am guessing the latter, but if you intended the former (a small opening preceding the larger opening), please reference Campbell et al. (2019) who also demonstrate this.

Grammar has been fixed. Citation has been added.

Discussion, Page 11, Lines 212-215: Ice may be advected without creating any new open water, i.e., ice advection does not necessarily imply ice divergence. You have not shown or calculated ice divergence, and so what you are claiming (pack ice "broken apart by wind") is unfounded speculation. However, you could reference other studies that have calculated or discussed ice divergence for the 2016 and 2017 polynyas:Campbell et al. (2019) [see their Methods section "Atmospheric reanalysis" and Extended Data Fig. 5] and Francis et al. (2019) [see their p. 10-11].

The indicated sentence has been omitted in the revision process, for reasons you mention, ice advection was no longer the focus of the discussion.

Discussion, Page 11, Lines 220-221: Reviewers #1 and #2 point out that this statement is not justified by your analysis. I agree.

Addressed in reply to other reviewer's comments. Sentence changed to: "Through the comparison of our SIC data with ERA5 atmospheric data we can speculate what wind conditions are favourable for polynya formation."

Discussion, Page 11, Lines 221-222: For your statement regarding 13 September 2017, please cite the prior studies, e.g., "..., corroborating the findings of Campbell et al. (2019) and Francis et al. (2019)."

Cited in a manner that was suggested.

Discussion, Page 11, Lines 222-224: Do you have a figure showing this? If not, please note in parentheses: "(not shown)".

Notice added.

Discussion, Page 11, Lines 228-230: This wording ("instead of") implies that you use SMOS before 2015 and SMOS-SMAP from 2015 onwards in Fig. 1, which contradicts your Fig. 1 caption. Please clarify your wording to make clear which is the case.

Grammar fixed; clarification made. Sentence changed to "Lastly, we use the SMOS SIT retrieval instead of the combined SMOS-SMAP to also include years before 2015 (the year when SMAP was put into orbit) to make a consistent 11 year SMOS SIT time series over the months of July, August, September and October (Fig. 1) that fully includes the freezing periods of the relevant region over the years."

Fig. 8 caption, Page 13: See my comment above on the Abstract regarding the earlier sea ice opening on September 3, 2017. I would say that the polynya "rapidly expanded" on September 13, rather than opened for the first time.

Phrasing changed as suggested.

Conclusions, Pages 14-15: In this section, I suggest that you consider discussing how your findings relate to those of Heuzé et al. (2021), whose recent analysis of past polynya events at Maud Rise is highly relevant. Could the SIT data you present offer an "early detection" system for polynyas, as their study aims to develop? In other words, do you find that SIT anomalies consistently precede sea ice openings at Maud Rise? How much lead time prior to a polynya opening could a SIT-based detection system offer?

That was not the intent of our study thus we can speculate on the usefulness of such an early detection system. Despite this, we now comment on this concept in lines 241-249 as outlook.

Conclusions, Page 14, Line 241: Reference my earlier comments regarding the opening date of the 2017 polynya.

Phrasing changed as indicated earlier.

Conclusions, Page 14, Line 255-256: Campbell et al. (2019) also show this using in situ ocean observations, as well as quantifying the rate of heat loss during the 2016 polynya (see their Fig. 3 and Extended Data Fig. 7).

Citation has been added.

Conclusions, Page 14, Line 261: Since you do not quantitatively test correlations, please change "has the most direct correlation" to "is most directly connected" or something similar.

Phrasing changed as suggested.

Conclusions, Page 14, Lines 264-265: This is not a good summary of the processes that Francis et al. (2020) arguecontributed to formation of the 2017 polynya. Please fix. Their study focuses on how atmospheric rivers changed the energy balance of sea ice and the overlying snow cover to favor surface ice melt and thinning.

Summary changed and corrected to the following: "Also worth mentioning is the work done by Francis et al. (2020) that demonstrate the impact of moisture-carrying atmospheric rivers during polynya years which in addition to increasing snow fall, which effectively decouples the sea ice from the cold atmosphere once precipitated, brings clouds that trap the outgoing long-wave radiation locally resulting in further ice melting."

Conclusions, Page 14, Lines 271-272: The fact that the SMOS-SMAP retrieval is influenced by SIC means that, by definition, it is not independent of SIC. Please change to something along the lines of "an additional source of Information".

Phrasing changed as suggested.

Conclusions, Page 15, Lines 282-283: I don't see how daily SMOS-SMAP snapshots allow monitoring "on a more frequent basis" than existing daily SIC data. Delete.

Erroneous text deleted.

Conclusions, Page 15, Line 290: I agree with Reviewer #1—the phrase "purely-open ocean polynya" is confusing and, in any case, neglects past work that has discussed atmospheric influences on the polynya.

Addressed in reply to other reviewer's comments. Sentence has been changed to: "In conclusion, through comparisons between SIT and ERA5 data we corroborate the idea that the Weddell Sea polynya is not purely ocean-driven and instead also facilitated by direct atmospheric forcing."

Data Availability / Acknowledgements, Pages 15 and 17: Please move the appropriate URLs, DOIs, and information about other data sources (e.g., ERA5) to your Data Availability statement; they do not belong in the Acknowledgments.

Data URLs moved oved as requested, however, an acknowledgement for ERA5 will remain.

Appendix A2 / Fig. A2, Pages 16-17: I'm not sure what you are aiming to illustrate with Fig. A2. It is not referenced in the text of your paper. If you want to include it, please discuss and reference it somewhere in the text; if not, Appendix A2 can be removed. Note that "snippets" is not a formal term; change to "images" or similar.

This figure is now mentioned in the main body of the text as follows: "For a better resolved image of both the Weddell Sea polynya of 2017 and the SIT anomaly of 2018 see Fig. A2."

Technical corrections / typographical comments:

All technical corrections/suggestions were accepted and were implemented as suggested.

**References:**

Dai L, Xie H, Ackley SF, Mestas-Nuñez AM. Ice Production in Ross Ice Shelf Polynyas during 2017–2018 from Sentinel–1 SAR Images. Remote Sensing. 2020; 12(9):1484. https://doi.org/10.3390/rs12091484

S. Nihashi, K. I. Ohshima and T. Tamura, "Sea-Ice Production in Antarctic Coastal Polynyas Estimated From AMSR2 Data and Its Validation Using AMSR-E and SSM/I-SSMIS Data," in IEEE Journal of Selected Topics in Applied Earth Observations and Remote Sensing, vol. 10, no. 9, pp. 3912-3922, Sept. 2017, doi: 10.1109/JSTARS.2017.2731995.

Huntemann, M., Heygster, G., Kaleschke, L., Krumpen, T., Mäkynen, M., and Drusch, M.: Empirical sea ice thickness retrieval during the freeze-up period from SMOS high incident angle observations, The Cryosphere, 8, 439–451, https://doi.org/10.5194/tc-8-439-2014, 2014.

Paţilea, C., Heygster, G., Huntemann, M., and Spreen, G.: Combined SMAP–SMOS thin sea ice thickness retrieval, The Cryosphere, 13, 675–691, https://doi.org/10.5194/tc-13-675-2019, 2019.

Lindsay, R. W., D. M. Holland, and R. A. Woodgate: Halo of low ice concentration observed over the Maud Rise seamount. Geophys. Res. Lett., 31, L13302, doi:10.1029/2004GL019831, 2004.

Martinson, D. G., P. D. Killworth, and A. L. Gordon, 1981: A convective model for the Weddell Polynya. J. Phys. Oceanogr., 11, 466–488, doi:10.1175/1520-0485(1981)011<0466:ACMFTW>2.0.CO;2.

de Steur, L., D. M. Holland, R. D. Muench, and M. G. McPhee, 2007: The warm-water "halo" around Maud Rise: Properties, dynamics and impact. Deep Sea Res. Part I Oceanogr. Res. Pap., 54, 871–896, doi:10.1016/j.dsr.2007.03.009.

Wilson, E. A., S. C. Riser, E. C. Campbell, and A. P. S. Wong, 2019: Winter upper-ocean stability and ice-ocean feedbacks in the sea ice-covered Southern Ocean. J. Phys. Oceanogr., 49, 1099–1117, doi:10.1175/JPO-D-18-0184.1.
Motoi, T., N. Ono, and M. Wakatsuchi, 1987: A mechanism for the formation of the Weddell Polynya in 1974. J. Phys. Oceanogr., 17, 2241–2247, doi:10.1175/1520-0485(1987)017<2241:AMFTFO>2.0.CO;2.

Comiso, J. C., and A. L. Gordon, 1987: Recurring polynyas over the Cosmonaut Sea and the Maud Rise. J. Geophys. Res., 92, 2819–2833, doi:10.1029/JC092iC03p02819.

McPhee, M. G., and Coauthors, 1996: The Antarctic Zone Flux Experiment. Bull. Am. Meteorol. Soc., 77, 1221– 1232, doi:10.1175/1520-0477(1996)077<1221:TAZFE>2.0.CO;2.

Cheon, W. G., and A. L. Gordon, 2019: Open-ocean polynyas and deep convection in the Southern Ocean. Sci. Rep., 9, 6935, doi:10.1038/s41598-019-43466-2.

Francis, D., C. Eayrs, J. Cuesta, and D. M. Holland, 2019: Polar cyclones at the origin of the reoccurrence of the Maud Rise Polynya in austral winter 2017. J. Geophys. Res. Atmos., 124, 5251–5267, doi:10.1029/2019JD030618.

Heuzé, C., L. Zhou, M. Mohrmann, and A. Lemos, 2021: Spaceborne infrared imagery for early detection of Weddell Polynya opening. Cryosph., 15, 3401–3421, doi:10.5194/tc-15-3401-2021.

Campbell, E. C., E. A. Wilson, G. W. K. Moore, S. C. Riser, C. E. Brayton, M. R. Mazloff, and L. D. Talley, 2019: Antarctic offshore polynyas linked to Southern Hemisphere climate anomalies. Nature, 570, 319–325, doi:10.1038/s41586-019-1294-0.

Goosse, H. and Fichefet, T.: Importance of ice-ocean interactions for the global ocean circulation: A model study, Journal of Geophysical Research Oceans, 104, 23 337–23 355, https://doi.org/https://doi.org/10.1029/1999JC900215, 2000.

Kaleschke, L., Maaß, N., Haas, C., Hendricks, S., Heygster, G., and Tonboe, R. T.: A sea-ice thickness retrieval model for 1.4 GHz radiometry and application to airborne measurements over low salinity sea-ice, The Cryosphere, 4, 583–592, https://doi.org/10.5194/tc-4-583-2010, 2010

Kaleschke, L., Tian-Kunze, X., Maaß, N., Mäkynen, M., and Drusch, M. (2012), Sea ice thickness retrieval from SMOS brightness temperatures during the Arctic freeze-up period, *Geophys. Res. Lett.*, 39, L05501, doi:10.1029/2012GL050916.

---

## Author Response (AR2)

**Reviewer 1 point-by-point reply**

All replies from the authors are written in **dark red font**, whereas the original text from the comments is in black.

I thank the authors for thoughtfully addressing the concerns raised in the first round of reviews. While I am generally satisfied with the revised manuscript, which has substantially improved, I think there are several issues that need to be addressed before publication can be recommended. My main issue is with the revised Introduction section, where the authors attempt to summarize the oceanographic literature on polynya formation. As I detail below, the authors introduced statements that I view as misleading and inaccurate.

That said, I think the issues mentioned above are relatively minor and the scientific core of the paper remains valuable. While this new sea ice thickness retrieval comes with many caveats, the authors are clear about its limitations and demonstrate that it provides information that is distinct and independent of existing sea ice area information. Future work is needed to validate these retrievals and quantify their uncertainties, but this study is a good start.

Detailed comments:

- Title: This title is fine but many people in the ocean community understand "Weddell Sea Polynya" to mean large open-ocean polynyas that span *most* of the Weddell Sea, which this paper does not analyze. I would suggest a more general title, such as "Analysis of open-ocean polynyas in the Weddell Sea using SMOS-SMAP apparent sea ice thickness retrieval"

> **Thank you for the suggestion but we decided to stick with our original title. While the Weddell Sea polynya is often synonymous with the Weddell polynya, it is to our understanding the latter name is what was first and is most commonly applied to the large polynya events of the 1970s (Carsey, 1980). At the same time, our current title is without an article and therefore aims not to reference any particular polynya event.**

- Line 2 (abstract): Related to the above comment, this definition of "Weddell Sea open-ocean polynyas" does not quite align with definitions found in the literature. While open-ocean polynyas are often found near the Maud Rise seamount, they are not exclusive to this specific region. Perhaps an opening sentence like the following would suffice, "The Weddell Sea is known to feature large openings in its winter sea ice field, otherwise known as open-ocean polynyas...."

> **The first 3 sentences are now written as follows: "The Weddell Sea is known to feature large openings in its winter sea ice field, otherwise known as open-ocean polynyas. An area within the Weddell Sea region that has repeatedly featured open-ocean polynyas in the past is that which encompasses the Maud Rise seamount. Within this area, after 40 years of intermittent, smaller openings, a larger, more persistent polynya appeared in early September, 2017, and remained open for approximately 80 days until spring ice melt."**

- Lines 3-4 (abstract): "2017, however, is far from the only year that the imprint of a polynya can be identified at this location." This is an odd sentence considering the previous sentence states that these polynyas have intermittently appeared over the past 40 years. Perhaps the authors wish to communicate that their new estimates of sea ice thickness reveal periods of the substantial thinning that are not apparent in existing observations of sea ice area/concentration.

> **The sentence has been changed to: "In this study we present proof that polynya-favourable activity in the Maud Rise area is taking place more frequently and on a larger scale than previously assumed."**

- Lines 23-34: I would delete "surprisingly".

> **The sentence has been changed as indicated.**

- Lines 26-28: I would suggest referring to these openings as "open-ocean polynyas". I think the use of lower-case "p" in the term "Weddell Sea polynya" only makes things more confusing.

> **Thank you for the suggestion. Since in the first paragraph of the introduction we define what we mean by that term, we believe it's sufficient as is. Changing the "P" to lower-case was suggested by reviewer 3 of the original submission in an effort to further standardize the nomenclature without referring to any one event. We believe for this study, including the Weddell Polynya and Maud Rise Polynya under a common name is necessary, and simply open-ocean polynyas is too vague. Meanwhile, open-ocean polynyas in the Weddell Sea or open-ocean polynyas near Maud Rise are too long to repeat regularly.**

- Line 29: "The Weddell Sea polynya is an anomalous opening in sea ice that is generally classified as an open-ocean polynya" See above comment.

> **See the above reply.**

- Line 32: "ocean processes" is rather vague. I prefer the original description that mentions the upwelling of warm water.

> **The sentence has been changed to: " An open-ocean or 'sensible heat' polynya is distinguished from the coastal 'latent heat' polynya by being maintained and opened by upwelling and/or mixing as opposed to wind-driven ice advection."**

- Lines 32-34: "The main mechanism that preconditions the Weddell Sea polynya is described by Martinson et al. (1981) as destratification of water masses leading to a raised pycnocline. The raised pycnocline compresses the winter surface mixed layer that is made dense through heat loss and brine rejection from ice formation, resulting in density overturning." This explanation is rather muddled and arguably incorrect. I would delete this sentence since the one that follows gets right message across.

> **The text has been modified as follows: "The main mechanism that preconditions the Weddell Sea polynya is described by Martinson et al. (1981) as the winter surface layer becoming dense through heat loss and brine rejection from ice formation, resulting in density overturning (Martinson et al., 1981; Martinson and Ianuzzi, 1998) which deepens the mixed layer and initiates deep convection. Deep convection and heat**

**ventilation into the mixed layer are thought to be primary causes for the Weddell polynya (Martinson et al., 1981; de Steur et al., 2007; Wilsonet al., 2019; Cheon and Gordon, 2019)"**

- Lines 37-45: This section contains several statements that are confusing, misleading, and at times inaccurate. This is perplexing since the section was mostly fine in the original submission. To give a few specific examples: I don't understand what is meant by "...destratification of water masses leading to a raised pycnocline..." or the idea that a raised pycnocline "compress[es]" the mixed layer. Additionally, the authors appear to use "weak stratification" and "lack of stratification" inter-changeably, which are very different.

**Lack of stratification has been corrected to lack of strong stratification. For the other remarks, see the modified text above.**

Martinson et al. (1981), which the authors try to summarize, proposes that open-ocean polynyas in the Weddell Sea may be initiated when the winter mixed layer becomes sufficiently dense that it triggers deep convection, which results in the upward mixing of warm deep water. These events may be preconditioned by processes that weaken the upper ocean stratification, such as the anomalous upwelling or advection of high salinity water into the surface mixed layer. I don't think the authors need to spend more than a couple sentences reviewing these mechanisms. Like I mentioned before, the initial text was mostly fine.

**The compression of the surface layer is written about in detail in the Martinson et al. 1981 (summarized in Conclusion point 3 and first proposed in the introduction part c. Convective processes), and the paper as a whole was referred to us multiple times by Reviewer 3 who emphasized the importance of original text on the polynya processes. Nevertheless this process is indeed not the primary cause for density overturning and instead more precedence should be given to the salinity increase - thus we thank you for having us correct the order of events as well as better briefly summarize the idea.**

- Lines 44-53: Reading this for a second time, I think this extended discussion about Taylor columns and flow-topography interactions lacks appropriate nuance and does not add much to the paper. With my previous comment in mind, I would also suggest reducing this entire paragraph to a 1-2 sentences. For the purposes of this study, I think it is fine to simply state (with supporting citations) that interactions between the mean flow and topography preconditions the area above Maud Rise for anomalously high vertical heat fluxes, which favors thinner sea ice.

**The extended description has been replaced by "Muench et al. (2001) went further to describe how interactions between the mean flow (eastern limb of the Weddell Gyre) and topography (Maud Rise) preconditions the area above Maud Rise for anomalously high vertical heat fluxes, which favors thinner sea ice." as you suggested.**

- Lines 58-59: "Direct atmospheric forcing" and "large-scale climate processes" are framed as two separate entities but the former is a subset of the latter.

**The term "direct atmospheric forcing" has been changed to "localized atmospheric forcing" to be more precise. What we aim to embody with this is the interaction between the atmosphere and ice occuring in a given location, in this case the Maud Rise area.**

- Lines 117-118: "Patilea et al. (2019) estimated the uncertainty at 90% SIC from SIT values up to 50%." While I get the gist of this discussion, I'm a bit confused by this line.

    **That was a mistake, it should be 50 cm. Thanks for pointing it out. It has now been corrected.**

- Lines 123-124: "...this compromises the validity the of the sea ice thicknesses retrieved..." Typo. "the" after validity is not necessary.

    **The sentence has been changed as indicated.**

- Lines 125-126: "While the degree by which the ice thins is difficult to quantify in terms of uncertainty, our analysis has shown that the pattern of thin ice anomalies above Maud Rise is not random nor is it identical to the distribution of low SIC areas..."

Is there a simple, quantifiable way of demonstrating this statement? I wish I suggested this in my first review, but it would be interesting to compare these SIT measurements with available in situ ocean salinity measurements or perhaps a data-assimilative model such as the Southern Ocean State Estimate. Periods of thinning should correspond to periods of mixed layer freshening. In fairness to the authors, I will not demand that they do this validation but I hope they consider doing so in the future.

    **That would indeed be very useful, and while it is beyond the scope of this manuscript, it will be considered for the future. Some in-situ measurements (moorings deployed in the area by the Alfred Wegener Institute) were analysed in the original master thesis that inspired this manuscript. However, this data was not sufficiently spatially coincident and therefore deemed unreliable when compared to the ASIT record. An additional sentence has been added to the outlook section to encourage further work on this necessary comparison.**

Lines 294-295: "anomalous episode of sea ice" is a bit vague. How about "...episodes of anomalous wintertime sea ice loss..."?

    **The phrasing has been changed as indicated.**

Line 321-322: "In the end, strength of the wind magnitude present above the region is most directly connected with drops in ASIT and SIC." This is an interesting finding!

**Reviewer 2 point-by-point reply**

All replies from the authors are written in **dark red font**, whereas the original text from the comments is in black.

Overall, I see the paper has been improved a lot, although I am not sure why they do not want to comment or discuss the Nihashi et al. 2017 and Dai et al. 2020 papers and cite them. If the method used in these two papers had problems in their opinion, it is a good place/time to say a few words, so future improvements/discussions can be made, right? I notice the English writing has been improved as well, but still need more work. For example, I went through the first part of the paper (up to page 7) and I list quite a few here. I would hope the authors to go through the entire paper and make corrections before final acceptance.

> **We are not opposed to the two mentioned papers. We just don't think citing them here is very relevant for our study. We are aware that substantial work was done for monitoring Antarctic coastal polynyas from space. This includes also some of our own work (Kern, S., G. Spreen, L. Kaleschke, S. de la Rosa Höhn & G. Heygster (2007). Polynya Signature Simulation Method polynya area in comparison to AMSR-E 89,GHz sea-ice concentrations in the Ross Sea and off the Adélie Coast, Antarctica, for 2002-05: first results. Ann. Glaciol., 46, 409-418.), which we also do not cite here. Importantly the open-ocean polynyas we analyse in this manuscript are different to coastal polynyas and methods used to analyse the latter might not be suitable for the former.**

Line 6, to make a consistent tense through the paper, I would think "we found" should be "we find"

> **The sentence has been changed as indicated.**

L7, change "occurring" to "occurred"

> **The sentence has been changed as indicated.**

L16-17, change "effect" to "phenomenon"? "is" to "was", "and similar anomalies are identified" to "but also identified"…

> **The sentence has been changed to: "This phenomenon, as we have shown in the 11-year SMOS record, was not unique to 2018 and was also identified in 2010, 2013 and 2014."**

L23, change "are" to "were"

> **The sentence has been changed as indicated.**

L26, change "will refer to" to "refer"

> **The sentence has been changed as indicated.**

L42, to cite all previous /published papers, I believe the past tense should be used. In this line and many other places please make the changes. "go further" to "went further"…

**All instances of "go further" in this context have been changed as indicated.**

L54, the same as comment above, "describe" to "described"
    **The sentence has been changed as indicated.**

L55, "discuss" to "discussed". Since this already happened, the past tense should be used. Everything you are doing, discussing and finding, should be present tense in this paper. So readers can separate this paper's work (present tense) and previous works (past tense).
    **The sentence has been changed as indicated.**

L63, "go further" to "went further"
    **The sentence is changed as indicated - see above.**

L64, "hypothesize" to "hypothesized"
    **The sentence has been changed as indicated.**

L65, "preventing stabilization" to "preventing that stabilization"
    **The sentence has been changed to: "Ice divergence due to strong winds enables rapid ice production and brine rejection that prevents stabilization from ice melt as wind-driven turbulent mixing entrains warm and saline water into the surface mixed layer."**

L76, add "that" before "the existence of …"; change "conditions that although" to "conditions although"
    **The sentence has been changed to: "Using the ASIT retrieval over years where the polynya did not occur, we aim to identify low sea ice thickness areas that demonstrate anomalous behaviour taking place in the absence of the Weddell Sea polynya suggesting that the existence of polynya-favorable conditions, although present, is insufficient to produce the polynya."**

L77, change "are" to "is"; change "use" to "used"
    **The sentence has been changed as indicated.**

L100, change "which" to "This"
    **The sentence has been changed as indicated.**

L158, change "report" to "reported"
    **The sentence has been changed as indicated.**

Figure 1, based on the 66S and 3E, I don't see the thin ice is on top of the Maud Rise? Do you mean north of the Maud Rise?
    **Figure 1 caption does not claim the sea ice thinning is directly on top of Maus Rise nor does the manuscript. Looking at the introduction as well discussion we**

**describe why we are more likely to expect low SIC and SIT on the flanks of the rise and indeed that is what is shown both in SIC and SIT records.**

L168, change "it the area" to "it is the area"
> **The sentence has been changed as indicated.**

L174, change "shows" to "showed"
> **The sentence has been changed as indicated.**

L175, change "there is" to "there was"
> **The sentence has been changed as indicated.**

**A message to both reviewers:**
**We thank you for reviewing our work for the second time and for once again helping us to improve the manuscript as a whole. Your comments and suggestions throughout the review process were both helpful and appreciated.**